# Reprogramming of bacterial virulence by lysine acetylation

Ole Schmöker [1], Britta Girbardt[1], Sabrina Schulze [1], Gottfried J. Palm [1], Leona Berndt[1], Jens Hoppen[1], Nilüfer Kara [1], Xenia Schöps[1], Ruba Al-Abdulla [2], Klara Garz[2], Heike Junker[2], Sophie Wolfgramm [2], Leif Steil [3], Christian Hentschker[3], Katrin Schoknecht[3], Lea-Maria Mayer [4], Leonie Speth [4], Vanessa Lachmayer[5], Mark Dörr [6], Stefan Kemnitz[7], Stefan Müller[8], Jan-Wilm Lackmann [8], Marcus Krüger [8], Kay Hofmann [5], Uwe T. Bornscheuer [6], Uwe Völker [3], Elke Krüger [2], Vera Kozjak-Pavlovic [4] & Michael Lammers [1] ✉

Gram-negative bacteria use a plethora of virulence factors to infect eukaryotic cells. CE-clan protease-related virulence factors were reported to act as deubiquitinases/ubiquitin-like specific proteases. Some have an additional acetyl-transferase activity. The molecular mechanisms underlying this dual activity and the physiological consequences are only marginally understood. Here, we report crystal structures for the *Simkania negevensis* virulence factor SnCE1 in apo-states and in complex with SUMO1. We confirm SnCE1 acting as an efficient deSUMOylase and discover an intrinsic autoacetyltransferase activity. Acetylation impairs SnCE1 tetramer formation structurally being incompatible with SUMO1 binding. We provide a model for regulation of SnCE1-mediated virulence by lysine acetylation modulating autoproteolytic processing and its subcellular distribution in the host cell. SnCE1 localizes to the endoplasmic reticulum in human cells and increases fragmentation of mitochondria. Our data provide mechanistic insights into how lysine acetylation of virulence factors is used to reprogram virulence adjusting it to the host cells' metabolic state.

Bacteria use a plethora of virulence factors injected into host cells to enable an efficient infection process[1–3]. These virulence factors act at various stages during the infection cycle, i.e., cellular uptake, intracellular maintenance, proliferation and the release from the host cells[4,5]. They often cause damage to the host cells, ultimately resulting in the development of diseases in the infected organisms. Recent data

suggest bacteria employing strategies to reprogram their virulence and to adjust the activity of virulence factors to changing conditions[6].

Secreted virulence factors are often injected into the host cells using sophisticated secretion systems, such as the type III (T3SS) or the type IV (T4SS) secretion system[7–9]. Recently, bacterial virulence factors were described to modulate host-cells ubiquitin(-like) modifications,

[1]Department Synthetic and Structural Biochemistry, Institute of Biochemistry, University of Greifswald, Greifswald, Germany. [2]Institute of Medical Biochemistry and Molecular Biology, University Medicine Greifswald, Greifswald, Germany. [3]Interfaculty Institute for Genetics and Functional Genomics, University of Greifswald, University Medicine Greifswald, Greifswald, Germany. [4]Department of Microbiology, University of Würzburg, Biocenter, Würzburg, Germany. [5]Institute for Genetics, University of Cologne, Cologne, Germany. [6]Department Biotechnology & Enzyme Catalysis, Institute of Biochemistry, University of Greifswald, Greifswald, Germany. [7]Department for High Performance Computing, University of Greifswald, University Computing Center, Greifswald, Germany. [8]Institute for Genetics, University of Cologne, Cologne Excellence Cluster on Cellular Stress Responses in Aging-Associated Diseases (CECAD), Cologne, Germany. ✉e-mail: michael.lammers@uni-greifswald.de

thereby counteracting pathogen clearance and supporting bacterial propagation and survival. Amongst those virulence factors, several bacterial E3 ubiquitin ligases were reported[10–15]. Other effectors such as *Legionella pneumophila* SidE catalyse an unconventional ubiquitination of host cell proteins without the need for an E1 ubiquitin-activating enzyme and E2 ubiquitin -conjugating enzyme[16,17].

According to the MEROPS database, all eukaryotic deubiquitinases (DUBs) using a cysteine as nucleophile belong to the CA-clan[18,19]. In contrast, eukaryotic ubiquitin-like specific proteases (ULPs) cleaving SUMO or NEDD8 chains mostly belong to the CE-clan. With some exceptions discovered so far, many bacterial deubiquitinases/ubiquitin-like specific proteases (ULPs) are assigned to the CE-clan due to their structural similarities[20,21].

Bacterial CE-clan protease-related DUBs show little variety in their cleavage specificity towards ubiquitin (Ub) linkages, with most of them being able to cleave multiple Ub chain types, mostly Lys11-, Lys48-linked chains and preferentially Lys63-linked chains. However, RavD from the genus *Legionella* has a papain-like fold and specificity towards linear Ub chains, while no eukaryotic counterpart was reported to cleave Met1-linked Ub chains[22,23]. Similarly, *Simkania negevensis* Met1-specific DUBs belonging to the CA-clan protease-containing OTU-family and Lys6-specific DUBs such as *Legionella* LotA were discovered[24–26]. Besides, *S. negevensis* encodes a virus tegument-like DUB, Josephin-like DUBs and ULPs, of which one enzyme even reacts with an activity-based probe of ISG15[26–29]. *Burkholderia* encodes a DUB not categorized to any of these groups, i.e., TssM belonging to the eukaryotic ubiquitin-specific protease (USP) DUB family[30,31].

*S. negevensis* encodes for five CE-clan DUBs, i.e., SnCE1-5, but only SnCE1 was found to be active as a DUB and deSUMOylase, similar as described for XopD of the plant pathogen *Xanthomonas campestris*[26,32,33]. These virulence factors belong to the YopJ family named after YopJ from *Yersinia* spp.[34]. They are conserved in a wide variety of pathogenic bacterial species, including the mammalian pathogens *Yersinia* spp., *Salmonella enterica*, *Vibrio parahaemolyticus*, *Aeromonas salmonicida* and the plant pathogens *Ralstonia solanacearum*, *Pseudomonas syringae*, *X. campestris*, *Erwinia amylovora*, as well as *Acidovorax citrulli*[33,35–39]. The high evolutionary conservation indicates these virulence factors target essential signaling pathways involved in host cell immune response and apoptosis. Initially regarded as a DUB targeting multiple signaling pathways YopJ from *Yersinia* ssp. was later confirmed to be a dedicated acetyltransferase (AcT) directly manipulating the mitogen-activated protein kinase and NF-κB pathways[34,40–43].

Further studies revealed that other bacterial species encode YopJ-family members acting as pure DUBs cleaving Ub chains with various preferences for different linkage types or ULPs, i.e., acting as isopeptidases to cleave SUMO chains, NEDD8 chains and/or ISG15 chains[26,33,44]. These enzymes, i.e., *S. enterica* SseL, *Rickettsia* sp. RickULP, *Rickettsia bellii* RickCE, *L. pneumophila* LegCE or *Escherichia coli* ElaD, were intensively characterized structurally and functionally[33,45]. The structures support the CE-clan protease fold and arrangement of the active site architecture, i.e., His-Asp/Glu-Cys in the primary structure. It has been suggested that variable regions (VRs) in structures mediate enzyme- and species-dependent functions such as substrate specificity. The preference of the (iso)peptidases for ubiquitin/SUMO/NEDD8/ISG15 was shown to be mediated by a so-called aromatic gatekeeper residue, i.e., a Trp, Phe or Tyr residue directly following the catalytic His base, sterically restricting active site access to an (iso)peptide bond formed between an (α)-amino group in linear chains, or an (ε)-amino group in substrate lysine side-chains, and an (α)-carboxyl group originating from the C-terminal di-Gly motif of a Ub or ubiquitin-like protein (Ubl)[20,21,26].

Besides enzymes of the YopJ family with dedicated deubiquitinase/ULP- or acetyltransferase-activity, ChlaDUB1 from *Chlamydia trachomatis* was recently reported to be active as DUB and acetyltransferase, i.e., having a dual catalytic activity catalyzing a hydrolase and a condensation reaction with the same active site[44]. ChlaDUB1 was reported to suppress NF-κB signaling by binding to the inhibitor IκB-α, preventing its ubiquitination and subsequent proteasomal degradation[46].

Here we report structure-function analyses on the bacterial virulence factor SnCE1 of the *Chlamydia*-like bacterial pathogen *S. negevensis*[26,47]. *S. negevensis* is an obligate intracellular pathogen linked to respiratory diseases in humans with long-lasting infections[48–52]. It was originally isolated from infected amoeba; however, studies with mammalian cells suggest that it is capable to infect various different cell types[48,53]. *S. negevensis* spends almost the entire developmental life cycle in an intracellular membraneous compartment, the so-called *S. negevensis* containing vacuoles (CVs). From these, it is supposed to inject effector proteins with varying activities into the host cell by the T3SS and/or T4SS[54–56]. Early during *S. negevensis* development elementary bodies (EBs) are formed that are metabolically inactive states of *S. negevensis*, which infect host cells. Upon uptake into the host cells, reticulate bodies (RBs) differentiate from EBs. These RBs are highly metabolically active forms in which *S. negevensis* replicates and sustains for several days before these RBs re-differentiate to EBs and are finally released from the host cells to infect other cells. As described for the SnCE1-related virulence factor ChlaDUB1 from *C. trachomatis*, we identified a dual catalytic activity for SnCE1. We discovered SnCE1 having an autoacetyltransferase (AcT) activity and confirmed earlier reports stating its weak DUB activity, cleaving Lys11-, Lys48- and Lys63-linked Ub chains but acting as an efficient deSUMOylase[26]. We solved several crystal structures of the acetylated wild type SnCE1, of the non-acetylated catalytic mutant SnCE1 C256A in the apo states, and a structure of SnCE1 in complex with SUMO1 occupying the S1 site. These analyses reveal SnCE1 lacks the extended VR-3 shown to mediate the dual AcT/DUB activity in ChlaDUB1[44]. Thus, different molecular strategies underlie this dual AcT/ULP-activity in SnCE1 and ChlaDUB1, respectively. We provide evidence for lysine acetylation of SnCE1, regulating its oligomeric state. Acetylation of SnCE1 stabilizes its monomeric form with consequences on its deSUMOylase activity and modulates its autoproteolytic processing, affecting the subcellular distribution of SnCE1 in the host cell. We show that SnCE1 localizes to the endoplasmic reticulum (ER) and observe an increase in mitochondrial fragmentation in cells expressing this protein. Overall, we provide a model according to which lysine acetylation of CE-clan protease-related virulence factors is used to reprogram bacterial virulence, adjusting it to the host cells´ metabolic state. We show that lysine acetylation enables the precise coordination of the activity of bacterial virulence factors within the host cells, precisely in space and time.

## Results

### Wild type SnCE1 is an active deSUMOylase forming a monomer in solution

*S. negevensis* was recently reported to encode several DUBs representing five DUB classes, including the CE-clan members SnCE1-5[26]. While SnCE1 was reported to act as an efficient deSUMOylase with weak DUB activity, for SnCE2-5 no activity neither as DUB, ULP nor reaction with activity-based probes, i.e., ubiquitin-propargylamide (Ub-PA), SUMO1-PA, SUMO3-PA, ISG15-PA or NEDD8-PA was observed[26,57]. We expressed the catalytic domain of SnCE1-5 in *E. coli* BL21 (DE3) and purified all proteins to a high level of purity in sufficient amounts to perform biochemical and biophysical studies (Supplementary Figs. 1 and 2a). If not otherwise stated, we used the construct SnCE1$_{74-310}$ for performing experiments. This construct lacks a postulated N-terminal single-pass transmembrane helix, which is also present in the related enzyme ChlaDUB1 from *C. trachomatis* (Supplementary Fig. 2b)[58]. Bioinformatic analysis and AlphaFold3 predictions support the presence of potential N-terminal helices

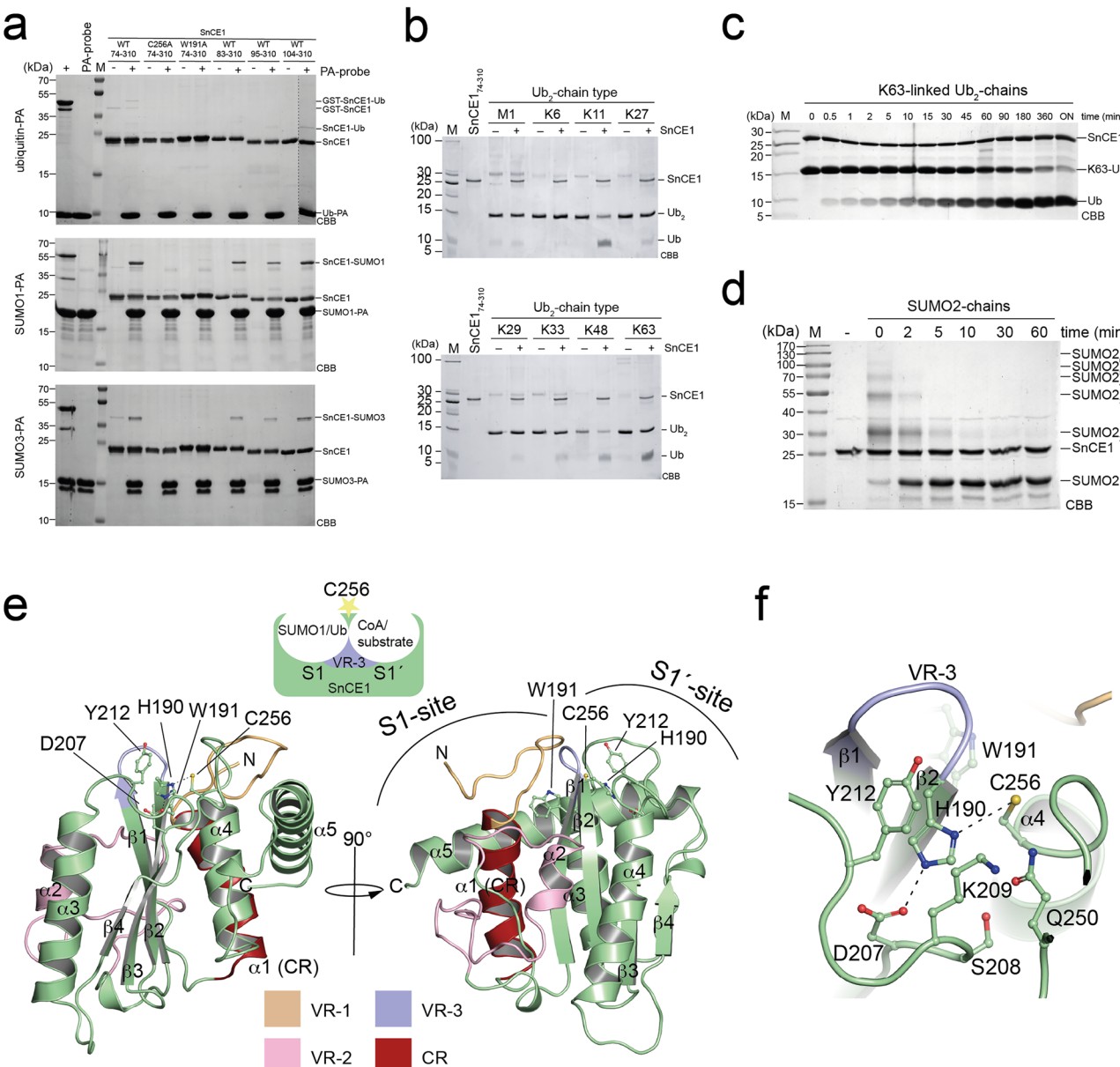

**Fig. 1 | SnCE1 belongs to CE-clan proteases with deubiquitinase and deSU-MOylase activity. a** SnCE1 forms a stable covalent adduct with SUMO1-PA and SUMO3-PA. Catalytically dead mutant SnCE1 C256A and the mutant SnCE1 W191A do not react. The Ub-PA activity-based probe does react weakly with SnCE1. The N-terminus of SnCE1 does not contribute to the reactivity of SnCE1 with the activity-based probes. *E. coli* DUB ElaD and human deSUMOylase SENP1 were used as positive controls. Shown is an SDS-PAGE gel stained with Coomassie brilliant blue (CBB). The experiment was performed in two independent technical replicates (*n* = 2), one example is shown. Source data are provided as Source Data file. **b** SnCE1 has DUB activity towards Lys11-, Lys48- and Lys63-linked di-ubiquitin (diUb) chains with preference for Lys63-linked chains. Shown is an SDS-PAGE gel stained with Coomassie brilliant blue (CBB). The experiment was performed in two independent technical replicates (*n* = 2), one example is shown. Source data are provided as Source Data file. **c** SnCE1 cleaves Lys63-linked diUb-chains with low efficiency. Shown is an SDS-PAGE gel stained with Coomassie brilliant blue (CBB). The

experiment was performed in two independent technical replicates (*n* = 2), one example is shown. Source data are provided as Source Data file. **d** SnCE1 efficiently cleaves polymeric SUMO2[2–6] chains to yield mono-SUMO2. SUMO2 chains are completely cleaved after app. 5–10 min confirming SnCE1 being an efficient deSUMOylase. Shown is an SDS-PAGE gel stained with Coomassie brilliant blue (CBB). The experiment was performed in two independent technical replicates (*n* = 2), one example is shown. Source data are provided as Source Data file. **e** Ribbon representation of the catalytic CE-clan protease-related core of wild type SnCE1. The variable regions 1-3 (VR-1-3) are colour coded (VR-1: orange; VR-2: pink; VR-3: blue). The structure is deposited in the PDB database (PDB: 9QTE). The figure was created with PyMOL[134,135]. **f** Closeup of the SnCE1 active site showing the catalytic triad residues, i.e., the base His190, the acid Asp207 and the nucleophile Cys256. Trp191 acts as gatekeeper mediating Ub/Ubl-specificity. The side chain of Gln250 forms the oxyanion hole. The figure was created with PyMOL[134].

preceding the CE-clan protease (α+β)-fold also in SnCE1 (Supplementary Fig. 2b)[59–61]. Analysis of the oligomeric state by analytical size-exclusion chromatography (SEC) suggests wild type SnCE1 eluting as a monomer (Supplementary Figs. 1 and 2c).

We confirmed, although less efficiently reacting with the Ub-PA probe compared to the SUMO1/3-PA probes, SnCE1 shows weak DUB

activity cleaving Lys11-, Lys48- and Lys63-linked di-ubiquitin (diUb) chains, with preference for the latter (Fig. 1a, b). Even with high concentrations of SnCE1 enzyme used and overnight incubation, we observe only partial cleavage of the Lys63-linked diUb chains (Fig. 1c). Moreover, we also reproduced the reaction of SnCE1 with SUMO1-PA and SUMO3-PA probes with higher efficiency towards SUMO1-PA

compared to SUMO3-PA and its efficient deSUMOylase activity cleaving polymeric SUMO2$_{[2-6]}$ chains to yield mono-SUMO2 (Fig. 1a, d)[26]. Already after 5–10 min we observed full cleavage to monomeric SUMO2 under assay conditions (Fig. 1d). Compared to well-known deSUMOylases human SENP1, human SENP6 and yeast Ulp1, SnCE1 cleaves SUMO2$_{[2-6]}$ chains slightly less efficient (Supplementary Fig. 3a). Presence of a reducing agent such as β-mercaptoethanol or TCEP, the latter not containing a thiol group, was essential for SnCE1 catalytic deSUMOylase activity (Supplementary Fig. 3b). Notably, we generated various N-terminal truncation constructs for SnCE1, i.e., SnCE1$_{74-310}$, SnCE1$_{85-310}$, SnCE1$_{95-310}$ and SnCE1$_{104-310}$, the catalytically inactive mutant SnCE1 C256A and the reported gatekeeper mutant SnCE1 W191A (Fig. 1a). We observed no reaction of the two latter mutants with the PA-probes while all N-terminally truncated proteins reacted as the longer SnCE1$_{74-310}$ validating the reactivity of the SUMO1-/SUMO3-PA probes and proofing the N-terminal region not essential for activity (Fig. 1a). Changes in the pH-value have no influence on the reactivity with the activity-based probes (Supplementary Fig. 4). To understand the mechanisms underlying this activity profile we performed structural analyses.

## SnCE1 adopts CE-clan protease fold lacking an extended VR-3
We obtained crystals for wild type SnCE1$_{104-310}$, which diffracted up to a resolution of 2.8 Å belonging to the tetragonal space group I4 containing three molecules in the asymmetric unit (PDB: 9QTE; Supplementary Table 1, and Supplementary Fig. 5). This structure confirmed the typical CE-clan protease (α+β)-fold, i.e., a central mixed β-sheet at the N-terminus, with the first three β-strands forming an antiparallel β-sheet while β4 is parallel to β3. This part is derived from a β-barrel in CE-clan proteases, which is surrounded by α-helices on both sides, forming an α-helical bundle in CE-clan proteases (Fig. 1e). The active site residues are located at the interface between these domains. The catalytic triad residues appear as His190-Asp207-Cys256 in sequential order in the primary structure with His190 base and Asp207 acid being located in the N-terminal β-subdomain and the Cys256 nucleophile in the C-terminal α-helical domain. This catalytic triad architecture enables a canonical Cys-protease catalytic mechanism (Supplementary Fig. 6). SnCE1 has three variable regions, i.e., VR-1-3 as described for other CE-clan protease-related bacterial virulence factors[33]. VR-1 is located at the N-terminal end of the CE-clan protease domain, VR-2 covers the region preceding β1. VR-3 separates the S1 site, binding the distal SUMO/Ub, and the S1′ site, binding the proximal SUMO/Ub/CoA or non-SUMO/Ub substrate (Fig. 1e). It resides between β1 and β2 in direct vicinity to the catalytic His190 and the gatekeeper Trp191 (Fig. 1e, f). Differences in sequence and structure of these variable regions were shown to be important in determining substrate specificity towards various Ub or Ubl chains and Ub chains connected by various linkage types (Supplementary Fig. 7)[33,44]. Notably, compared to the related enzyme ChlaDUB1 from *C. trachomatis* the α-helix in the extended VR-3 is missing in SnCE1 (Fig. 1e). For ChlaDUB1 this α-helix was shown to mediate the dual DUB and AcT activity[44]. This suggests that if SnCE1 had a dual activity, it should employ alternative catalytic strategies. We continued to explore the catalytic strategies and molecular mechanisms underlying the efficient deSUMOylase activity of SnCE1.

## Crystal structure of SnCE1-SUMO1-PA reveals preference to act as deSUMOylase
To unravel the molecular mechanisms underlying the preference of SnCE1 towards SUMO chains compared to Ub chains we next performed further structure-function analyses (Fig. 2, Supplementary Table 1, and Supplementary Figs. 8 and 9). As shown above, SnCE1 reacted with the SUMO1-/SUMO3-PA probes independent of the N-terminal truncation suggesting the CE-clan protease domain is essential and sufficient for the reaction (Fig. 1a). We next analyzed the cleavage of SUMO2$_{[2-6]}$ chains by these truncated SnCE1 constructs. All

show a similar efficiency in cleaving the chains yielding monomeric SUMO2 (Fig. 2a). Staining for SnCE1 was done using an antibody raised against recombinant SnCE1$_{95-310}$ (Supplementary Fig. 10). As expected, the mutant SnCE1 C256A lost the deSUMOylase activity (Fig. 2b). Notably, for SnCE1 C256A we observed slight remaining activity after 60 min suggesting that Ser208 can replace Cys256 as nucleophile to some extent, however, only in the context of the mutation C256A (Fig. 2b). As support, the double mutant SnCE1 S208A/C256A is catalytically inactive (Fig. 2b).

For structural characterization, we analyzed the complex of SnCE1$_{74-310}$ and SUMO1-PA, as SnCE1 reacted more efficiently with SUMO1-PA compared to SUMO3-PA[26]. We produced SnCE1-SUMO1-PA protein in a preparative scale as described earlier[57,62]. The formation of the covalent vinyl thioether bond resulted in stabilization of the otherwise transient covalent acyl-enzyme intermediate formed during catalysis of proteases/isopeptidases, enabling us to obtain crystals for SnCE1-SUMO1-PA (Fig. 1f, and Supplementary Fig. 9)[57,62]. The crystals diffracted up to a resolution of 1.55 Å and belonged to the orthorhombic space group P2$_1$2$_1$2$_1$ with one molecule of SnCE1-SUMO1-PA per asymmetric unit (PDB: 9QTG). This structure shows the C-terminal propargylamide of SUMO1-PA reacted with the Cys256 nucleophile, confirming the covalent vinyl thioether product (Fig. 2c, and Supplementary Fig. 9). The covalently-bound SUMO1 occupies the S1 site, i.e., it represents the distal SUMO molecule of a SUMO chain, as also observed for Ub in the structure of ChlaDUB1-Ub-PA (Fig. 2c, d)[44]. The structure represents the covalent acyl-enzyme intermediate, which cannot be resolved by a water molecule as nucleophile in this case (Supplementary Fig. 6). In the pre-catalysis state, the S1′ site would be occupied by the proximal SUMO2/3, a non-SUMO substrate protein, or by (acetyl-)CoA, as shown for ChlaDUB1[44]. The S1′ site is not occupied in the SnCE1-SUMO1-PA structure.

Comparison of the crystal structure of SnCE1-SUMO1-PA with ChlaDUB1-Ub-PA reveals molecular determinants creating substrate preference of SnCE1 for SUMO chains (Fig. 2d, e). SnCE1 uses all variable regions, i.e., VR-1 located at the N-terminus, VR-2 preceding the first β-strand, VR-3 and its far C-terminal residues including the α-carboxylate for SUMO1 binding at S1 (Fig. 2c, d, and Supplementary Fig. 11). In SnCE1 VR-3 a short Gly-rich β-turn is connecting the first two β-strands of the central β-sheet (Fig. 1e, f). A negatively-charged surface in SUMO1, in the area of Leu65$^{\#2}$ (superscript #2: SUMO1) corresponding to the Ile44 patch in Ub, is pointing towards a positively-charged surface area in SnCE1, suggesting that electrostatics play a role for SUMO steering towards S1 in SnCE1 (Fig. 2c). Ub binding in ChlaDUB1 is mediated predominantly by hydrophobic interactions with residues in the α-helix in the extended VR-3 (Fig. 2c, d)[44]. Ub binds ChlaDUB1 via the Ile44 patch and is pointing with its Ile36 patch towards the α-helix in the extended VR-3, which is missing in SnCE1 (Figs. 1d, e and 2c–e). Leu65$^{\#2}$ in SUMO1 points towards the hydrophobic patch formed by Tyr91$^{\#2}$ of SUMO1, Leu308$^{\#1}$ (superscript #1: SnCE1) in the SnCE1 C-terminus and Ala168$^{\#1}$ in the VR-2 of SnCE1 (Fig. 2c). The orientation of SUMO1 binding to SnCE1 is similar to that observed in ChlaDUB1-Ub-PA. However, compared to ChlaDUB1 Ub binding, in SnCE1 specificity towards SUMO1 is created by formation of several side-chain interactions of SUMO1 with SnCE1 such as side chain of Gln94$^{\#2}$ with the hydroxyl group of side chain Thr146$^{\#1}$ in VR-2, Gln29$^{\#2}$ with main chain of Thr146$^{\#1}$, side chain of Tyr91$^{\#2}$ with main chain of Leu308$^{\#1}$ in the SnCE1 C-terminal region and side chain of Glu89$^{\#2}$ with the imidazole side chain of His163$^{\#1}$ (Fig. 2c, and Supplementary Fig. 11). Further specific interactions explain the observed higher specificity and efficiency of SnCE1 as deSUMOylase compared to its DUB activity (Fig. 1, and Supplementary Fig. 11). Notably, the side chains Arg70$^{\#2}$, Glu89$^{\#2}$ and Tyr91$^{\#2}$ in SUMO1 are replaced by Phe, Asp and Pro in SUMO2,3,4, respectively, providing an additional explanation for the observed higher reactivity of SnCE1 towards the SUMO1-PA probe compared to SUMO2/3-PA (Supplementary Fig. 8)[26]. We next analyzed whether SnCE1 has an acetyltransferase (AcT) activity

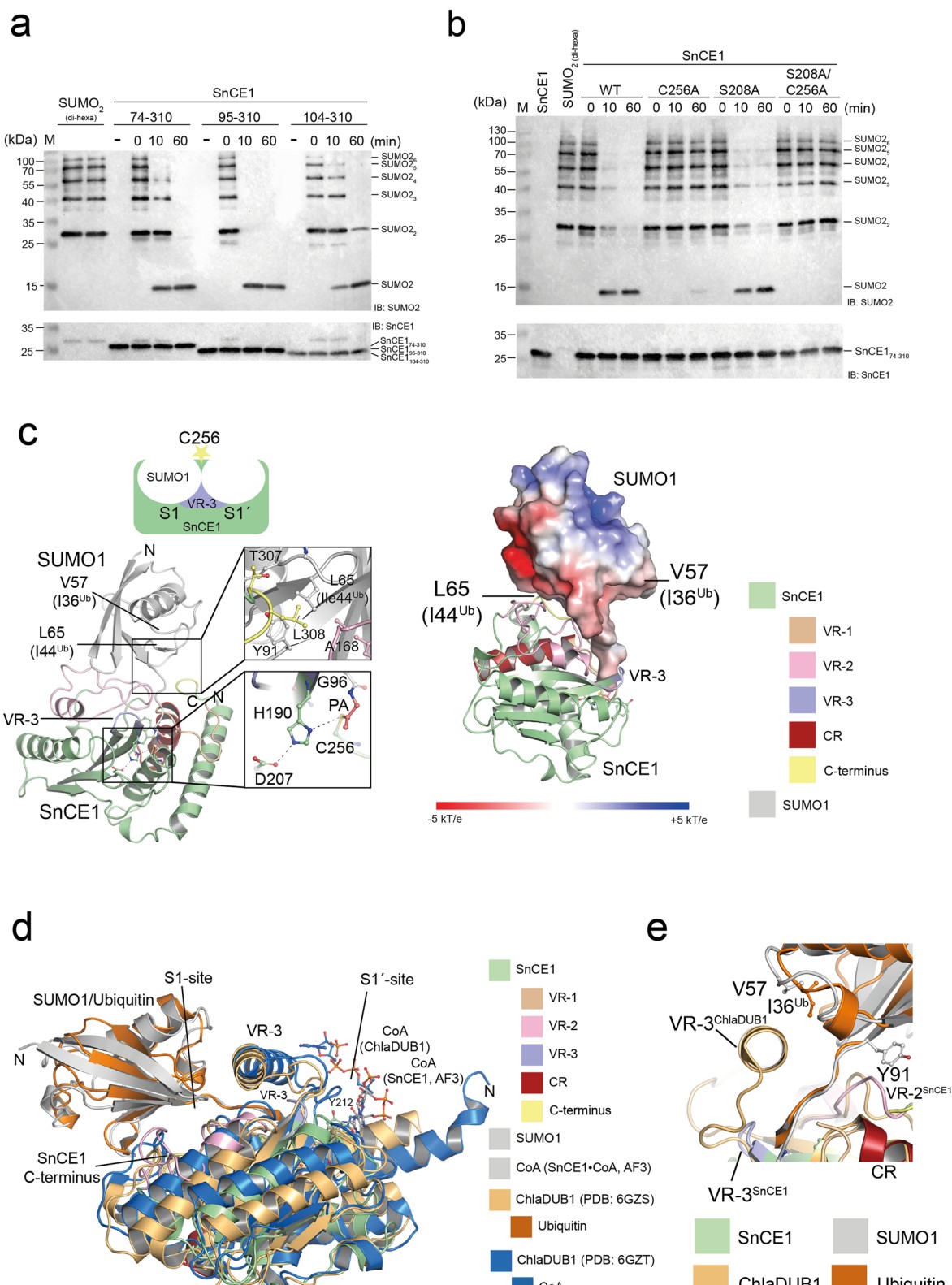

although the α-helix of the extended VR-3 in ChlaDUB1 is missing in SnCE1.

## SnCE1 has an autoacetyltransferase activity affecting its oligomeric state

As the related enzyme ChlaDUB1 from *C. trachomatis* was shown to have an intrinsic auto-AcT and being capable to acetylate other bacterial proteins in trans, we next analyzed whether SnCE1-5 are also lysine acetylated (Fig. 3a, and Supplementary Figs. 12 and 13). Immunoblotting using an anti-acetyl-lysine antibody (anti-AcK AB) revealed that only SnCE1 from the five CE-clan protease-related proteins shows a strong acetylation signal (Fig. 3a, and Supplementary Fig. 13). The acetylation signal was increased upon addition of acetyl-CoA, suggesting SnCE1 uses acetyl-CoA as a donor molecule for an

**Fig. 2 | Structure-function analyses of SnCE1 deSUMOylase activity. a** The SnCE1 N-terminal region does not contribute to its deSUMOylase activity. SnCE1$_{74-310}$, SnCE1$_{95-310}$ and SnCE1$_{104-310}$ degrade polymeric SUMO2$_{[2-6]}$ chains with similar efficiency to mono-SUMO2. SUMO2 chains were stained with anti-SUMO2 antibody (IB: SUMO2), loading control was done using an anti-SnCE1 antibody (IB: SnCE1). The result was obtained in two independent experiments ($n = 2$), one example is shown. Source data are provided as Source Data file. **b** The SnCE1 deSUMOylase activity is abolished by mutation of the catalytic Cys256. The SnCE1 mutants SnCE1 C256A and SnCE1 S208A/C256A are inactive; the single mutant SnCE1 S208A is active. SUMO2 chains were stained with anti-SUMO2 antibody (IB: SUMO2), loading control was done using anti-SnCE1 antibody (IB: SnCE1). The experiment was done in two independent technical replicates ($n = 2$), one example is shown. Source data are provided as Source Data file. **c** Crystal structure of SnCE1-SUMO1-PA. The crystal structure shows SUMO1 occupying the S1 site (distal SUMO1) in SnCE1. The covalent vinyl thioether formed by reaction of the catalytic Cys256 nucleophile is visible (lower closeup). The upper closeup shows selected interactions of residues in VR-1, VR-2 and VR-3 and the C-terminus of SnCE1 with SUMO1. The electrostatics was calculated with the APBS Electrostatics Plugin in PyMOL[134,135]. The figure was created with PyMOL[134]. **d** Structural prerequisites in SnCE1 underlying its deSUMOylase activity. SnCE1 does not contain an extended, α-helix-containing variable region-3 (VR-3) as ChlaDUB1 as shown by superposition of the SnCE1-SUMO1-PA structure solved here (PDB: 9QTG) with ChlaDUB1-Ub-PA (PDB: 6GZS) and ChlaDUB1•CoA (PDB: 6GZT). CoA from an AlphaFold3 predicted structure in complex with SnCE1 is shown in grey. The variable regions 1-3 (VR-1-3) responsible to determine substrate specificity towards SUMO are colour coded (VR-1: orange; VR-2: pink; VR-3: blue; C-terminus: yellow; constant region (CR): dark red). The figure was created with PyMOL[134]. **e** Closeup of the superposition of SnCE1-SUMO1-PA (PDB: 9QTG) and ChlaDUB1-Ub-PA (PDB: 6GZS) in the area of VR-3 (blue). SUMO1 is pointing with its Val57 patch of SUMO1 (Ile36 patch in ubiquitin) towards VR-3$^{SnCE1}$ (blue)/VR-3$^{ChlaDUB1}$ (dark orange). The figure was created with PyMOL[134].

enzymatically catalysed autoacetylation. This was not observed for SnCE2-5 indicating SnCE1 is the only *S. negevensis* CE-clan protease-related DUB/ULP active as AcT (Supplementary Fig. 13). The catalytically inactive mutant SnCE1 C256A showed no acetylation even upon incubation with acetyl-CoA (Fig. 3a, and Supplementary Fig. 13). Acetylated SnCE1 wild type was not deacetylated by the *E. coli* sirtuin deacetylase EcCobB (Fig. 3a, and Supplementary Fig. 13). These results suggest SnCE1 uses Cys256 as nucleophile for the hydrolysis reaction and the condensation reaction, i.e., the deSUMOylase activity and the AcT activity. Notably, analytical SEC experiments reveal that catalytically dead, non-acetylated SnCE1 forms a tetramer in solution existing in an equilibrium with a monomer while acetylated SnCE1 wild type exclusively forms a monomer (Fig. 3b, Supplementary Figs. 1 and 2b and 14). This suggests the oligomeric state of SnCE1 is modulated by lysine acetylation.

To investigate whether acetylation of SnCE1 affects its structure, we performed structural analyses of the catalytically dead, non-acetylated SnCE1$_{74-310}$ C256A mutant. We obtained crystals belonging to the monoclinic space group P12$_1$1 that diffracted up to 2.21 Å containing eight SnCE1 molecules per asymmetric unit organized as two pairs of tetramers with C$_4$ symmetry (PDB: 9QTF; Fig. 3c). In the tetramer, the interface area between two individual monomers is on average 724 Å$^2$ as determined by the PISA server (http://www.ebi.ac.uk/pdbe/prot_int/pistart.html)[63]. This furthermore indicates the tetramer being of moderate stability, confirming the analytical SEC experiments (Fig. 3b, and Supplementary Fig. 1). Moreover, as VR-3 is directly involved in providing the interface for the subunits forming the tetramer, a similar tetramerization would not be possible in ChlaDUB1[44]. Most of the inter-subunit interactions in this region are main-chain interactions such as between main-chains of Gly188 of VR-3 of one chain and Gly235 in the loop connecting α3 and β4 of the other neighboring chain. Moreover, tetramer formation is mediated by an extensive water network in the interface of the chains established, amongst others, by the residues Tyr212, Asn189 and His190 (Fig. 3d). One prominent salt bridge is formed between residues in α3, i.e., Glu217 and Lys231 located in opposite sites in α3 (Fig. 3d). Notably, AlphaFold3 does not predict the same SnCE1 tetramer and the pTM +ipTM values of 0.38 + 0.48 suggest this prediction to be of low quality (Fig. 3c, and Supplementary Fig. 15)[59-61].

Binding of a distal SUMO at S1 and either a proximal SUMO, a non-SUMO substrate or acetyl-CoA/CoA at S1' would be possible to occur simultaneously on a SnCE1 monomer (Fig. 4b, c). However, when superpositioning the SnCE1 C256A tetramer structure with the structure of SnCE1-SUMO1-PA it becomes obvious that SnCE1 tetramer formation is incompatible with SUMO binding at S1 (Fig. 4c). These structural data suggest the tetrameric form of SnCE1 might be impaired regarding the deSUMOylase activity. Moreover, His190 the base of the catalytic triad is located in the interface of SnCE1 subunits within the SnCE1 tetramer, implying tetramer formation has

consequences on catalysis. (Fig. 4c). The mutant SnCE1 H190A shows an almost non-detectable autoacetylation, suggesting a catalytic role also for the AcT activity, as reported earlier (Figs. 3d and 4e)[41]. Overall, these data suggest oligomerization of SnCE1 is regulated by auto-acetylation and that, structurally, this oligomerization interferes with SnCE1 SUMO binding at S1 with potential consequences on SnCE1 deSUMOylase activity. To investigate the mechanism underlying the AcT activity, we conducted mutational studies complemented by AlphaFold3 structure predictions.

## Aromatic side chain Tyr212 is essential for SnCE1 AcT activity

AlphaFold3 structure prediction of the complex structure of mono-meric SnCE1 with coenzyme A (CoA) (pTM+ipTM: 0.78 + 0.80) sug-gests the validity of the model and binding of CoA to S1' in SnCE1 being possible without the presence of an extended VR-3 in SnCE1 (Fig. 4a)[59-61]. According to the AlphaFold3 model the CoA occupies the S1' site with the CoA's sulfhydryl group capable to form a disulfide with the active site Cys256 (Fig. 4d, and Supplementary Fig. 16). Tyr212 forms a π-stacking interaction with the adenine base of CoA (Fig. 4d). Additionally, the main-chain of Asn189 forms a hydrogen bond to the main-chain amide connecting the cysteamine with the β-alanine of CoA and side-chain of Lys209 forms a hydrogen bond with the amide connecting β-alanine to pantoic acid of CoA (Fig. 4d). To validate the AlphaFold3 model, we mutated Tyr212 to Ala and analyzed the acet-ylation state by immunoblotting (Fig. 4e). As expected, SnCE1 Y212A was not acetylated supporting Tyr212 being essential for (acetyl-)CoA binding and/or orienting it for catalysis (Fig. 4e). Moreover, analytical SEC suggests that SnCE1 Y212A is properly-folded, however, in contrast to non-acetylated SnCE1 C256A it elutes as sole monomer indicating the mutation affects the oligomeric state (Supplementary Fig. 1). Tyr212 is located on a loop preceding α3 containing Glu217 and Lys231 forming the salt bridge in the tetramer (Fig. 1e, f). To this end, the mutation Y212A in SnCE1 might affect the precise orientation and position of α3, thereby reducing the propensity of tetramer formation. Moreover, Tyr212 is indirectly involved in tetramer formation by mediating the formation of a water network in the interface of the subunits (Fig. 3c, d). Interestingly, acetylation signals for the mutants SnCE1 K231R, SnCE1 E217D K231R and of the gatekeeper SnCE1 W191A were strongly reduced suggesting a potential structural role interfer-ing with (acetyl-)CoA binding (Fig. 4e). Notably, non-acetylated, monomeric SnCE1 Y212A is active as deSUMOylase with an effi-ciency only slightly reduced compared to acetylated, monomeric wild type SnCE1, supporting the observation that the monomer formation in SnCE1 is important for its deSUMOylase activity (Fig. 5a). These data show the mutation Y212A in SnCE1 allows to discriminate between AcT activity and deSUMOylase activity.

AlphaFold3 structure predictions of apo SnCE1-5 and of CoA-bound SnCE1-5 and superposition of these structural models sug-gest that all proteins share a CE-clan protease fold with varying

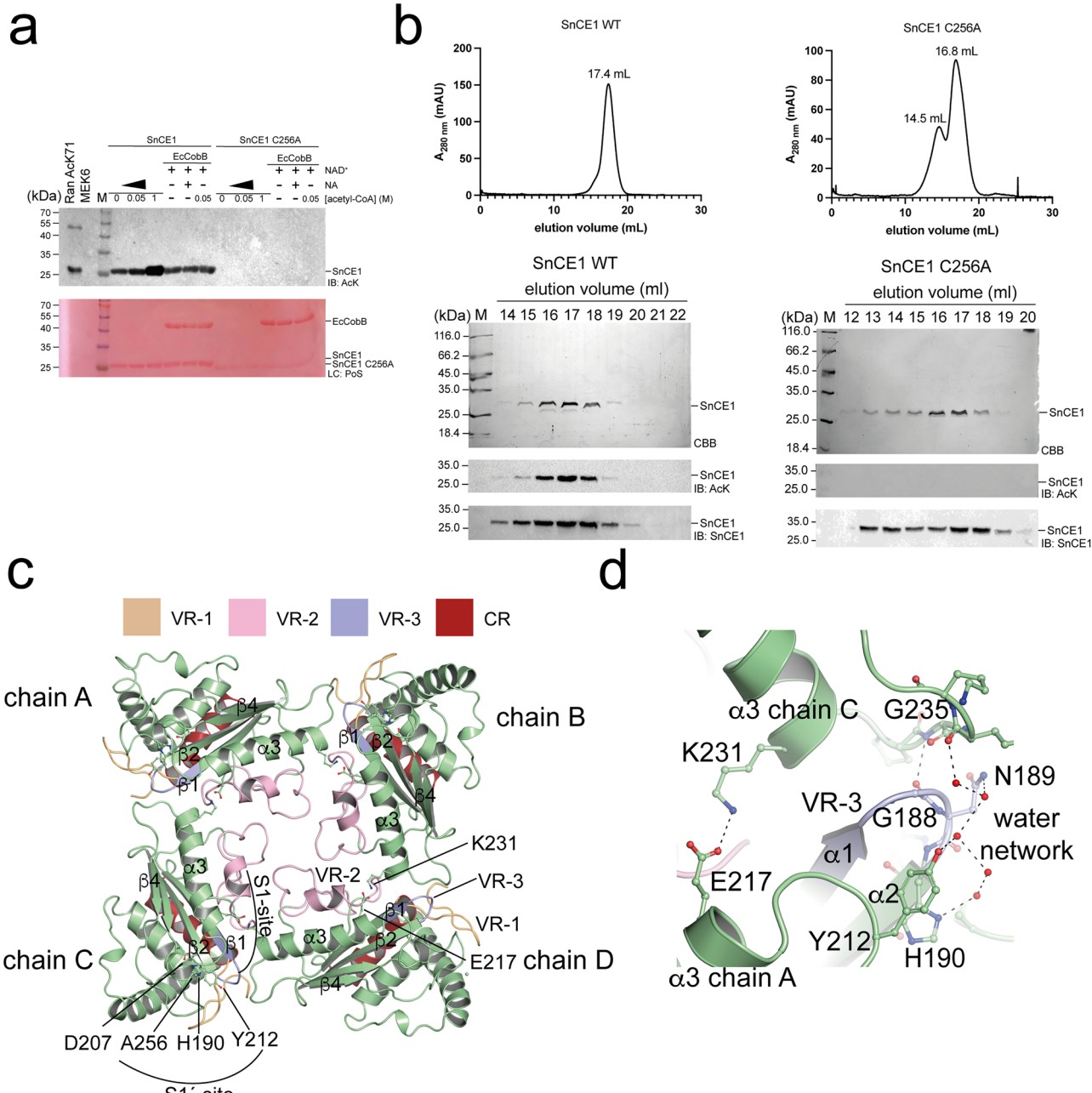

**Fig. 3 | SnCE1 has an intrinsic autoacetyltransferase activity. a** SnCE1 catalyses its own acetylation in an enzymatic autoacetyltransferase activity. The immunoblotting (IB) using an anti-AcK AB (IB: AcK) shows the signal being increased upon addition of acetyl-CoA and the mutant SnCE1 C256A is not acetylated suggesting the SnCE1 AcT activity to be enzymatically catalysed. The *E. coli* sirtuin CobB, i.e., EcCobB, is not able to deacetylate SnCE1 in presence of the co-substrate NAD⁺. NA: nicotinamide. Ponceau S red staining was done as loading control (LC: PoS). The experiment was performed in two independent technical replicates ($n = 2$), one example is shown. Source data are provided as Source Data file. **b** Wild type SnCE1 is acetylated and monomeric while SnCE1 C256A is not acetylated and elutes in a tetramer-monomer equilibrium from the analytical SEC column. The analytical SEC was conducted on a calibrated S200 10/300 column (Supplementary Fig. 2). The immunoblot was done with anti-acetyl-lysine antibody (IB: AcK). Loading control of the immunoblots were done by immunoblot using an anti-SnCE1 antibody (IB:

SnCE1). The experiment was performed in two independent technical replicates ($n = 2$), one example is shown. Source data are provided as Source Data file. **c** Crystal structure of catalytically inactive SnCE1 C256A confirms tetramer formation. The tetramer is formed by interactions of residues of the loop connecting α3 and β4 of one monomer and the Gly-rich VR-3 of the neighboring monomer. From the structure it is obvious that in contrast to the S1′ site the S1 site is not accessible in the tetramer. The figure was created with PyMOL[134]. **d** Closeup of the interface region of SnCE1 monomers within the tetramer. Besides from several hydrophobic interactions and main chain interactions such as between Gly235 of one chain and Gly188 of the neighboring chain, one prominent interchain salt bridge is formed between Lys231 of one monomer and Glu217, both located on α3, of the neighboring monomer. Moreover, interaction is facilitated by an extended water network established with residues Tyr212 and His190. The figure was created with PyMOL[134].

degree of similarity towards SnCE1 (Supplementary Fig. 16). Moreover, key residues needed for CoA binding in SnCE1, including Tyr212, are missing in SnCE2-5 providing a potential explanation for the lack of AcT activity for SnCE2-5 (Supplementary Fig. 16). To

understand which sites are acetylated in SnCE1 and whether the acetylation state of SnCE1 is enzymatically regulated by host cell deacetylases we next performed enzyme assays and mass spectrometry.

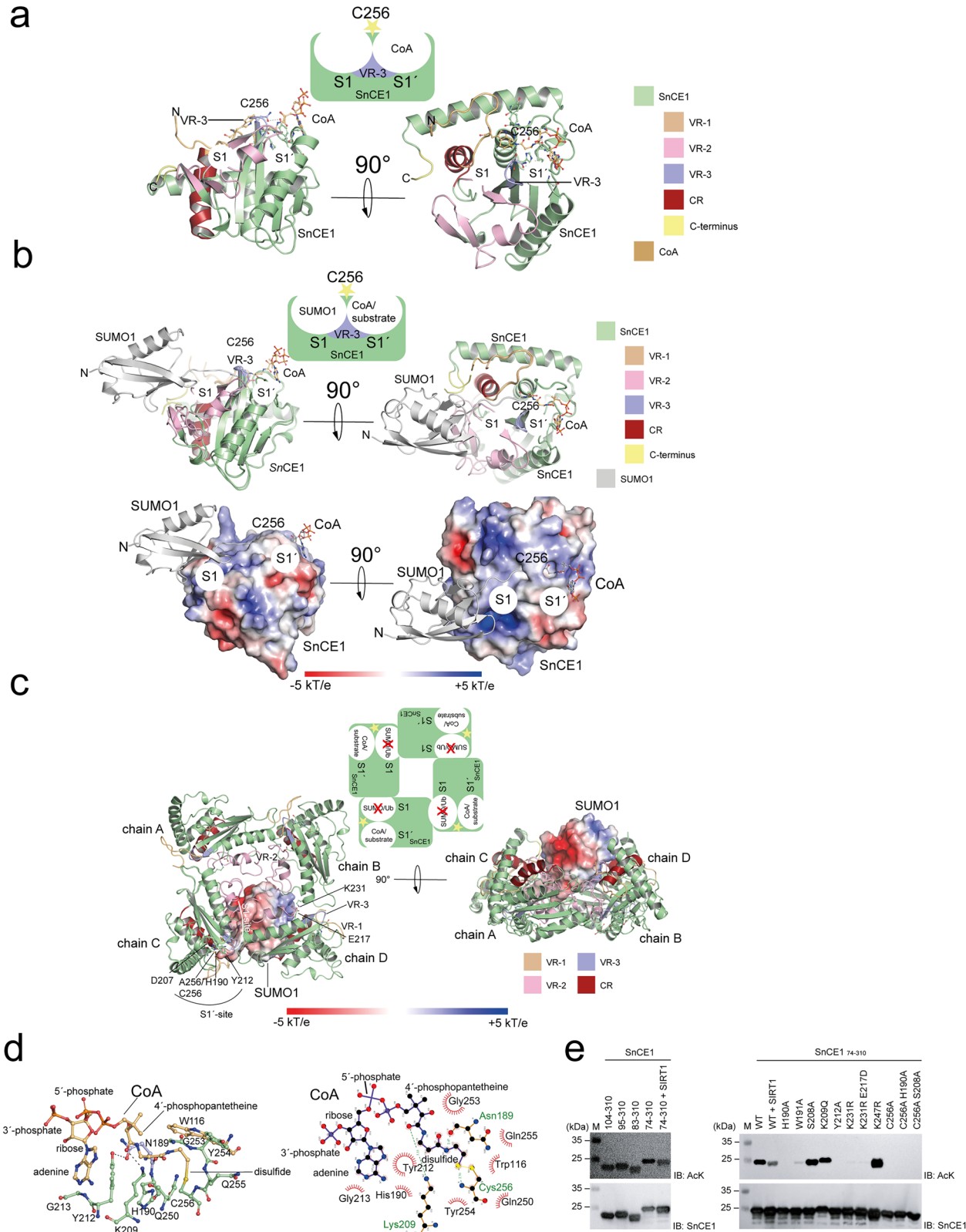

## SnCE1 autoacetylation occurs on multiple sites and is regulated enzymatically by sirtuins

While we discovered SnCE1 acetylation not being reversed by the NAD$^+$-dependent *E. coli* deacetylase CobB, i.e., EcCobB, we next analyzed whether SnCE1 is deacetylated by known and robust human deacetylases, i.e., the sirtuin deacetylases SIRT1, SIRT2 and SIRT3 (Figs. 3a and 5b, c). The analyses revealed that the lysine acetylation of

SnCE1 can be reversed by all three human sirtuins with similar efficiency in an NAD$^+$-dependent manner. However, no complete deacetylation was achieved as shown by immunoblotting using an anti-acetyl-lysine antibody (Fig. 5b, c). This suggests lysine acetylation being irreversible at one or several positions or other deacetylases being able to completely deacetylate SnCE1. Notably, SIRT1-deacetylated SnCE1 showed a similar deSUMOylase activity

**Fig. 4 | SnCE1 tetramer formation is incompatible with SUMO-binding at S1 while CoA binding at S1′ is possible. a** AlphaFold3 structure prediction suggests binding of acetyl-CoA/CoA at S1′ without presence of an extended VR-3. In the predicted AlphaFold3 structure the sulfhydryl group of CoA forms a covalent disulfide bond with the catalytic Cys256. The figure was created with PyMOL[134]. **b** Simultaneous binding of a distal SUMO1 and proximal SUMO/CoA/non-SUMO substrate to S1 and S1′, respectively, is possible in the SnCE1 monomer. The interactions involve VR-3. The CoA was obtained from superposition of this AlphaFold3 model to the SnCE1-SUMO1-PA structure. The distal SUMO1 binds to a positively-charged surface at S1. The electrostatics was calculated with the APBS Electrostatics Plugin in PyMOL[134,135]. **c** SnCE1 tetramer formation and SUMO binding at the distal site S1 are mutually exclusive. Superposition of the structures of the tetrameric SnCE1 C256A and the monomeric SnCE1-SUMO1-PA is shown. While (acetyl-)CoA-binding at the proximal site S1′ is compatible with tetramerformation of SnCE1, SUMO binding at S1 is not due to steric clashes. The figure was created with

PyMOL[134]. **d** CoA binding to SnCE1 is mediated by several non-covalent interactions as shown by AlphaFold3 prediction. The model has pTM+ipTM value of 0.78 + 0.80. The adenine base of CoA is bound by a π-stacking interaction with Tyr212 in SnCE1, the main-chain of Asn189 forms a hydrogen bond to the amide bond connecting the cysteamine with the β-alanine of CoA and side-chain of Lys209 forms a hydrogen bond with the amide bond connecting β-alanine to pantoic acid of CoA. The figure in the upper panel was created with PyMOL[134]. The left panel shows interactions of residues in SnCE1 and CoA. This figure was created by LigPlot⁺ v.2.2[136,137]. **e** Acetylation status of SnCE1 proteins. Immunoblotting by an anti-acetyl-lysine antibody (IB: AcK) shows N-terminally truncated SnCE1 proteins are acetylated. The mutants SnCE1 C256A, SnCE1 Y121A and SnCE1 H190A are not acetylated. Wild type SnCE1 is acetylated and can be deacetylated with SIRT1 (SnCE1deac.). Loading control was done by immunoblotting using an anti-SnCE1 antibody (IB: SnCE1). The experiment was performed once ($n = 1$). Source data are provided as Source Data file.

---

compared to acetylated wild type SnCE1 (Fig. 5a). Moreover, we observed a similar reactivity of deacetylated SnCE1 and wild type SnCE1 with the SUMO1-PA/SUMO3-PA and Ub-PA probes (Fig. 5d).

To identify the acetylation sites in SnCE1 and to narrow down the sites that are enzymatically regulated we performed mass spectrometry on acetylated wild type SnCE1, SIRT1-treated wild type SnCE1 and analyzed the non-acetylated mutant SnCE1 Y212A as a control (Supplementary Fig. 17). This study revealed some lysine acetylation (AcK) sites being accessible to SIRT1-catalysed deacetylation, i.e., AcK94, AcK98, AcK103, AcK237 and AcK289, and some sites identified by mass spectrometry (MS) resist deacetylation by SIRT1, i.e., AcK157, AcK172, AcK223 and AcK231, not excluding the possibility other deacetylases being capable to catalyse their deacetylation (Supplementary Fig. 17). This shows lysine acetylation sites distributed over almost the entire SnCE1 protein, i.e., AcK94, AcK98 and AcK103 in the N-terminal region and AcK157, AcK172, AcK223, AcK231, AcK237 and AcK289 in the central and C-terminal part of the SnCE1 protein (Fig. 5e). Notably, SnCE1 might be acetylated at further sites not being identified by mass spectrometry. Inspection of the acetylation sites in the SnCE1 structure shows lysine acetylation sites AcK94, AcK98, AcK103 and AcK237 being accessible to SIRT1-catalysed deacetylation and are located in flexible, unstructured regions such as the N-terminal region preceding the CE-clan protease domain. Other non-regulated sites are present in regions with profound secondary structure, i.e., AcK223 and AcK231 in α-helices (Fig. 5e). However, AcK289 located in an α-helix is accessible for SIRT1-catalysed deacetylation whereas AcK157 present in a loop as well as AcK172 located in a β-hairpin according to the AlphaFold3 model, but present in flexible VR-2 in the experimental structure, are both not deacetylated[59–61]. These data suggest that structure and sequence determine which lysine acetylation sites are accessible for enzymatic deacetylation, supporting earlier findings of our laboratory[64].

We did not observe a deacetylation of SnCE1 by the only deacetylase CobB from *S. negevensis*, i.e., SnCobB (Fig. 5f, and Supplementary Figs. 18 and 19). SnCobB is a NAD⁺-dependent sirtuin deacetylase with sequence and structural similarity to human and bacterial sirtuins (Supplementary Figs. 18 and 19). As direct consequence, some SnCE1 acetylation sites are enzymatically deacetylated upon exposure in the host cell following injection of SnCE1 while others might remain stable as they are not accessible for enzymatic deacetylation. Notably, we did not detect any acetylated Ser or Thr residues in SnCE1, suggesting that in contrast to *Yersinia* ssp. YopJ, SnCE1 is only able to acetylate lysine side-chains. To narrow down which acetylation sites modulate the SnCE1 oligomeric state, we next applied the genetic code expansion concept (GCEC) to produce site-specifically lysine acetylated SnCE1 proteins.

### Acetylation of SnCE1 at K231 is irreversible and mediates SnCE1 monomerization

We site-specifically incorporated acetyl-lysine (AcK) in SnCE1 C256A at different positions, i.e., AcK78, AcK94, AcK98, AcK103, AcK106,

AcK209, AcK231 and AcK248 for further analyses (Supplementary Figs. 20 and 21). These sites were selected as they were either identified by LC-MS/MS analyses or are present in functionally or structurally important regions in the SnCE1 protein but might have resisted identification by mass spectrometry. The studied acetylation sites include sites in the N-terminus of SnCE1 (AcK94, AcK98, AcK103, AcK106), sites that might affect SnCE1 oligomerization (AcK231) and catalytically relevant (AcK209) sites pointing towards the active site or sites being located in a reactive poly-basic patch (AcK248), i.e., 246-KKK-248, known to increase the nucleophilicity of lysine side-chains (Figs. 1f and 5e, and Supplementary Fig. 12). The site-specific incorporation of acetyl-lysine was achieved by applying the GCEC using a synthetically evolved acetyl-lysyl-tRNA-synthetase/tRNA$_{CUA}$-pair from *Methanosarcina barkeri* as described earlier (Supplementary Fig. 20)[65–68]. Notably, we produced acetylated SnCE1 protein on the background of the catalytically inactive mutant SnCE1 C256A in order to be able to assess a possible impact of acetylation on the SnCE1 oligomeric state and to obtain SnCE1 protein acetylated specifically on these individual lysine side-chains as wild type SnCE1 is acetylated at multiple sites and forming a monomer in solution (Supplementary Figs. 1, 2c, 20 and 21).

As observed earlier, the individual acetylation sites in SnCE1 are detected with different efficiency by the anti-acetyl lysine antibody during immunoblotting (Supplementary Fig. 21b)[69,70]. We confirmed the incorporation additionally by intact mass determination (Supplementary Data 3 and 4). To confirm the obtained MS data and to investigate which of these sites are regulated by *S. negevensis* SnCobB we next investigated whether site-specifically acetylated SnCE1 proteins are deacetylated by human SIRT1, SIRT2 and by *S. negevensis* SnCobB (Supplementary Fig. 21c). These analyses confirm all acetylation sites present in flexible loop regions, i.e., Lys78, Lys94, Lys98, Lys103 and Lys106, are deacetylated in an NAD⁺-dependent manner by the human sirtuins, while the sites not solvent-exposed orienting to the core of SnCE1 or present in areas of profound secondary structure, i.e., Lys209, Lys231 and Lys248, are not deacetylated (Figs. 5e and 6a, and Supplementary Fig. 21c). Interestingly, neither the site-specifically acetylated SnCE1 variants nor wild type SnCE1 were deacetylated by SnCobB suggesting SnCE1 is deacetylated at the regulated sites after being injected into the host cell by host cell deacetylases (Fig. 5f, and Supplementary Fig. 21c).

We analyzed all acetylated proteins by analytical SEC regarding their oligomeric states. Except from acetylated wild type SnCE1 and SnCE1 C256A AcK231 forming exclusively monomers in solution all other acetylated SnCE1 C256A variants are capable to form tetramers in equilibrium with monomers in solution (Figs. 3b and 6b, and Supplementary Figs. 1, 2c and 20). These data confirm our structural assumptions that Lys231 in SnCE1 mediates an important interchain interaction by forming a salt bridge with Glu217 of a neighboring chain, stabilizing the SnCE1 tetramer (Fig. 3c, d). Acetylation at Lys231

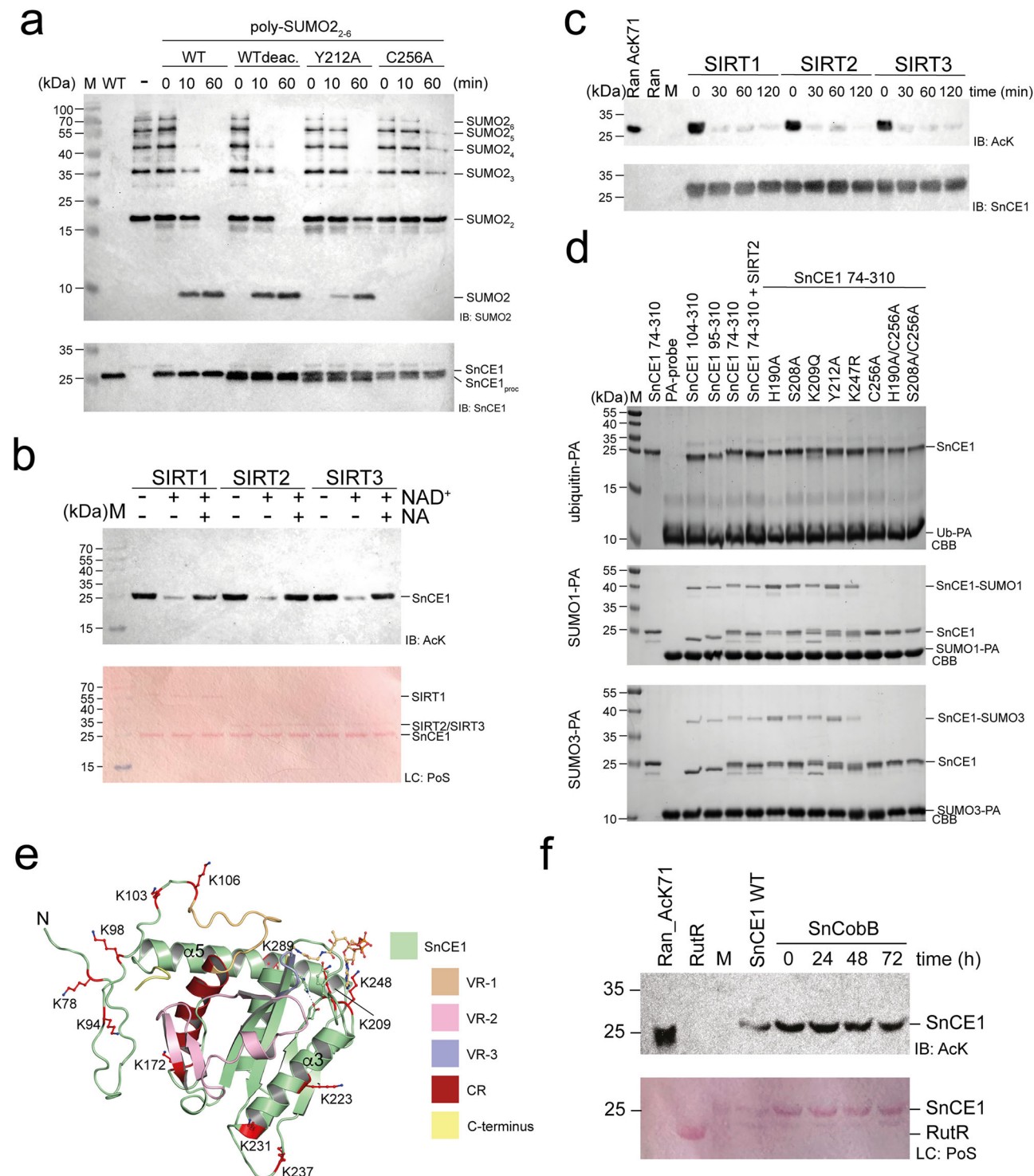

electrostatically and sterically abolishes formation of this salt bridge by neutralizing the positive charge at the lysine side chain and by exchanging a lysine with a sterically more demanding acetyl-lysine. As the data show SnCE1 C256A AcK231 neither being deacetylated by *S. negevensis* SnCobB nor by human SIRT1/SIRT2 under assay conditions, we next studied whether other deacetylases are capable to deacetylate SnCE1 AcK231 (Fig. 6a, c). This showed that none of the sirtuin deacetylases, i.e., human SIRT1-7, nor any of the tested classical $Zn^{2+}$-dependent deacetylases, i.e., human HDAC1,2,3,4,6,7,8,9,11, were capable to deacetylate SnCE1 C256A at AcK231 (Fig. 6c). This suggests acetylation at Lys231 being irreversible to stabilize the SnCE1 monomer.

As further support, we analyzed SIRT1-treated SnCE1 C256A AcK231 and SnCE1 Y212A AcK231 by analytical SEC confirming this protein eluting as monomer while non-acetylated SnCE1 C256A eluting as tetramer (Fig. 3b, and Supplementary Fig. 22a). As the SnCE1 tetramer formation is structurally incompatible with SUMO1 binding at S1, we hypothesize Lys231-acetylation of SnCE1 promoting its deSUMOylase activity by stabilization of the SnCE1 monomeric state (Fig. 4c). To test this hypothesis, we aimed at constructing a catalytically active SnCE1 protein capable to form a tetramer. To this end, we designed two SnCE1 mutants, i.e., SnCE1 K231R and SnCE1 E217D K231R, which cannot be acetylated on Lys231 and should be capable to form a salt bridge shown to be essential for

**Fig. 5 | SnCE1 is acetylated at multiple sites regulated by human sirtuin deacetylases. a** Deacetylated SnCE1 is active as deSUMOylase. SnCE1 is deacetylated with SIRT1 (WTdeac.) and the non-acetylated mutants SnCE1 Y212A are active as deSUMOlyases cleaving polymeric SUMO2$_{[2-6]}$ with similar efficiency. SnCE1 C256A is inactive. The data indicate proteolytic processing of WTdeac. and the non-acetylated mutants SnCE1 Y212A/SnCE1 C256A as we observe processed SnCE1 (SnCE1$_{proc}$) next to SnCE1$_{74-310}$ (SnCE1). SUMO2 chains were stained with anti-SUMO2 antibody (IB: SUMO2), loading control was done using an anti-SnCE1 antibody (IB: SnCE1). The experiment was done in two independent technical replicates ($n = 2$), one example is shown. Source data are provided as Source Data file. **b** Wild type SnCE1 is deacetylated by sirtuins SIRT1-3. Deacetylation is NAD$^+$-dependent and inhibited by nicotinamide (NA) as shown by immunoblotting using an anti-acetyl-lysine antibody (IB: AcK). SnCE1 cannot be completely deacetylated by SIRT1-3. Immunoblotting with anti-SnCE1 antibody was done as loading control (IB: SnCE1). The experiment was performed in three independent technical replicates ($n = 3$). Source data are provided as Source Data file. **c** Wild type SnCE1 is efficiently deacetylated by sirtuins SIRT1-3. After 30 min SnCE1 is deacetylated to its final state, while not completely deacetylated by SIRT1, SIRT2, SIRT3. Immunoblotting with anti-SnCE1 antibody was done as loading control (IB: SnCE1). The experiment was performed in three independent technical replicates ($n = 3$). Source data are provided as Source Data file. **d** SIRT2-deacetylated SnCE1 reacts with the activity-based SUMO1-PA and SUMO3-PA probes similar as acetylated wild type SnCE1. No reaction occurs with ubiquitin-PA (Ub-PA). The truncated SnCE1 proteins behave similar to SnCE1$_{74-310}$. Mutants containing C256A are not reacting. The SDS-PAGE gels were stained with Coomassie brilliant blue (CBB). The experiment was performed in two independent technical replicates ($n = 2$). Source data are provided as Source Data file. **e** Localization of identified lysine acetylation sites in the SnCE1 structure. The acetylation sites are broadly distributed on SnCE1. The figure was created with PyMOL[134]. **f** *S. negevensis* CobB, i.e., SnCobB, does not deacetylate SnCE1. SnCE1 is not deacetylated even after incubation for 72 h. The experiment was performed in two independent technical replicates ($n = 2$). Source data are provided as Source Data file.

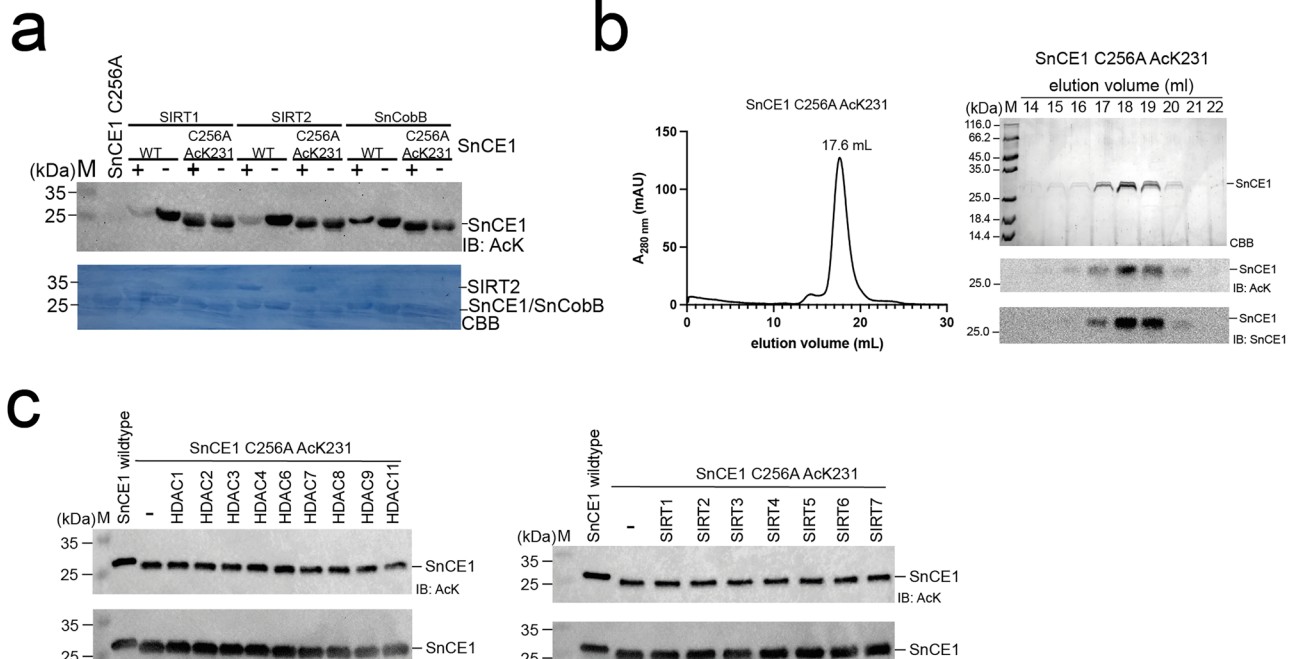

**Fig. 6 | Acetylation of SnCE1 at Lys231 stabilizes the SnCE1 monomer. a** SnCE1 C256A AcK231 is neither deacetylated by human SIRT1 and SIRT2 nor by *S. negevensis* CobB, i.e., SnCobB. Deacetylation of acetylated wild type SnCE1 and site-specifically acetylated SnCE1 C256A AcK231 was assessed by immunoblotting with anti-acetyl lysine antibody (IB: AcK) and Coomassie Brilliant Blue staining was done as loading control (CBB). The experiment was performed in two independent technical replicates ($n = 2$). Source data are provided as Source Data file. **b** SnCE1 C256A AcK231 elutes as monomer from analytic SEC column. Wild type SnCE1 elutes as monomer, SnCE1 C256A as tetramer. A$_{280}$ is the absorption at 280 nm. mAU: milli absorbance units. Fractions were analyzed by SDS-PAGE and gels were stained with Coomassie brilliant blue (CBB), immunoblotting with anti-acetyl-lysine antibody confirms the acetylation (IB: AcK) and staining with anti-SnCE1 antibody was done as loading control (IB: SnCE1). The experiment was performed once ($n = 1$). Source data are provided as Source Data file. **c** SnCE1 C256A AcK231 is neither deacetylated by human sirtuins (SIRT1-SIRT7) nor by selected human classical HDACs. The acetylation state of SnCE1 was assessed by immunoblotting using an anti-acetyl lysine antibody (IB: AcK). Immunoblotting with anti-SnCE1 antibody served as loading control (IB: SnCE1). The experiment was performed once ($n = 1$). Source data are provided as Source Data file.

tetramer formation. We observed that both mutants eluted as pure monomer from analytical SEC (Supplementary Fig. 23). To exclude the possibility that other lysine acetylation sites in SnCE1 K231R or SnCE1 E217D K231R might prohibit tetramer formation, we deacetylated SnCE1 K231R and SnCE1 E217D K231R by SIRT1 and performed analytical SEC. However, also these proteins only eluted as pure monomers (Supplementary Fig. 24). These data suggest that both mutants impair SnCE1 tetramer formation. As a consequence, we are not able to experimentally validate the impaired deSUMOylase activity due to formation of the SnCE1 tetramer. However, our structural data in combination with analytical SEC experiments both suggest that oligomerization of SnCE1 modulated by its acetylation state affects its deSUMOylase activity.

As a summary, our model suggests acetylation of SnCE1 at Lys231 is catalysed by irreversible autoacetylation and a mechanism to increase SnCE1 deSUMOylase activity by shifting the tetramer-monomer equilibrium towards the SnCE1 monomeric form. We observed that SnCE1 is prone to proteolytic processing that might be dependent on its acetylation state. To this end, we next analyzed whether SnCE1 proteolytic processing correlates with its lysine acetylation state, where the proteolytic processing occurs and whether this can also be attributed to individual lysine acetylation sites (Fig. 5a).

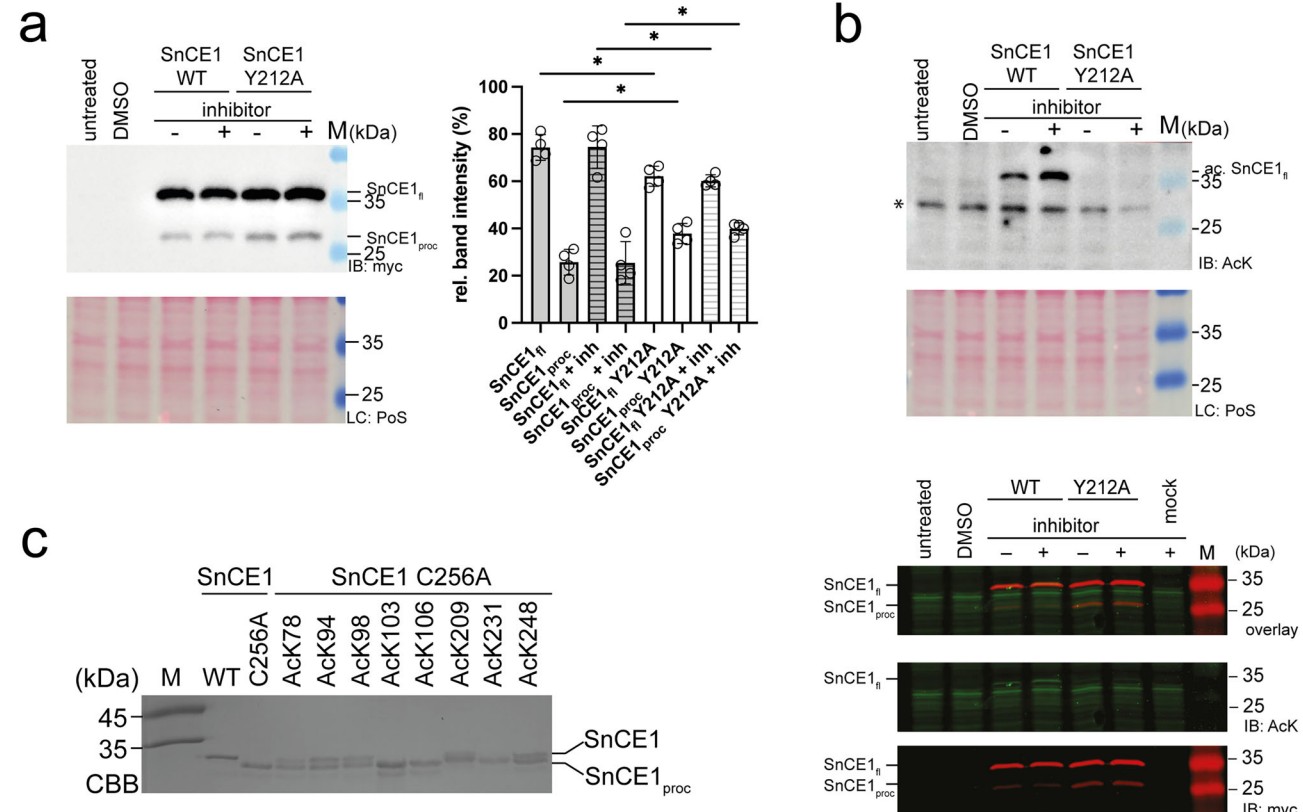

**Fig. 7 | Acetylation of SnCE1 modulates its proteolytical processing at the N-terminus. a** SnCE1 full length (SnCE1$_{fl}$) wild type and SnCE1$_{fl}$ Y212A are proteolytically processed at the N-terminus in eukaryotic HEK-293 cells. Immunoblotting using an anti-myc antibody shows SnCE1$_{fl}$ and the proteolytically processed form (SnCE1$_{proc}$) were stained suggesting processing occurs from the N-terminus (IB: myc). Treatment with lysine deacetylase inhibitors TSA, SAHA and nicotinamide (NA) were used to inhibit deacetylases. Ponceau S red staining was done as loading control (LC: PoS). Experiments were performed in four biological replicates and bars depict means ± standard deviations. Statistical significance (*$p \leq 0.05$) was tested using *t*-tests. Source data are provided as Source Data file. **b** SnCE1 is acetylated in human HEK-293 cells. Upper panel: Staining with an anti-acetyl-lysine antibody (anti-AcK AB) shows SnCE1 is acetylated in human cells (IB: AcK). The proteolytically processed SnCE1 is not acetylated suggesting acetylated lysine side chains to reside in the N-terminus. Treatment with lysine deacetylase inhibitors TSA, SAHA and nicotinamide (NA) were used to inhibit deacetylases. Ponceau S red staining was done as loading control (LC: PoS). Lower panel: The staining with

fluorescently labelled secondary antibodies shows the signals obtained with the anti-AcK AB and anti-myc antibody do merge for unprocessed full length SnCE1 (SnCE1$_{fl}$), while processed SnCE1 (SnCE1$_{proc}$) is not acetylated. Treatment with lysine deacetylase inhibitors shows enhanced acetylation for SnCE1 wild type (WT). Staining with anti-AcK AB shows similar loading. *: non-specific acetylated human protein. Experiments were performed in two biological replicates ($n = 2$). Source data are provided as Source Data file. **c** Autoproteolytic processing of site-specifically acetylated SnCE1 C256A proteins. Wild type SnCE1 is not processed, while SnCE1 C256A is quantitatively processed and occurs only in the processed form. Except from SnCE1 C256A AcK103, SnCE1 C256A AcK106 and SnCE1 C256A AcK231 showing full proteolytic processing resembling SnCE1 C256A, the other N-terminal sites as well as SnCE1 C256A AcK248 show impaired processing as full length and processed bands are visible. SnCE1 C256A AcK209 shows diminished processing resembling wild type SnCE1. The SDS-PAGE gel was stained with Coomassie brilliant blue (CBB). Source data are provided as Source Data file.

## Acetylation modulates N-terminal autoproteolytic processing of SnCE1

The data shown above indicate lysine acetylation in wild type SnCE1 prohibits proteolytic processing, while all non-acetylated mutants, including SnCE1 Y212A and SnCE1 C256A, show proteolytic processing as visible by immunoblotting with a specific anti-SnCE1 antibody (Figs. 5a and 7a). As clear evidence the processing of SnCE1 is mechanistically achieved by an autoproteolytic activity and not by an exogenous proteolytic activity, deacetylation of wild type SnCE1 by SIRT1 induces its proteolytic processing (Figs. 5a and 7a). As support for an autoproteolytic processing, we did not identify any exogenous protease in the samples as by LC-MS/MS (Supplementary Data 2). This furthermore suggests that proteolytic processing of SnCE1 being directly dependent on the SnCE1 acetylation state. Importantly, also SnCE1 C256A AcK231 eluting only as monomers from analytical SEC shows proteolytic processing (Figs. 6b and 7a, and Supplementary Fig. 2c). This confirms the tetramer formation of SnCE1 is not a prerequisite for proteolytic processing. As wild type SnCE1 is not deacetylated by the only *S. negevensis* deacetylase, i.e., SnCobB, we postulate

proteolytic processing of SnCE1 being elicited upon injection of SnCE1 into the host cell initiated by enzymatic deacetylation catalysed by host cell deacetylases (Figs. 5a,f and 7a, and Supplementary Fig. 21c).

To narrow down the region processed by this autoproteolytic cleavage we expressed the N-terminally truncated SnCE1$_{104-310}$ as wild type and catalytically dead SnCE1 C256A mutant protein (Supplementary Fig. 22b). For these truncated proteins we did not observe proteolytic processing suggesting that processing occurs from the N-terminus (Supplementary Fig. 22b). Apart from the truncation at the N-terminus SnCE1$_{104-310}$ and the mutant SnCE1$_{104-310}$ C256A recapitulate all effects observed for the longer SnCE1 construct, i.e., SnCE1$_{74-310}$. SnCE1$_{104-310}$ wild type protein is active as deSUMOylase (Fig. 2a) and reacts with the SUMO1-PA-/SUMO3-PA-probes suggesting it to be properly folded and active (Figs. 1a and 5d). Besides, the wild type SnCE1$_{104-310}$ is acetylated as shown by immunoblotting and it elutes as monomer from analytical SEC (Fig. 4e, and Supplementary Fig. 22b). In contrast, SnCE1$_{104-310}$ C256A is not acetylated and elutes in a monomer-tetramer equilibrium as judged by analytical SEC, similar as observed for longer SnCE1 C256A constructs (Supplementary

Figs. 1, 2c and 22b). These data suggest the residues covering side-chains 74-103 neither modulate the SnCE1 activity nor its oligomeric state. Moreover, this indicates the proteolytic cleavage site should be located in the region between residue 74 and 104 in the N-terminal region (Supplementary Figs. 12 and 22b). Next, we analyzed whether N-terminal proteolytic processing of SnCE1 also occurs in human cells.

### SnCE1 is N-terminally proteolytically processed in eukaryotic cells

We expressed full length wild type SnCE1 and the non-acetylated mutants SnCE1 Y212A and SnCE1 C256A with C-terminal myc-His$_6$-tag in HEK-293 cells and performed immunoblotting to analyze whether processing occurs from the N- or C-terminus (Fig. 7a, and Supplementary Fig. 25a). We obtained signals for both full length SnCE1 and the processed SnCE1 confirming SnCE1 being processed by app. 10 kDa from the N-terminus encompassing the predicted transmembrane region (Fig. 7a, and Supplementary Figs. 2b and 25a). Treatment of cells with deacetylase inhibitors, i.e., SAHA and TSA, to inhibit classical HDACs and nicotinamide (NA) to inhibit sirtuins, shows that SnCE1 acetylation increases suggesting acetylation to be enzymatically regulated by human deacetylases (Fig. 7b, and Supplementary Fig. 25b). We observed only the full length SnCE1 being lysine acetylated whereas the autoproteolytically processed SnCE1 is not acetylated, which supports the conclusion that SnCE1 deacetylation is a prerequisite for proteolytic processing of SnCE1 (Fig. 7a,c, and Supplementary Fig. 25b). Comparison of the mutant SnCE1 Y212A and wild type SnCE1 reveals SnCE1 Y212A shows a significantly stronger autoproteolytic processing, i.e., app. 40% of SnCE1 Y212A being processed. For comparison, app. 25% is processed of SnCE1 wild type (Fig. 7a). The mutant SnCE1 C256A shows a tendency, however not statistically significant, for less efficient processing compared to wild type SnCE1 (Supplementary Fig. 25a). From these results, we conclude that the deSUMOylase activity is not essential for autoproteolytic processing of SnCE1. As SnCE1 Y212A shows the strongest processing, its existence as monomer, as shown by analytical SEC, together with being non-acetylated, might account for the improved autoproteolytic processing compared to SnCE1 C256A and wild type SnCE1. Although the mutant SnCE1 Y212A shows that the acetylation state of SnCE1 directly affects autoproteolytic processing, the treatment of the cells with deacetylase inhibitors for 6 h or 12 h before harvesting did not interfere with proteolytic processing of wild type SnCE1. This suggests that processing occurs on the non-acetylated or deacetylated fraction of SnCE1 before the inhibitor treatment becomes effective (Fig. 7b, c, and Supplementary Fig. 25a,b).

Overall, these mutational studies provide evidence for SnCE1 lysine acetylation counteracting N-terminal autoproteolytic processing of SnCE1. Mechanistically stabilization of the SnCE1 non-acetylated monomer might contribute to this effect of SnCE1 acetylation, as exemplified by the observation that monomeric, non-acetylated SnCE1 Y212A shows stronger processing compared to tetrameric, non-acetylated SnCE1 C256A and monomeric, acetylated wild type SnCE1 (Fig. 7a, and Supplementary Fig. 25b). The observation that SnCE1 Y212A shows a stronger processing than the likewise monomeric wild type SnCE1 might be due to the fact SnCE1 must be deacetylated initially to elicit proteolytic processing, whereas SnCE1 Y212A is produced in a non-acetylated state. The autoproteolytic processing results in the formation of a defined SnCE1 truncation product, suggesting N-terminal proteolytic cleavage occurring at a specific site in terms of precision autoproteolysis by an unknown mechanism. We next investigated the mechanism underlying autoproteolytic processing and whether modulation of SnCE1 autoproteolytic processing can be attributed to individual lysine acetylation sites, as demonstrated for SnCE1 oligomerization.

### Acetylation of SnCE1 at multiple sites impairs autoproteolysis

We discovered SnCE1 autoproteolytic processing is initiated upon SnCE1 deacetylation (Figs. 5a and 7a). To show whether SnCE1 precision autoproteolysis is regulated by acetylation at individual lysine acetylation site(s) we analyzed the site-specifically acetylated SnCE1 C256A proteins produced by the GCEC regarding its proteolytic processing (Fig. 7c, and Supplementary Fig. 21a). As all proteins were constructed on the background of non-acetylated and proteolytically processed SnCE1 C256A this allowed to also assess a potential effect of site-specific lysine acetylation on SnCE1 autoproteolytic processing next to their impact on their oligomeric states (Figs. 5a and 7c, and Supplementary Fig. 20). As we expect full length SnCE1 being N-terminally processed by app. 10 kDa, acetylation of lysine side-chains within this N-terminal region might directly affect autoproteolysis. Alternatively, acetylation sites in the CE-clan protease domain might also interfere with the autoproteolytic mechanism. According to the structures solved here and the AlphaFold3 model, two sites, i.e., Lys209 and Lys248, were in the vicinity to the bound CoA, and acetylation might therefore sterically and electrostatically interfere with CoA binding (Figs. 1a and 4b,d). Our data show acetylation of SnCE1 C256A at Lys103, Lys106 and Lys231 in the CE-clan domain does not interfere with proteolytic processing as it is almost quantitatively processed to similar extend as observed for SnCE1 C256A (Fig. 7c). In contrast, the site-specifically acetylated proteins SnCE1 C256A AcK78, AcK94, AcK98 and AcK248 show an impaired autoproteolytic processing when comparing with SnCE1 C256A (Fig. 7c, and Supplementary Figs. 20 and 21c). However, none of these analyzed individual lysine acetylation modifications completely block SnCE1 autoproteolysis, suggesting modification at multiple sites might have an additive effect. As our data show, SnCE1 proteolytic processing occurs upon SIRT1-catalysed SnCE1 deacetylation, and AcK209 and AcK248 are not deacetylated by SIRT1, we conclude that Lys78, Lys94 and Lys98 in the N-terminal region are acetylation sites being involved in SnCE1 autoproteolytic processing. At this stage, we were not able to fully resolve the mechanism underlying SnCE1 autoproteolytic processing.

To obtain further insights into the molecular mechanism underlying SnCE1 autoproteolytic processing, we constructed a mutant of SnCE1, i.e., SnCE1 C256A S/T/C_to_A. In this mutant, several Ser/Thr/Cys residues in the sequence range of Ser74-Ser90 (SnCE1 C256A S74A/T76A/S77A/C81A/ S84A/S86A/S87A/S90A), in which we expect the autoproteolysis site to reside, were mutated to Ala and analyzed regarding its autoproteolysis. In several reported examples, cis-autoproteolysis often proceeds by a nucleophile, i.e., Ser/Thr/Cys, attacking the preceding peptide bond[71–75]. We produced and purified the recombinant SnCE1 C256A S/T/C_to_A mutant protein and additionally, we transiently expressed this mutant in HEK-293 cells. Both results show SnCE1 C256A S/T/C_to_A is still processed, suggesting an autoproteolytic processing mechanism not involving these residues (Supplementary Fig. 25c). As SnCE1 lysine acetylation at several regulated lysine side-chains, i.e., AcK78, AcK94 and AcK98, impairs autoproteolysis these residues might be important for autoproteolytic processing. As SnCE1 AcK209 located at the rim of the active site, shows an impaired autoproteolysis, which indicates an intact and accessible active site might be important for autoproteolysis. Further studies are needed to explore the underlying mechanism for SnCE1 autoproteolysis.

As a model, this autoproteolytic processing initiated by SnCE1 deacetylation by host cell deacetylases at several acetylated lysine side-chains would result in removal of the N-terminal transmembrane region with consequences on the subcellular distribution in the host cell, favoring the cytosolic/mitochondrial localization (Supplementary Fig. 2b). We next performed cellular studies to further investigate the consequences of SnCE1 autoproteolysis.

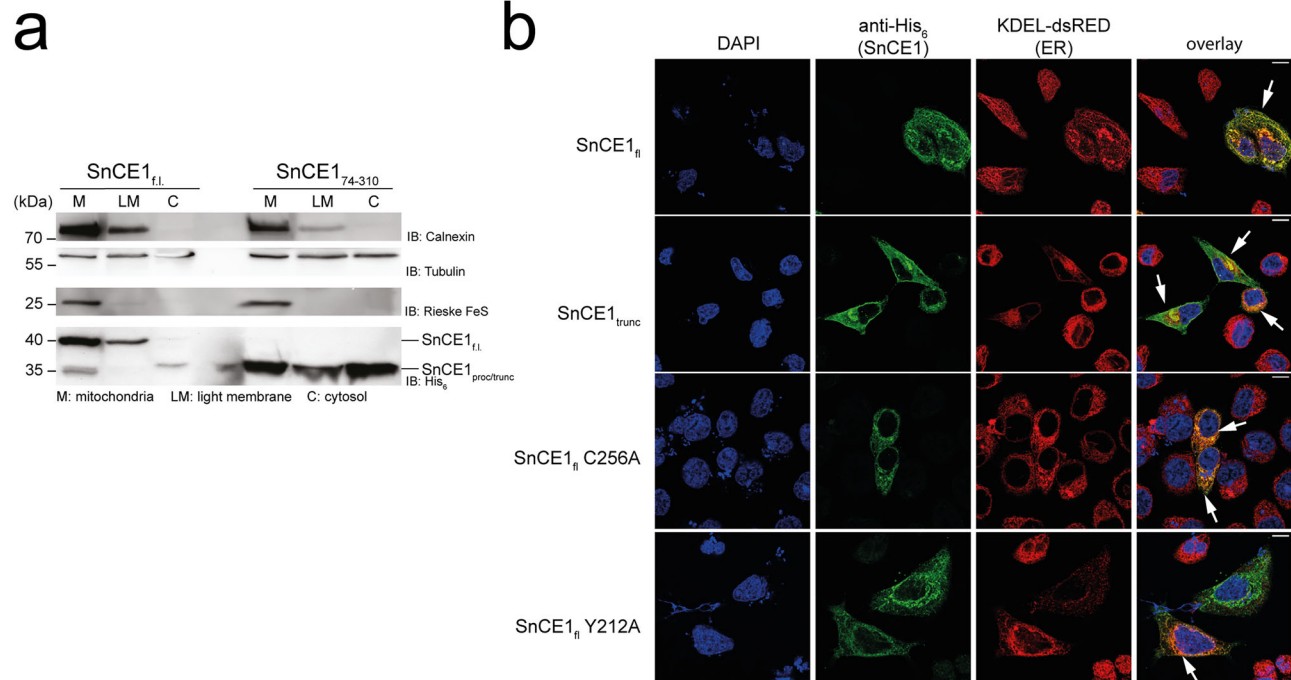

**Fig. 8 | SnCE1 localizes to the endoplasmic reticulum and processing favours cytosolic distribution. a** Subcellular fractionation of HEK-293T cells either expressing full length SnCE1 or truncated SnCE1$_{74\text{-}310}$. The full length SnCE1 was identified in the mitochondrial fraction (M) and the light membrane fraction (LM) containing ER and Golgi-membranes. The cytosolic fraction (C) contained exclusively processed SnCE1, i.e., SnCE1$_{trunc}$, suggesting autoproteolytic processing resulting in cytosolic distribution of SnCE1. In contrast, upon expression of truncated SnCE1$_{74\text{-}310}$ it can be found in all three fractions to a similar extend. Staining was done by anti-Calnexin antibody for the light membrane fraction, anti-Tubulin antibody for cytosolic fraction and anti-Rieske FeS antibody for mitochondrial fraction. SnCE1 was stained with anti-His$_6$ antibody. The experiment was performed in two independent biological replicates (*n* = 2), one example is shown. Source data are provided as Source Data file. **b** SnCE1 full length and N-terminally truncated SnCE1$_{74\text{-}310}$ was expressed in HeLa cells containing ER-targeted dsRED (KDEL-dsRED, red channel). Full length and truncated SnCE1 localize to the endoplasmic reticulum. The mutants SnCE1 C256A and SnCE1 Y212A behave similar as wild type SnCE1 suggesting localization at the ER membrane is possibly independent of its deSUMOylase activity. Truncated SnCE1 shows a more cytosolic distribution confirming the N-terminus being important for membrane binding. Staining of the DNA was done with DAPI, SnCE1 full length and SnCE1$_{74\text{-}310}$ was stained with anti-His$_6$-antibody (with Alexa488-coupled secondary antibody, green channel). The overlay of protein and ER signal was assessed (white arrows). The scale bar is 10 μm.

## SnCE1 locates to the endoplasmic reticulum with observed increase in mitochondrial fragmentation

To confirm the N-terminus being important for subcellular localization of SnCE1 in the host cell, the SnCE1 acetylation state modulating its autoproteolytic processing and to uncover the cellular processes regulated by SnCE1 in the host cell, we next performed subcellular fractionations and microscopy of human cells expressing SnCE1 and variants thereof. Notably, although we raised an antibody against SnCE1 we were not able to detect SnCE1 at endogenous levels in *S. negevensis*-infected human cells, in microscopy experiments and only weakly in Western blots. To this end, we transiently transfected HEK-293T cells with vectors encoding full-length wild type SnCE1 and truncated SnCE1$_{74\text{-}310}$ with a C-terminal myc-His$_6$-tag and performed subcellular fractionation to reveal to which cellular compartments SnCE1 localizes and whether the autoproteolytic processing modulated by the SnCE1 acetylation state affects the localization of SnCE1. To assess the localization, we studied the presence of SnCE1 in the light membrane fraction (LM) representing localization to the ER and Golgi apparatus, the crude mitochondrial fraction (M) and cytosolic fraction (C) (Fig. 8a). These studies show enrichment of full length SnCE1, i.e., SnCE1$_{fl}$, predominantly in the light-membrane fraction (LM: ER and Golgi) but not in the cytosolic fraction (C). Moreover, full-length SnCE1 was also present in the mitochondrial fraction (M), which can be explained by the contamination of this crude fraction by the ER membranes, as supported by calnexin staining (Fig. 8a). In contrast, the endogenously autoprocessed SnCE1 protein, i.e., SnCE1$_{proc}$, was present also in cytosolic (C) fractions. These data might indicate autoproteolytic processing of full-length SnCE1, resulting in release

from membranes as processing results in cleavage of the N-terminal transmembrane region (Fig. 8a). As control, we also expressed the truncated SnCE1$_{74\text{-}310}$, i.e., SnCE1$_{trunc}$. Truncated SnCE1$_{74\text{-}310}$ can be identified in all fractions to a similar extend (Fig. 8a). These data indicate the N-terminal region in SnCE1 contains a functional transmembrane region and separation of this region upon autoproteolysis results in a more cytosolic distribution of SnCE1.

To analyze which cellular processes are affected by expression of SnCE1 and to confirm the subcellular localization of SnCE1 in human cells we next performed microscopy of HeLa cells transiently expressing full length SnCE1, i.e., SnCE1$_{fl}$, as well as truncated SnCE1$_{74\text{-}310}$, i.e., SnCE1$_{trunc}$ (Figs. 8b and 9a). As indication for SnCE1 acetylation affecting localization of SnCE1 and/or cellular processes, we additionally expressed the non-acetylated, active full-length SnCE1 Y212A and non-acetylated, inactive full-length SnCE1 C256A (Figs. 8a,b and 9a). These studies show full-length SnCE1 primarily localizing to the ER as visible by the overlay of the ER marker KDEL-dsRed with the green SnCE1 anti-His$_6$ fluorescence (Fig. 8b). The mutants SnCE1$_{fl}$ C256A and SnCE1$_{fl}$ Y212A show a similar localization, suggesting a functional deSUMOylase activity not being essential for its localization to the ER. As observed for SnCE1$_{fl}$, the truncated SnCE1$_{trunc}$ lacking the N-terminal transmembrane region, also localizes to the ER. However, in contrast to SnCE1$_{fl}$ it shows a more cytosolic distribution (Fig. 8b). This supports our hypothesis that removing the N-terminal transmembrane region results in a stronger cytosolic intracellular distribution in the host cells.

Next to the localization to the ER, SnCE1 shows strong co-localization with a mitochondrial marker (mito-GFP), suggesting an

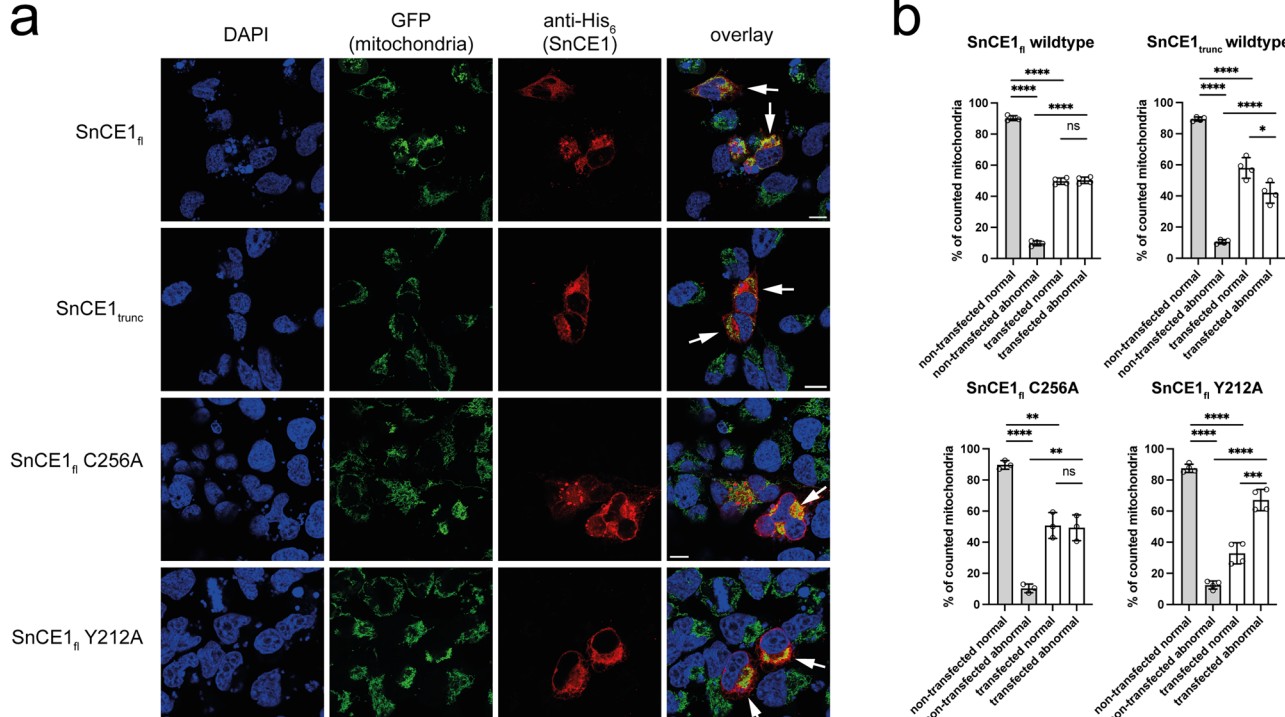

**Fig. 9 | SnCE1 induces increased fragmentation of mitochondria. a** SnCE1 full length and N-terminally truncated SnCE1$_{74\text{-}310}$ was expressed in HeLa cells containing mitochondria-targeted GFP (green channel). Expression of SnCE1 and the mutants SnCE1 C256A and SnCE1 Y212A result in formation of fragmented mitochondria. As shown in **b**, truncated SnCE1 shows a more cytosolic distribution. Overall SnCE1 Y212A shows the most fragmented mitochondria. Staining of the nucleus was done with DAPI, SnCE1 full length and SnCE1$_{74\text{-}310}$ was stained with anti-His$_6$ antibody (with Alexa555-coupled secondary antibody). To show localization of SnCE1 to mitochondria and the induced mitochondrial fragmentation the overlay was assessed (white arrows). The scale bar is 10 μm. **b** Quantification of the mitochondrial fragmentation observed upon expression of SnCE1 in HeLa mito-GFP

cells. The fluorescence images shown in **c** were quantified by counting at least 50 cells for each category and the number of cells with non-fragmented and fragmented mitochondria was expressed as % of all cells in that category. The expression of SnCE1 increases the formation of fragmented mitochondria. Comparing wild type SnCE1 or SnCE1 C256A shows a similar percentage of normal and fragmented mitochondria. However, for SnCE1 Y212A a significantly higher percentage of fragmented mitochondria was observed. Experiments were performed in two biological replicates each consisting of two technical replicates ($n = 4$). Bars depict mean ± SD of determined miller units. Statistical significance ($^*p < 0.05$; $^{**}p \leq 0.01$; $^{***}p \leq 0.001$; $^{****}p \leq 0.0001$; ns: not significant) was tested using unpaired, two-sided $t$-tests. Source Data are provided as Source Data file.

enrichment of SnCE1 at mitochondria or at ER-mitochondria contact sites. This is accompanied by an alteration of mitochondrial morphology, i.e., an observed increase in formation of fragmented mitochondria (Fig. 9a, and Supplementary Fig. 26). This was visible for all constructs of SnCE1, including the non-acetylated, catalytically dead mutant SnCE1 C256A and the non-acetylated, active mutant SnCE1 Y212A. This suggests the catalytic deSUMOylase activity is not essential for this phenotype. The quantification of the mitochondrial fragmentation revealed it was most pronounced for the non-acetylated, active mutant SnCE1$_{fl}$ Y212A (Fig. 9b). From our data, we cannot derive which features of SnCE1 Y212A, i.e., being monomeric, non-acetylated, autoproteolytically processed and/or active as deSUMOylase, are the major factors responsible for this phenotype. However, it suggests that the SnCE1 acetylation state has direct or indirect consequences on mitochondrial morphology. Further studies are needed to understand the mechanism underlying this observation, the substrates of SnCE1, and the (patho)physiological role of SnCE1 in *S. negevensis* infection.

## Discussion

We performed structure-function analyses on the CE-clan protease-related virulence factor SnCE1 from the Gram-negative bacterial species *S. negevensis*, which is closely related to ChlaDUB1 from *C. trachomatis*[56]. We confirmed that SnCE1 is a deSUMOylase rather than a DUB, although it has some remaining DUB activity towards Lys63-, Lys48- and Lys11-linked diUb (Fig. 1)[26]. Structurally, SnCE1 lacks the α-helix in the VR-3 but still has a dual activity as deSUMOylase and AcT.

Our data show SnCE1 is able to create specificity towards SUMO chains over Ub chains by forming several specificity-creating interactions to bind SUMO in S1 position. The Gly-rich VR-3 β-turn connecting β-strands β1 and β2 of the central β-sheet contributes to binding (acetyl-) CoA in the S1' site. This shows that during evolution, several strategies were developed to achieve this dual activity.

SnCE1 is a true moonlighting enzyme, i.e., it has the primary catalytic activity acting as deSUMOylase, in which S1 and S1' are bound by a SUMO2/3 chain, by a SUMO2/3-SUMO1 chain or by a non-SUMO substrate capped with SUMO1. However, SnCE1 has a second catalytic activity, specifically acting as autoacetyltransferase (auto-AcT), in which the S1' position is bound by acetyl-CoA needed as acetyl group donor molecule. From a structural perspective, our data indicate that binding of a SUMO moiety or of a non-SUMO substrate to the S1' site and binding of (acetyl-)CoA are mutually exclusive. As a consequence, if SnCE1 was able to acetylate substrate proteins in trans as described for ChlaDUB1, no ternary complex between acetyl-CoA, SnCE1 and substrate protein at S1' could be formed[58]. Instead, the acetylation substrate would need to bind to S1 if the mechanism included the formation of a ternary complex in a sequential catalytic mechanism. Alternatively, if the acetylation substrates were to bind to the S1' site a ping pong catalytic mechanism would be likely, i.e., the acetyl group is then transferred from acetyl-CoA to the active site Cys256, forming a covalent acyl-enzyme adduct, and afterwards CoA is released and a substrate protein could bind to S1' site of SnCE1 following acetyl group transfer to the substrate's lysine side chain. In this

mechanism described for the related YopJ class of acetyltransferases, the catalytic triad His-base was reported to play a catalytic role also for the AcT activity, activating the substrate lysine side chain for nucleophilic attack on the acetyl-cystein adduct[41]. Our data indicate a potential acetyl-cystein state only occurring transiently as acetylated wild type SnCE1 is active as deSUMOylase and an accessible Cys256 is essential for catalysis. In turn, this means the acetyl group must be removed from the catalytic Cys256, maybe by transferring it to SnCE1 lysine side chains. Future studies are needed firstly to clarify a potential catalytic mechanism and secondly whether SnCE1 is able to acetylate protein substrates in trans similar to that described for ChlaDUB1[58].

We identified several lysine acetylation sites covering the SnCE1 protein. This raises the question of how SnCE1 is capable to acetylate all of these sites. Our LC-MS/MS data reveal that acetylation at identified sites is significantly reduced or even not detected in the SnCE1 Y212A mutant compared to wild type SnCE1. To this end, we propose that these modifications are the results of a bona fide autoacetylation reaction. Our data indicate the catalytic Cys256 is needed for the auto-AcT activity, suggesting the lysine side chains must come into close spatial proximity to the active site in order to become acetylated. Several lysine side chains are located in a highly flexible N-terminal region, which might be accessible for acetylation. However, other sites are found within the CE-clan protease domain, and it is not directly obvious how mechanistically these are acetylated. It is tempting to speculate that these sites become acetylated due to the increase of the local concentration of acetyl-CoA at the SnCE1 protein in either its monomeric or tetrameric state. Residues such as Lys209 and Lys248 are directly pointing towards the CoA molecule in the AlphaFold3 model, suggesting these might take over the acetyl groups from acetyl-CoA acting as nucleophiles upon binding of acetyl-CoA in a potential unproductive conformation before adopting its productive conformation (Figs. 4d and 5e).

Our data show auto-AcT activity is used by SnCE1 to regulate important protein functions, i.e., affecting autoproteolytic processing needed for efficient cytosolic and mitochondrial distribution in the host cell and to stabilize the SnCE1 monomer, thereby improving its deSUMOylase activity as structurally tetramer formation and SUMO binding at the S1 site are mutually exclusive. The structure of the tetrameric form of the non-acetylated SnCE1 C256A suggests binding of (acetyl-)CoA at the S1′ site being possible, while SUMO-binding at S1 is not due to steric clashes with neighboring subunits. This might indicate autoacetylation in SnCE1 at sites including Lys231, shifting the equilibrium from the tetrameric form towards the monomeric form, acting as an efficient deSUMOylase.

We were not able to attribute the impact of lysine acetylation on SnCE1 autoproteolytic processing to an individual lysine acetylation site. Instead, we observe acetylation of SnCE1 at multiple acetylation sites, i.e., Lys78, Lys94 and Lys98, in the N-terminal region, impairing autoproteolytic processing. Thus, we postulate that these multiple lysine acetylation events might be additive. We show deacetylation of wild type SnCE1 by NAD⁺-dependent host cell deacetylases but not by S. negevensis SnCobB, triggering autoproteolytic processing, resulting in the removal of the transmembrane helix-containing N-terminal region. This in turn might allow efficient release of SnCE1 from the S. negevensis containing vacuole membrane and modulate its subcellular localization in the host cells, i.e., increasing cytosolic and mitochondrial distribution. Similarly, in other Ulps/SENPs a region preceding the catalytic domain in the enzymes' N-terminus was shown to regulate their intracellular distribution[76,77].

We observed that SnCE1 localizes to the ER upon expression and affects mitochondrial morphology, resulting in fragmented mitochondria. This is in agreement with earlier studies describing that *Simkania* containing vacuoles (SnCV) are in close contact to the ER and recruit and affect mitochondria of infected host cells[47,78]. Notably, analysis of the SnCE1 sequence by DeepLoc (https://services.

healthtech.dtu.dk/services/DeepLoc-2.0/) confirms our experimental findings, suggesting that SnCE1 localizes to the ER and to the cytosol (Supplementary Fig. 27)[79]. The exact mechanism underlying the observation that SnCE1 results in fragmentation of mitochondria is unclear, but it might be due to a dysregulation of mitochondrial fission and/or fusion, for which ER is known to play a role[80]. The strongest phenotype of mitochondrial fragmentation was observed for monomeric, non-acetylated and deSUMOylase-active SnCE1 Y212A. The degree of autoproteolytic processing correlates with the severity of mitochondrial fragmentation as SnCE1 Y212A was the most efficiently processed SnCE1 variant. Notably, we also observed mitochondrial fragmentation, albeit less severe, upon expression of monomeric, acetylated and catalytically active wild type SnCE1 and of tetrameric, non-acetylated and catalytically inactive SnCE1 C256A. Comparison of the results obtained for these SnCE1 variants indicates the SnCE1 acetylation level and its consequences stabilizing the SnCE1 monomeric state directly correlating with the degree of autoproteolytic processing and mitochondrial fragmentation. The fact that wild type SnCE1 shows less mitochondrial fragmentation compared to SnCE1 Y212A, although both are monomeric and active deSUMOylases, might be because the proteolytic processing is only triggered after SnCE1 deacetylation for wild type SnCE1 while SnCE1 Y212A is not acetylated when it is produced and autoproteolytic processing can start immediately following translocation into the host cell as supported by the more pronounced processing observed for SnCE1 Y212A in HEK-293T cells (Fig. 7a). These comparisons furthermore suggest mitochondrial fragmentation at least being partly evoked independently from SnCE1 catalytic deSUMOylase activity as SnCE1 C256A also induces mitochondrial fragmentation to some extent. This discovery might have implications also for SnCE2-5 for which no enzymatic activity could be identified so far maybe exerting a dominant negative effect[26]. Notable, SnCE2-5 lack the aromatic gatekeeper, i.e., Trp191 in SnCE1, known to mediate specificity towards ubiquitinated/SUMOylated substrates, indicating these might be proteases with other unknown substrates (Supplementary Fig. 12,16). Along this line, for other eukaryotic DUBs/ULPs, functions were reported which were independent of their catalytic activities[81,82].

We have to stress that we analyzed the impact of SnCE1 on cellular function by expressing SnCE1 and variants thereof in non-infected cells. This was due to the fact that we were not able to raise an antibody enabling robust detection of SnCE1 at endogenous levels in *S. negevensis* infected cells, although SnCE1 is the most expressed CE-clan protease-related virulence factor in *S. negevensis*[26]. As *S. negevensis* can to date not be genetically manipulated, we are also not able to conduct a genomic deletion of the gene encoding SnCE1 to understand the role and impact of SnCE1 on *S. negevensis* infection. Future *S. negevensis* infection studies need to reveal SUMOylated SnCE1 host cell substrates. Notably, we identified K524-SUMO1ylated RanGAP1, which is located at the cytosolic site of the nuclear pore and acts as GTPase-activating protein for the small-GTP-binding protein Ran, as substrate for SnCE1 in vitro, showing SnCE1 being capable of removing SUMO1 from a non-SUMO target protein (Supplementary Fig. 28).

Our proposed model suggests that the specificity to act as deSUMOylase might preferentially be determined by the specificity in targeting SUMO1 or SUMO2/3 in the S1 site. Our data show SnCE1 is capable of cleaving SUMO chains acting as endo-isopeptidase, i.e., cleaving isopeptide bonds within the chain. Besides, as it reacts with SUMO1-PA/SUMO3-PA probes, suggesting it to be active as exo-isopeptidase acting at the distal end of SUMO2/3 chains terminated by SUMO2/3 or SUMO1. Furthermore, it can act at the proximal end, removing SUMO1 from non-SUMO substrates as demonstrated here for RanGAP1-SUMO1[83,84]. It was shown that the mitochondrial SUMOylation state affects mitochondrial morphology and function[85,86]. Other SUMO-deconjugating enzymes such as human sentrin-specific proteases, i.e., SENP3 and SENP5, were reported to

localize to the mitochondria, playing a role on mitochondrial fission and mitophagy[87,88]. Further SUMOylated substrates of SnCE1 in ER-mitochondria contact sites and/or the outer mitochondrial membrane, regulating mitochondrial fission or fusion, such as dynamin-related mitofusins or DRP1, might be identified in the future[86,89–95].

Interestingly, prediction of SUMO-interaction motifs and SUMOylation sites in SnCE1 suggests SnCE1 might be SUMOylated and interacting with SUMO (Supplementary Table 2). Amongst those residues, Lys231 was predicted to be SUMOylated. This might indicate a crosstalk between lysine acetylation and SUMOylation in SnCE1.

Notably, AlphaFold3 is not able to predict the correct structure for the SnCE1 tetramer (Supplementary Fig. 15). This might be due to the fact the SnCE1 tetramer being of moderate stability existing in an equilibrium with a monomeric state as shown by analytical SEC. This is supported by the interface area between the SnCE1 subunits in the tetramer (724 Å$^2$), indicating the dissociation equilibrium constant of the individual SnCE1 molecules forming the oligomer to be of medium affinity.

*S. negevensis* was shown as other *Chlamydia* to use a T3SS for injection of virulence factors into host cells. Furthermore, genes encoding parts of a T4SS were identified on a large conjugative plasmid in *S. negevensis*[56]. Often pathogenic bacteria inject membrane-bound and transmembrane virulence factors into host cells[96–98]. Data on the T3SS and the T4SS provide evidence that substrate proteins are fully unfolded when they are translocated into the host cells[99,100]. To this end, it is speculative at which stage acetylation of SnCE1 occurs either in the folded state within the *S. negevensis* cells before getting (partially) unfolded for translocation or following translocation in the host cell. Our data indicate that the irreversible acetylation at Lys231 would stabilize the SnCE1 monomer with potential consequences on unfolding and translocation of SnCE1 by the secretion machinery. Secondly, the acetylation in the N-terminal region, shown to impair autoproteolytic processing, would ensure a binding of SnCE1 to the vacuole membrane in early phases of infection upon exposure of SnCE1 to the host cell. The processing is afterwards triggered by deacetylation of SnCE1 by host cell deacetylases.

The exact SnCE1 acetylation state depends on several factors such as the cellular ratio of acetyl-CoA:CoA as acetyl-CoA and CoA might compete for binding to SnCE1 as shown for other lysine acetyltransferases[101,102]. Moreover, the cellular NAD$^+$-level is important as it is needed as a stoichiometric co-substrate used by host cell sirtuins for SnCE1 deacetylation. These molecules represent the energy state in the host cells. Under conditions of high cellular acetyl-CoA, SnCE1 might be acetylated, resulting in blockage of autoproteolytic processing. Upon a decrease in cellular acetyl-CoA and an increase in cellular CoA and NAD$^+$ levels, monomerization is constant due to irreversibility in Lys231 acetylation, but autoproteolytic processing and cytosolic/mitochondrial distribution is increased due to SnCE1 deacetylation. This elegant mechanism allows precise coordination of the virulence factor activity in time and space depending on the metabolic state of the host cells, which directly contributes to a successful infection of host cells by *S. negevensis*.

Notably, the mechanism underlying the precise autoproteolytic processing of SnCE1 is unresolved. Several mechanisms underlying autoproteolysis were reported, but which is relevant for SnCE1 is unknown[75,103]. Often, the reported mechanisms underlying proteolysis in cis, i.e., autoproteolysis, involve attack of a nucleophile, i.e., either Ser, Thr or Cys side chains, on the N-terminally preceding peptide bond through an N-O(S) acyl shift resulting in formation of an instable ester, which is hydrolyzed ultimately resulting in cleavage of the peptide chain[71–74,103–105]. Remarkably, SnCE1 contains a total of eight Ser/Thr or Cys side chains in the N-terminal region encompassing residues 74–90, i.e., 44%, which might act as nucleophiles in an autoproteolytic catalytic mechanism. We constructed the mutant SnCE1 S/T/CtoA, i.e., SnCE1 C256A S74A/T76A/S77A/C81A/S84A/S86A/

S87A/S90A, in which all potential nucleophiles in this region were mutated to Ala. However, this mutant still shows proteolytic processing in the recombinant protein as well upon transient expression in eukaryotic cells, suggesting processing does not use these residues for catalysis (Supplementary Figs. 22c,d and 25c). For related CE-clan-related virulence factors of the YopJ family it is reported to be able to also acetylate Ser and Thr side chains[106,107]. However, mass spectrometry data show no acetylated Ser or Thr residues in SnCE1.

Our data indicate acetylation at lysine side chains in SnCE1, establishing either direct or indirect mechanistic constraints to impair autoproteolysis as deacetylation of wild type SnCE1 by SIRT1-stimulates autoproteolysis. We analyzed the impact of acetylation on autoproteolysis at several lysine side chains located in the region we expect proteolytic processing of SnCE1 C256A to occur, i.e., Lys78, Lys94 and Lys98 in the N-terminal region, as well as AcK103, AcK106, AcK209 and AcK248 in the CE-clan protease domain. While we observed no autoproteolytic processing for recombinantly expressed and purified SnCE1 being acetylated at multiple lysine side chains, for SnCE1 C256A site-specifically acetylated at Lys78, Lys94, Lys98 (and Lys248) we observe an impaired SnCE1 autoproteolysis compared to non-acetylated SnCE1 C256A. This indicates an additive effect of acetylation at multiple lysine side chains rather than on a specific site ultimately blocking autoproteolysis (Fig. 7c). This in turn would allow for more precise adjustment of the level of SnCE1 autoproteolysis to the degree and stoichiometry of acetylation occurring at several sites compared to a mechanism depending on acetylation at one individual lysine side chain. The fact that we observe autoproteolytic processing after deacetylation of wild type SnCE1 by SIRT1 suggests acetylation at enzymatically accessible sites in the N-terminal region, i.e., Lys78, Lys94 and Lys98, to regulate autoproteolysis in cells rather than acetylation at Lys209 and Lys248 as these were not deacetylated by SIRT1 (Supplementary Fig. 21c). For acetylation at Lys209, directly pointing into the SnCE1 active site, we found a strongly impaired autoproteolysis suggesting the access and integrity of the active site in SnCE1 is important for SnCE1 autoproteolysis. Moreover, as SnCE1 is not deacetylated by the only deacetylase SnCobB in *S. negevensis* suggests autoproteolytic processing of SnCE1 following presentation of SnCE1 to the host cell cytosol resulting in release from the vacuole membrane and thereby improving cytosolic/mitochondrial distribution (Figs. 5f, 8a,b and 9a).

SnCE1 was shown earlier to be the most expressed of all five CE-clan-proteases in *S. negevensis*[26]. Moreover, for SnCE2-5, no enzymatic activity could be demonstrated so far not excluding that these have unknown (proteolytic) activities or a very narrow substrate range[26]. Along that line, we show catalytically inactive SnCE1 C256A inducing mitochondrial fragmentation to some extent similar as observed for wild type SnCE1, suggesting this phenotype may be elicited independently from its catalytic activity, maybe by having a dominant-negative effect. Similarly, catalytically inactive SnCE2-5 might interfere with host cell function; however, this needs further studies. Our results suggest that SnCE1 is a promising drug target to develop therapeutic strategies to fight *S. negevensis* infections. *S. negevensis* is not in the center of investigation and infections as *S. negevensis* are often not assessed in standard medical tests, due to the low pathogenic potential of this bacterium. A relevance of *S. negevensis* infection in the development of long-lasting respiratory diseases in human was shown, especially in immunocompromised individuals[48,50–52,55]. However, the ability of *S. negevensis* to survive within the amoeba host, similar to *L. pneumophila*, as well as its capability to infect almost all eukaryotic cell lines, including most human cell lines analyzed, suggests *S. negevensis* infections might be more distributed and therefore of higher medical relevance than originally believed.

Overall, our results suggest acetylation of SnCE1 structurally improves SnCE1's deSUMOylase activity by stabilizing the monomeric state by irreversible acetylation at Lys231. Besides, SnCE1 acetylation at

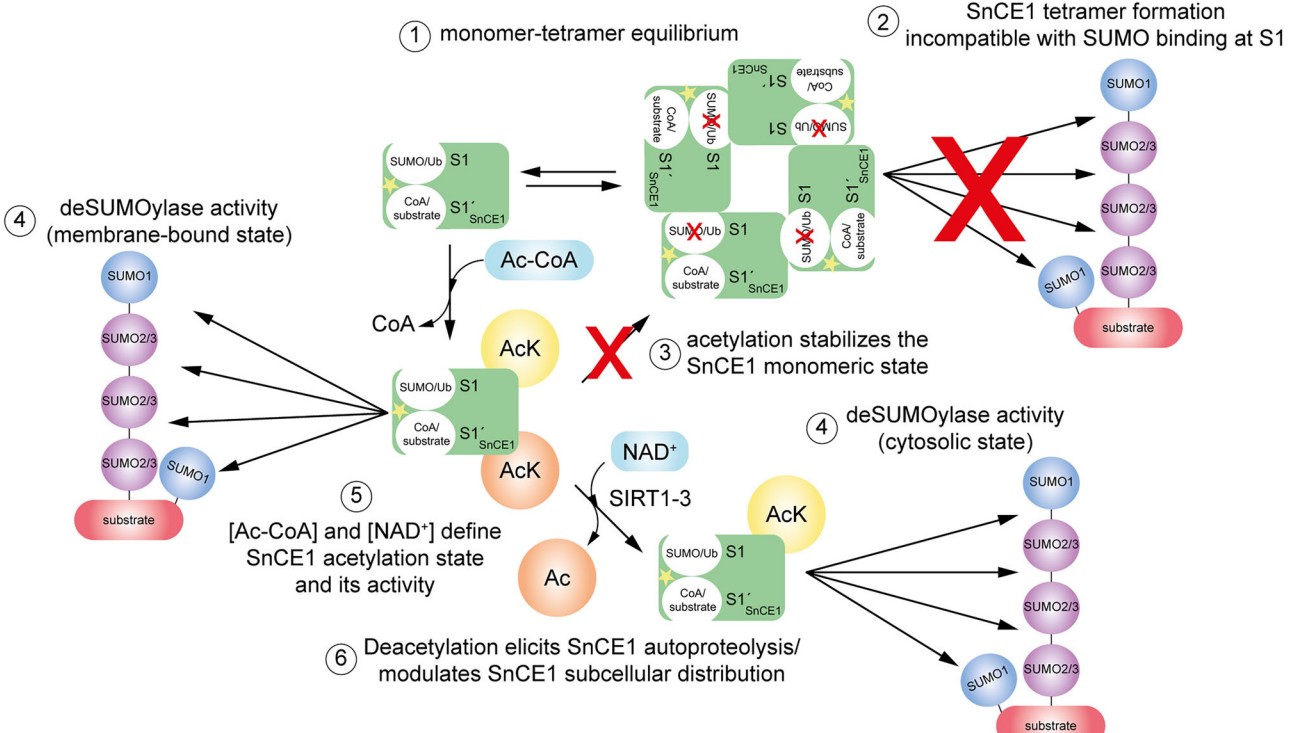

**Fig. 10 | Model for the concept of using lysine acetylation of SnCE1 to reprogram bacterial virulence.** Our data show non-acetylated SnCE1 being in a monomer-tetramer equilibrium (1). Structurally, SnCE1 is unable to bind SUMO at the S1 site in its tetrameric state, thereby impairing its deSUMOylase activity (2). We discover SnCE1 having an intrinsic autoacetyltransferase (auto-AcT) activity using acetyl-CoA as donor molecule. Binding of (acetyl-)CoA, a proximal SUMO or a non-SUMO substrate at S1' is compatible with SnCE1 tetramer formation. Lysine acetylation at Lys231 of SnCE1 stabilizes the SnCE1 monomeric state (3). SnCE1 is an efficient deSUMOylase in its membrane-bound and autoprocessed, cytosolic form (4). We provide evidence for SnCE1 being capable to cleave SUMO2/3 chains acting as endo-isopeptidase, cleaving the isopeptide bond connecting SUMO1 capping a SUMO2/3 chain acting as exo-isopeptidase at the distal end of a SUMO chain and also to remove SUMO1 from non-SUMO substrates, i.e., acting at the proximal end. The SnCE1 autoacetylation results in acetylation at some lysine side chains, including Lys94, Lys98, Lys103 and Lys106. These are reversed by eukaryotic

deacetylases, including SIRT1-3. Deacetylation of SnCE1 elicits autoproteolytic processing, removing the transmembrane helix-containing N-terminal region by an unknown mechanism favouring SnCE1 cytosolic distribution in the host cell (5). As a consequence, the cellular concentrations of acetyl-CoA and NAD$^+$ determine the acetylation state and activity of SnCE1 (6). At high concentrations of acetyl-CoA needed for autoacetylation and low concentrations of NAD$^+$, needed as stoichiometric co-substrate for deacetylation catalysed by sirtuins, SnCE1 exists primarily in the membrane attached form. Upon deacetylation of SnCE1 under cellular conditions of low acetyl-CoA and high NAD$^+$, SnCE1 is released from the *Simkania*-containing vacuole membrane and distributed in the cytosol and to the mitochondria. Our data indicate a so far undescribed concept using lysine acetylation to reprogram virulence by tightly coordinating the activity of virulence factors precisely in space and time. SnCE1 has a primary activity as deSUMOylase and a second moonlighting AcT activity enabling to adjust its activity to the cellular metabolic state.

multiple lysine side chains in the N-terminal region, including Lys78, Lys94 and Lys98, directly preceding the CE-clan domain is important to stabilize the unprocessed, membrane-attached form of SnCE1. Upon deacetylation, driven by sirtuins and/or other deacetylases in the host cell, SnCE1 is autoproteolytically processed, modulating its subcellular localization, i.e., processed SnCE1 being more cytosolic. SnCE1 localizes to the ER and directly affects mitochondrial morphology, i.e., SnCE1 results in fragmented mitochondria, suggesting SnCE1 either directly or indirectly targets the mitochondrial fission and/or fusion machinery. Our findings suggest that pathogenic intracellular bacteria use sophisticated regulatory mechanisms to precisely adjust and coordinate the activity and effectivity of virulence factors within the host cell in time and space (Fig. 10). Using post-translational lysine acetylation to coordinate the activity of the virulence factors allows the adjustment of their activity to the availability of (acetyl-)CoA and NAD$^+$. This represents a so far unexplored mechanism of how pathogens are able to reprogram their virulence and to precisely adjust their infectivity to the environment, i.e., the metabolic state, and to their own fitness and/or the fitness of the infected host cells. The post-translational modification of virulence factors puts another level of complexity on the regulation of their activity. This, however, increases

the molecular targets that can be therapeutically tackled to control bacterial infections[108].

## Methods

### Plasmids, enzymes and inhibitors

**Plasmids.** For expression in bacterial cells, synthetic codon-optimized genes in pGEX6P.1 (BioCat GmbH, Heidelberg, DE) were used. Mutations at indicated sites were introduced by site-directed mutagenesis according to the PCR Protocol for Phusion High-Fidelity DNA Polymerase (New England Biolabs GmbH, Frankfurt/Main DE). For cloning *E. coli* DH5α cells were used. Truncated constructs of SnCE1 were generated by Gibson assembly and cloned into pGEX6P.1 (95–310, 104–310) and pOPIN-S (83–310). For cloning, primers (Sigma Aldrich GmbH, Taufkirchen DE), Phusion DNA polymerase, Taq-DNA-ligase, T5 exonuclease and restriction enzymes were used (New England Biolabs GmbH, Frankfurt/Main DE). The oligonucleotides used for cloning and mutagenesis are listed in Supplementary Table 3. Site-specifically acetylated SnCE1 proteins were expressed from a pRSFDuet-1 vector additionally encoding the synthetically evolved *Methanosarcina barkeri* MbtRNA$_{CUA}$ and the acetyl-lysyl-tRNA-synthetase (AcKRS3) as described previously[73–75]. Constructs for mammalian expression were

ordered as synthetic codon-optimized genes in pcDNA3.1(+) myc_His A/B (BioCat GmbH, Heidelberg, DE). pOPIN-S constructs for SnCE2-5 and HsSENP1 (415 - 644), pET17b construct for HRV 3C protease, as well as pTXB1 constructs for PA-probes of ubiquitin, SUMO1 and SUMO3 were kind gifts from Dr. Thomas Hermanns (AG Hofmann, University Cologne).

### Enzymes and inhibitors

Deacetylation experiments were carried out using recombinantly expressed and affinity purified human sirtuin deacetylases SIRT1 (aa 225-664, N-terminal His$_6$-tag), SIRT2 (aa 50-356, with N-terminal GST-tag or N-terminal His$_6$-tag), SIRT3 (aa 118-399), SIRT5 (aa 34-302) and commercially obtained SIRT6 (full length, C-terminal His$_6$-tag, Sigma Aldrich, cat no. SRP5273) and SIRT7 (full length, C-terminal FLAG-tag, BPS Bioscience, cat no. 50018). Human classical deacetylases HDAC1 (full length, N-terminal His$_6$-tag, Enzo Life Sciences, cat no. BML-SE456-0050), HDAC2 (full length, N-terminal His$_6$-tag, Enzo Life Sciences, cat no. BML-SE500-0050), HDAC3/NCor2 (full length, HDAC3 C-terminal His$_6$-tag, NCoR2_DAD N-terminal GST-tag, BPS Bioscience, cat. no. 50003), HDAC4 (aa 627-1084, N-terminal GST-tag and C-terminal His$_6$-tag, BPS Bioscience, cat no. 50004), HDAC6 (full length, N-terminal GST-tag, Sigma-Aldrich, cat no.382180), HDAC7 (aa 518-952, N-terminal GST-tag, BPS Bioscience, cat no. 50007), HDAC8 (full length, C-terminal His$_6$-tag, Sigma-Aldrich, cat no. 382184), HDAC9 (aa 604-1066, C-terminal His$_6$-tag, BPS Bioscience, cat no.50009), HDAC11 (full length, BPS Bioscience, cat no. 50021) were of commercial source. To preserve autoacetylation of overexpressed SnCE1 in cell culture experiments, classical deacetylase inhibitors Trichostatin A/TSA (Tokyo Chemical Industry, cat no. T2477) and suberoylanilide hydroxamic acid/SAHA (Sigma-Aldrich, cat no. SML0061) as well as sirtuin deacetylase inhibitor nicotinamide/NA (Sigma Aldrich, cat no. 72340) were added.

### Expression and purification of proteins

**Heterologous expression in E. coli BL21 (DE3).** The catalytic domain of *S. negevensis* SnCE1 (UniProt: F8L4W9) wild type and mutants was expressed as fusion protein with N-terminal GST-tag in *E. coli* BL21 (DE3) cells using pGEX6P.1 vector with HRV 3C protease cleavage site (BioCat GmbH, Heidelberg DE). The catalytic domains of *S. negevensis* SnCE2-5 (UniProt: F8L8N5, F8L3F3, F8L927, F8L668) were expressed as fusion proteins with N-terminal His$_6$-Smt3-tag using the pOPIN-S vector enabling tag cleavage by human sentrin-specific proteases (SENP). Cells were grown in LB or TB full media at 37 °C, 130 rpm to an optical density (OD) of 0.6–0.8 before protein expression was induced by adding 200–400 μM of isopropyl-β-D-thiogalactopyranoside (IPTG, Carl Roth GmbH, cat no. 2316.1). Following cultivation at 18 °C for 16–20 h, cells were harvested and stored at −20 °C. A complete list of all DNA constructs used in this study is included in Supplementary Table 4.

### Affinity chromatographic purification of GST-tagged and His$_6$-Smt3-tagged proteins

All proteins were purified by affinity chromatography and subsequent SEC on an FPLC system using either ÄKTApurifier (Cytiva Germany GmbH, Freiburg, DE) or NGC Discovery (Biorad GmbH, Feldkirchen, DE). Every purification step was sampled and analyzed by SDS-PAGE with Coomassie Brilliant Blue (CBB) staining. For this, pelleted cells were thawed and resuspended in binding buffer (20 mM Tris-HCl pH 7.0, 300 mM NaCl, 2 mM β-mercaptoethanol) supplemented with 0.2 mM serine protease inhibitor AEBSF-HCl (Carl Roth GmbH, cat no. 2931.1). After cell lysis by using a high-pressure homogenizer HPL6 (Maximator GmbH, Nordhausen, DE), the crude lysate was centrifuged for 45 min at 23,545 × *g*, 4 °C and the supernatant passed through a 0.2 μM filter. For GST-tagged proteins, the filtered supernatant was applied to an equilibrated prepacked 5 mL Protino GST/4B FPLC column (MACHEREY-NAGEL GmbH & Co. KG, Düren, DE) and after

binding to the column matrix was washed extensively with high salt washing buffer (20 mM Tris-HCl pH 7.0, 500 mM NaCl, 2 mM β-mercaptoethanol). The column-bound fusion proteins were eluted by adding 20–25 mM reduced glutathione in binding buffer and adjusting the pH to pH 8.0. The GST-tag was cleaved overnight by adding HRV 3C protease to the elution fraction, and the solution was dialysed against binding buffer overnight at 4 °C. To remove liberated GST-tag and His$_6$-tagged protease, a 5 mL PureCube Ni$^{2+}$-NTA agarose column (Cube Biotech GmbH, Monheim a.R. DE) was coupled to a 5 mL Protino GST/4B column and a second round of affinity chromatography was performed. Purification of His$_6$-Smt3-tagged proteins was carried out analogously. The cells were resuspended in binding buffer (20 mM Tris-HCl pH 7.0, 300 mM NaCl, 10 mM or 20 mM imidazole, 2 mM β-mercaptoethanol) supplemented with 0.2 mM AEBSF-HCl. After cell lysis and centrifugation, the cleared supernatant was applied to an equilibrated 5 mL PureCube Ni$^{2+}$-NTA column and washed extensively with high salt washing buffer (20 mM Tris-HCl, 500 mM NaCl, 10 mM or 20 mM imidazole, 2 mM β-mercaptoethanol). The fusion proteins were eluted by applying a gradient of 10 mM or 20–500 mM imidazole in binding buffer. Fractions containing fusion protein were pooled and dialysed overnight against binding buffer without imidazole. The expression tag was cleaved overnight by adding His$_6$-tagged HsSENP1 (415–644) to the pooled elution fraction before dialysis. The mixture was reapplied to an equilibrated 5 mL PureCube Ni$^{2+}$-NTA column to remove liberated His$_6$-Smt3 and HsSENP1 before SEC.

### Preparative size-exclusion chromatography (SEC)

The flow-through of the second round of affinity chromatography containing the mature target protein was concentrated by ultrafiltration using Amicon 10 kDa MWCO ultra-centrifugal units (Merck KgaA, Darmstadt DE, cat no. UFC9010/UFC8010) and applied to an equilibrated HiLoad 16/600 Superdex 75 pg or 200 pg column (Cytiva Germany GmbH, Freiburg DE). Prior to each run the column was equilibrated with SEC buffer (25 mM Tris-HCl pH 7.5, 150 mM NaCl, 2 mM β-mercaptoethanol). Elution fractions containing pure protein were pooled and concentrated before protein concentration was determined by measuring absorption at 280 nm using a spectrophotometer DS-11 FX (DeNovix, Wilmington USA). ExPASy ProtParam was used to obtain extinction coefficients and molecular weights of each purified protein construct[109].

### Expression and purification of site-specifically acetylated proteins

The site-specific incorporation of N-(ε)-acetyl-L-lysine was conducted by addition of 10 mM N-(ε)-acetyl-L-lysine (CHEM-IMPEX INT´L LNC, cat no. 05364) and 20 mM nicotinamide (Sigma Aldrich, cat no. 72340) to block the *E. coli* deacetylase CobB to the *E. coli* BL21 (DE3) culture at an OD$_{600}$ of 0.6 (37 °C). Protein expression was induced by addition of 200 μM IPTG following a cultivation of additional 30 min (160 rpm, 37 °C). Importantly, incorporation of N-(ε)-acetyl-L-lysine was done on the background of the C256A mutant, as wild type SnCE1 was observed underlying autoacetylation. The incorporation of N-(ε)-acetyl-L-lysine in *E. coli* is done co-translationally as a response to an amber stop codon (UAG). All site-specifically acetylated GST-tagged SnCE1 proteins were affinity purified and their oligomerisation state was analyzed by analytical SEC as described above. The successful incorporation of acetyl-L-lysine was checked by immunoblotting of SEC elution fractions.

### Analytical size-exclusion chromatography (SEC)

In order to analyze the ability of SnCE1 and variants thereof to form higher oligomers, analytical SEC was performed using a calibrated Superdex 200 10/300 GL column (Cytiva Germany GmbH, Freiburg DE) (Supplementary Figs. 1, 14 and 18). Prior to the analytical SEC runs the column was equilibrated with SEC buffer (25 mM Tris-HCl pH 7.5,

150 mM NaCl, 2 mM β-mercaptoethanol) and protein concentrations adjusted to 1.0–2.0 mg/mL in same buffer in a final applied volume of 100 μL. By following the absorption at 280 nm protein containing fractions were identified and verified by SDS-PAGE and anti-SnCE1 immunoblotting using an anti-SnCE1 antibody (Davids Biotechnologie GmbH, Regensburg DE, dilution 1:50) and secondary HRP-coupled anti-rabbit IgG antibody (abcam, cat no. ab6721, dilution 1:10,000). Additionally, fractions were immunoblotted using an anti-acetyl-lysine antibody (abcam, ab21623, dilution 1:5000) in combination with an HRP-coupled anti-rabbit IgG antibody (abcam, cat no. ab6721, dilution 1:5000) to check the acetylation status of each SnCE1 variant/mutant. Calculation of the proteins' molecular weight and their respective oligomeric state was done by using a calibration curve as described before (Supplementary Fig. 2c). In short, a commercially available calibration kit for low and high molecular weight (MW) proteins (Cytiva Germany GmbH, Freiburg DE) was used to calculate elution volumes ($V_e$) of respective standard proteins. The columns' void volume ($V_O$) was determined by the elution volume of blue dextran dye and the geometrical column volume ($V_c$) calculated by the column dimensions. By using the following equation, partition coefficients ($K_{av}$) were calculated: $K_{av} = (V_e - V_O)/(V_c - V_O)$. The $K_{av}$ values were plotted as a function of the logMW and fitted using a linear equation. The resulting calibration equations are shown with the coefficient of determination $R^2$, showing the accuracy of the fit (Supplementary Fig. 2c).

## Immunoblotting

Following SDS-PAGE, gels were covered with transfer buffer to prepare them for protein transfer onto a 0.2 μm PVDF membrane (SERVA Electrophoresis GmbH, cat no. 42515.01). PVDF membrane was methanol-activated and soaked together with blotting paper in transfer buffer (25 mM Tris base, 150 mM glycine, 10% (v/v) methanol). Protein transfer was achieved by electrotransfer at 20 V, 140 mA for 1 h 15 min using a semi-dry blotter unit V20-SDB (SCIE-PLAS, Cambridge UK, serial no. 30039058) coupled to a power supply EV2310 (Consort bvba, B-2300 Turnhout Belgium). Transfer efficiency was checked afterwards by Ponceau S-red staining (Carl Roth GmbH, cat no. 5938.2). PVDF membranes were blocked in 5% (w/v) milk powder in PBST (140 mM NaCl, 2.7 mM KCl, 10.1 mM Na$_2$HPO4, 1.8 mM KH$_2$PO4, 0.1% (v/v) Tween-20) or TBST (150 mM NaCl, 20 mM Tris/HCl, pH 7.5, 0.1 % (v/v) Tween-20) for 1 h at room temperature. The respective primary antibody was diluted in blocking solution according to the manufacturer and incubated with the membrane on a rocking shaker over night at 4 °C. After washing the membrane three times 5–10 min with PBST or TBST the corresponding HRP-coupled secondary antibody was diluted in blocking solution and incubated with the membrane for 1 h at room temperature. Following three washes for 5–10 min with PBST or TBST signals were read out using enhanced chemiluminescence (ECL). For this, 1 mL of ECL solution A (100 mM Tris-HCl pH 8.5, 2.5 mM luminol, 0.4 mM p-coumaric acid) and solution B (100 mM Tris-HCl pH 8.5, 0.02% (v/v) H$_2$O$_2$) were mixed in a 1:1 ratio and evenly applied to the membrane. Signal development was carried out by using a chemiluminescence imager Octoplus QPLEX (Intas/NHDyeagnostics, Halle DE). Uncropped and unprocessed scans of the blots are provided in the Source Data file. For reprobing membranes were stripped according to a commercial mild membrane stripping protocol for Western blots (abcam, Cambridge UK) and further treated as described above. A complete list of all used antibodies and their respective dilutions is found in Supplementary Table 5.

## Acetylation assay

Intrinsic acetyltransferase activity of SnCE1 was determined by SnCE1 autoacetylation. For this, 10 μM wild type SnCE1 and inactive mutant C256A were incubated with 0.05 mM and 1 mM of acetyl group donor acetyl-CoA (Cayman Chemicals, cat no. 16160) and incubated for 1 h at 37 °C. The potential reversibility of SnCE1 acetylation was studied by adding 10 μM of E. coli GST-tagged sirtuin deacetylase CobB and 5 mM of its cofactor NAD$^+$ (Sigma Aldrich, cat no. N7004) to the reaction. To confirm enzymatic removal of acetylation by EcCobB, 10 mM of nicotinamide (Sigma Aldrich, cat no. 72340) were added to inhibit EcCobB. All acetylation reactions were conducted in 25 mM Tris-HCl pH 8.0, 150 mM NaCl, 2 mM β-mercaptoethanol. The acetylation reaction was stopped by adding 3× SDS sample buffer and boiling at 95 °C for 10 min. Samples were separated by SDS-PAGE and subjected to immunoblotting using a primary anti-acetyl-lysine antibody (abcam, ab21623, dilution 1:10,000) in combination with a secondary HRP-coupled anti-rabbit IgG antibody (abcam, ab6721, dilution: 1:10,000). The small GTP-binding nuclear protein Ran and its site-specifically acetylated variant Ran_AcK71 were used as negative and positive controls, respectively.

## Deacetylation of SnCE1 by human deacetylases

To investigate the reversibility of SnCE1 acetylation, SnCE1 was incubated with a panel of human deacetylases. For the deacetylation reaction 2–5 μM SnCE1 were incubated with either 0.1–0.2 μM HDAC1/2/3/4/6/7/8/9/11 or 1–2 μM SIRT1-7 and 10–20 mM NAD$^+$. To confirm enzymatic deacetylation by sirtuins, 50 mM nicotinamide was added. All deacetylation reactions were conducted in 25 mM Tris-HCl pH 7.5, 150 mM NaCl, 2 mM β-mercaptoethanol. The reaction was quenched by adding 3x SDS sample buffer and samples were denatured for 10 min at 95 °C, before being separated by SDS-PAGE and subjected to Western blot with subsequent immunoblotting using primary anti-acetyl-lysine (AcK) antibody (abcam, cat no. ab21623, dilution 1:10,000) and secondary HRP-coupled anti-rabbit IgG antibody (abcam, cat no. ab6721, dilution 1:5000).

## Cleavage of di-ubiquitin chains

Cleavage of di-ubiquitin chains of different isopeptide linkages (UbiQ Bio, Amsterdam NL, cat no. UbiQ-L01) was performed as described before[30,57]. In short, 5 μM SnCE1 was pipetted together with 25 μM di-ubiquitin and incubated overnight and the chain cleavage was monitored by SDS-PAGE. Prior incubation concentrations of di-ubiquitin chain and SnCE1 were adjusted to a twofold concentration in 25 mM Tris-HCl pH 7.5, 150 mM NaCl, 10 mM DTT and left at room temperature for 15 min to allow thiols to fully reduce. SnCE1 and di-Ub-chains were mixed 1:1 and incubated overnight at room temperature. For time-resolved cleavage of K63-linked di-ubiquitin the reaction was stopped at indicated time points. The reaction was quenched by adding 3xSDS sample buffer and boiling at 95 °C for 10 min, before monitoring di-ubiquitin cleavage by SDS-PAGE with CBB-staining.

## Cleavage of poly-SUMO2 chains

Cleavage of authentically linked poly-SUMO2$_{2-6}$ chains (UBP Bio, cat. no D9110) was performed as described before[30]. In short, 3 μg poly-SUMO2 (2 μg/μL) were incubated with 1 μM SnCE1 at room temperature for indicated time points. Prior incubation SnCE1 was prediluted in 25 mM Tris-HCl pH 7.5, 150 mM NaCl, 10 mM DTT and thiols allowed to fully reduce for 15 min at room temperature. The reaction was stopped by adding 3x SDS sample buffer and boiling at 95 °C for 10 min before the samples were separated by SDS-PAGE and chain cleavage was analyzed by immunoblotting using a primary anti-SUMO2 antibody (Thermo, cat no. MA5-37627, dilution 1:2000) and secondary HRP-coupled anti-mouse IgG antibody (abcam, ab6728, dilution 1:2000). Additionally, after mild stripping the membrane was stained with a primary anti-SnCE1 antibody (Davids Biotechnologie GmbH, Regensburg DE, dilution 1:50) and a secondary HRP-coupled anti-rabbit IgG antibody (abcam, cat no. ab6721, dilution 1:10,000).

## deSUMOylation of human RanGAP1-SUMO1

To demonstrate the ability of SnCE1 to cleave SUMO directly from protein substrates and not only poly-SUMO conjugates, 48 nM HsRanGAP1-SUMO1 was incubated with 144 nM wild type SnCE1 or the inactive mutant SnCE1 C256A at room temperature as described before[110]. As control, the dedicated human deSUMOylase SENP1 in wild type and inactive form (C603A) was used. The deSUMOylation reaction was stopped after indicated time points by adding 3x SDS sample buffer and boiling the samples at 95 °C for 10 min. Samples were separated by SDS-PAGE and subjected to immunoblotting using a primary anti-RanGAP1 antibody (dilution 1:1000) in combination with an HRP-coupled secondary anti-goat IgG antibody (abcam, ab6728, dilution 1:2000).

## Preparation of activity-based SUMO1-/SUMO3-/Ub-propargylamide (PA) probes

Generation of activity-based probes was achieved by using an intein-mediated semi-synthetic approach as described before[111,112]. In short, ubiquitin, SUMO1 and SUMO3 were expressed in E. coli BL21 (DE3) as fusion proteins carrying a C-terminal intein-chitin binding domain (intein-CBD) and affinity purified in binding buffer (20 mM HEPES, Na-acetate pH 6.5, 75 mM NaCl) on a chitin matrix gravity flow column (New England Biolabs, Frankfurt/Main DE, cat. no S6651L). After extensive washing with high salt washing buffer (20 mM HEPES, sodium acetate pH 6.5, 500 mM NaCl) on-column cleavage was induced by adding 100 mM MesNa (Sigma Aldrich GmbH, Taufkirchen DE, cat no. 8.10595.0010) in binding buffer and incubating for 24–36 h at 25 °C on a rotating shaker. Following washing with binding buffer, the eluted MesNa-functionalized probes were concentrated by ultra-filtration to a volume of 1–2 mL and applied to an equilibrated HiLoad 16/600 Superdex 75 pg column (Cytiva Germany GmbH, Freiburg DE). Elution fractions containing Ub/Ubl-MesNa were analyzed by SDS-PAGE and afterwards pooled and concentrated using Amicon 3k MWCO ultracentrifugal units (Merck KgaA, Darmstadt DE, cat. no. UFC9003). Synthesis of propargylated probes was achieved by dissolving 0.7 g of propargylamine (PA) hydrochloride (Sigma Aldrich GmbH, Taufkirchen DE, cat. no. P50919) in binding buffer and adjusting the pH to 8.0–8.5 using NaOH. The solution was incubated with 300–500 μM Ub/Ubl-MesNa for 3 h at room temperature before a second round of SEC was performed to remove unreacted propargylamine. The PA-probes were finally concentrated and flash-frozen and stored at −80 °C.

## Crystallization and data collection

For crystallization of catalytically inactive SnCE1 C256A and SnCE1 complexed with HsSUMO1-PA the construct 74–310 was used. wild type SnCE1 was crystallized using a further truncated construct 104–310 as the N-terminus was predicted by AlphaFold2 to be unstructured and therefore flexible. All proteins used for crystallization were stored in 25 mM Tris-HCl pH 7.5, 150 mM NaCl, 2 mM β-mercaptoethanol. Prior to crystallization SnCE1 C256A and wild type SnCE1 were concentrated by ultrafiltration to a concentration of 9.8 mg/mL, respectively. The covalent SnCE1-SUMO1-PA complex was prepared by incubating SnCE1 overnight with SUMO1-PA at room temperature. To improve crystallization, an N-terminally truncated construct of SUMO1 (20-96) was used. Stable complex formation was checked by SDS-PAGE and the complex was separated from unreacted SnCE1 and SUMO1-PA by performing SEC on an equilibrated HiLoad 16/600 Superdex 75 pg column (Cytiva Germany GmbH, Freiburg DE). Elution fractions containing SnCE1-SUMO1-PA with highest purity were pooled and concentrated to 3.6 mg/mL. Crystallization was carried out in 96-well CrystalQuick LP plates (Greiner Bio-one, Frickenhausen DE, cat. no. 609180) using the sitting drop vapor diffusion technique with commercially available crystallization screens. By using the CyBi-HTCP robotic platform (CyBio AG, Jena DE), 0.3 μL protein and reservoir

were mixed in a 1:1 ratio and the drops sealed with EasySeal adhesive foil (Greiner Bio-one, Frickenhausen DE). Plates were stored at 20 °C and regularly checked for crystal formation. Crystals of all constructs appeared in various conditions for which in the following only the conditions resulting in best-diffracting crystals is shown. wild type SnCE1 crystallized in 0.2 M Na-malonate pH 5.0, 20% (w/v) PEG3350 (PEG/Ion 2 Screen, Hampton Research, Cologne DE, cat no. HR2-098) and crystals were upscaled to a 24-well format (Greiner Bio-one, Frickenhausen DE, cat no. 662160) and further optimized, leading to improved crystallization in 0.18 M Na-malonate pH 5.0, 20% (w/v) PEG3350. For upscaled crystallization protein and reservoir were mixed in ratios of 1:1, 1:2 and 2:1 with volumes of 1 μL and 2 μL. SnCE1 C256A crystallized in 0.6 M NaCl, 0.1 M MES pH 6.5, 20% (w/v) PEG4000 (JBScreen Classic 3, JenaBioscience, Jena DE, cat. no. CS-103L). SnCE1 complexed with SUMO1-PA crystallized in 0.2 M Ca-acetate, 20% PEG3350 (PEG/Ion Screen, Hampton Research, Cologne DE, cat. no. HR2-126). Notably, formation of SnCE1-SUMO1-PA crystals were observed the fastest, appearing within a week compared to apo SnCE1 crystals which took weeks to months to grow. Crystals of apo wild type SnCE1 and C256A were cryoprotected using their respective crystallization condition supplemented with D-glucose (Carl Roth GmbH, cat. no. 6780.1) until saturation. Crystals of SnCE1-SUMO1-PA were cryoprotected in 0.2 M Ca-acetate, 15% PEG3350, 10% PEG400. Cryoprotected crystals were collected in loops of size 0.1–0.2 mm, flash-frozen in liquid nitrogen and stored until data collection. Diffraction data for apo SnCE1 C256A and wild type were collected at Beamline P13, DESY Hamburg at a wavelength of 0.8856 Å and 0.9763 Å, respectively. Diffraction data for complex of SnCE1-SUMO1-PA were collected at Beamline 14.1, BESSY Berlin at a wavelength of 0.9184 Å. All datasets were collected from single crystals.

## Structure solution and refinement

Collected data sets for all structures obtained were processed using XDS package[113,114]. All further procedures were carried out within the CCP4 program suite (v7.1.0.18 or v8.0.0.19)[115–117]. All SnCE1 structures were determined by molecular replacement (MR) employing the PHASER Basic MR pipeline[118]. As a search template for MR an Alpha-Fold2 model of the SnCE1 catalytic domain was used for solving SnCE1 C256A structure since this was the first one obtained. The refined monomeric model of SnCE1 C256A was used as a search template for determining both wild type structures. The number of SnCE1 molecules used for each structure solution was chosen based on predicted Matthews coefficients. Eight molecules of SnCE1 C256A were modeled in the asymmetric unit arranged as two tetramers. The asymmetric units of apo wild type SnCE1 and SnCE1-SUMO1-PA were analyzed to contain either three molecules or one molecule of SnCE1, respectively. The crystal structure of SUMO1•Daxx (PDB-ID: 4WJQ) was used for modeling the SnCE1-SUMO1-PA as a 1:1 heterodimer. Model building and inspection of electron density maps was done in COOT (v0.9.6)[119]. The structures were subsequently refined at 2.82 Å (apo wild type SnCE1), 2.21 Å (apo SnCE1 C256A) and 1.55 Å (SnCE1-SUMO1-PA) in REFMAC5 including TLS refinement[120]. Additionally, for both apo structures local non-crystallographic symmetry was included in the refinement process. The quality of the protein models was validated using MolProbity[121]. Complete data collection and refinement statistics are summarized in Supplementary Table 1. All diffraction data and structural models are deposited in the Protein Data Bank (PDB) with accession codes as stated in the "Data availability" section.

## Protein digestion and LC-MS/MS analysis

Protein samples (SnCE1$_{74-310}$, SnCE1$_{74-310}$ deacetylated by SIRT1, SnCE1$_{74-310}$ Y212A; three technical replicates each; $n$ = 3) underwent enzymatic digestion with sequencing-grade trypsin/lys-C mix (Promega) following the Single-Pot Solid-Phase-enhanced Sample Preparation (SP3) protocol, adapted from Blankenburg et al.[122]. A total of

1 µg protein was digested using a protein-to-enzyme ratio of 1:25 (w/w) at 37 °C overnight (16 h). Peptide samples were analyzed using data-dependent acquisition (DDA) mode on a Q Exactive Plus mass spectrometer (Thermo Fisher Scientific) coupled to an Ultimate3000 nano-LC system (Thermo Fisher Scientific; details in Supplementary Tables 6 and 7). Acetylated peptides were identified and quantified using Spectromine™ software (version 4.4. Biognosys AG), implementing a false discovery rate (FDR) of 1% at both peptide and protein level. The search database comprised protein sequences of the target protein, all used enzymes, and the *E. coli* BL21-DE3 proteome (UniprotSwissprot_2018) as background. Tryptic peptides (cleavage at R,K) with a minimum length of 7AA, a maximum of two missed cleavages, oxidation (M), and acetylation (CKHSTY) (Unimod #1) as variable modifications were considered. The minimal localization threshold for PTMs was set to 0.75. The peptide precursor intensities were exported for downstream analysis using R (version 4.5.0)[123]. Intensities of peptide precursors with different charge states were normalised using mean adjustment, filtered for acetylation status and target protein. Statistical significance between the different samples was determined using unpaired Student's *t*-tests with *p*-values < 0.05 considered significant. The analysis was done using the following packages: tidyverse[124], ggplot2[125], ggsignif[126].

### Intact protein mass determination by mass spectrometry

Samples were analyzed on an Orbitrap Eclipse mass spectrometer coupled to an UltiMate 3000 nHPLC system (both Thermo Scientific) operating in direct injection mode. Samples were loaded onto an in-house packed column (10 cm length, 100 µm inner diameter, filled with MapPak material, Thermo Scientific). Proteins were separated using a 35 min gradient running 0.1% formic acid (eluent A) against 80% acetonitrile with 0.1% formic acid (eluent B) at a constant flow of 600 nl/min. The gradient started at 10% B with linear increase to 60% B in 18 min, followed by an increase to 95% B in 5 min, followed by column washing and equilibration to initial conditions. The mass spectrometer was operated in intact protein mode with RF lens at 60% using either the Orbitrap detector at 7.5k resolution and 11 ms maximum injection time, or the ion trap detector using Enhanced scan rate.

### Cell culture to assess SnCE1 proteolytic processing

Human embryonic kidney cell line (HEK-293) obtained from ATCC (CRL-1573) was used. Cells were cultivated in RPMI medium supplemented with 10% FCS and 1% penicillin/streptomycin and incubated at 37 °C and 5% CO$_2$ conditions. For transfection pcDNA™3.1/MycHis B, pcDNA™3.1/MycHis B_SnCE1_WT, pcDNA™3.1/MycHis B_SnCE1_trunc, pcDNA™3.1/MycHis B_SnCE1_C256A and pcDNA™3.1/MycHis B_SnCE1_Y212 were amplified using *E. coli* XL1-blue using heat shock transformation followed by mini prep using peqGOLD plasmid Mini-Prep kit (VWR International) for the confirmation of the transformation using restriction enzyme digestion (*Sma*I and *Hin*dIII) of the plasmid followed by agarose gel electrophoresis. A Midi prep was carried out using NucleoBond®Xtra Midi kit (MACHEREY-NAGEL GmbH & Co. KG, Düren DE). Plasmid concentration was determined using NanoDrop for the transfection. Cells were plated in 6 cm plates (600,000 cells/plate). 24 h later, transfection with pcDNA™3.1/MycHis B_SnCE1_WT and pcDNA™3.1/MycHis B_SnCE1_Y212 was achieved using JetPRIME® Reagent (Polyplus, Illkirch), pcDNA™3.1/MycHis B was used as a control (Mock cells). 1,2 µg DNA was used for the transfection and a ratio of 1:2 for DNA: transfection reagent. 48 h after transfection, cells were treated with a mixture of deacetylase inhibitors (10 mM nicotinamide, 2 µM SAHA, and 1 µM Trichostatin A). 6 h later cells were collected, and directly lysed using RIPA buffer (150 mM NaCl, 50 mM Tris-HCl [pH 8.0], 5 mM EDTA pH 8.0, 0.5% NP-40, 0.5% Triton X-100, 0.5% sodium deoxycholate, 0.1% SDS, 1× cOmplete Mini protease inhibitor cocktail (Roche), 1× PhosSTOP phosphatase inhibitor cocktail

(Roche), 10 µM MG132). For immunoblotting cells were lysed in RIPA buffer. Lysis was achieved using 4 cycles of freezing and thawing. Protein concentration was determined using Bradford method. Proteins were separated using SDS-PAGE. Transfer was carried out using semi-dry transfer method. After confirmation of protein transfer using Ponceau S red staining, membranes were blocked in RotiBlock® (Carl Roth GmbH & Co.KG) for 30 min. Membranes were then incubated with the corresponding primary antibodies (Supplementary Table 5). After overnight incubation, membranes were washed, incubated with the secondary antibody (Supplementary Table 5). After washing, signal was detected using Clarity™ western -ECL substrate (Bio-Rad Laboratories, Inc). Image-J was used for staining quantification and GraphPadPrism (version 8 or 9.5.1) was used for statistical analysis. The HEK-293 cells were regularly tested to exclude *Mycoplasma* contamination (abmR, cat. No. G238/Mycoplasma PCR Detection Kit.

### Transfection of cells, Immunofluorescence microscopy and Subcellular fractionation of transfected cells

For subcellular fractionation, HEK293T cells originating from ATCC (CRL-3216) were transfected using Ca-phosphate precipitation method. For this, $12 \times 10^6$ cells were plated on 150 mm cell culture dishes in DMEM medium and on the next day, chloroquine was added to final concentration of 2.5 mM. 40–50 µg of plasmid DNA was mixed with 200 µl of 2.5 M CaCl$_2$ and 1.7 ml H$_2$O, after which 2 ml of 2xHBS buffer (50 mM HEPES pH 7.05, 140 mM NaCl, 1.5 mM Na$_2$HPO$_4$×2H$_2$O) was added dropwise while vortexing or bubbling the solution to enable formation of precipitates. After 15 min incubation, the DNA mixture was added to HEK293T cells, and left to stand overnight, with medium exchange on the following day. 36 h post transfection, cells were harvested and resuspended in mitochondria isolation buffer (20 mM HEPES pH 7.6, 220 mM mannitol, 70 mM sucrose, 1 mM EDTA, 0.5 mM PMSF). Cells were homogenized and the fractions were isolated by differential centrifugation. 10 min at $10,000 \times g$ was used to isolate crude mitochondrial fraction, whereas 1 h at $100,000 \times g$ was used to isolate the light membrane fraction. The remaining cytosolic fraction was precipitated using trichloroacetic acid.

For immunofluorescence microscopy, HeLa 299 cell lines originating from ATCC collection (CCL-2.1) stably expressing mitochondria-targeted green fluorescent protein (mitoGFP) or expressing ER-targeted dsRED (KDEL-dsRED) were transfected using PEI (Polyethylenimine hydrochloride, linear, Polyscience Inc, Pennsylvania, USA). For this, 1–2 µg of plasmid DNA was mixed with serum-free medium and 4 µl of 1 mg/ml PEI solution. The mixture was added to cells in serum-free medium. After overnight incubation, medium was exchanged for RPMI supplemented with 10% FCS. 36 h post transfection, cells were fixed and decorated with antibodies directed against the His$_6$ and with DAPI according to the already published protocol, followed by microscopy analysis using Leica Stellaris 5 confocal microscope[127]. For the analysis of mitochondrial morphology, at least 50 transfected and non-transfected cells with intact nuclei and no visual apoptotic characteristics were imaged using Leica LAS X Navigator and categorized according to the fragmented or normal mitochondrial phenotype. The cell lines were regularly tested to exclude *Mycoplasma* contamination by PCR using following primers: GPO-1: ACTCCTACGGGAGGCAGCAGTA and MGSO: TGCACCATCTGTCACTCTGTTAACCTC

### Sequence- and structure alignments

All protein sequences used for alignments were obtained from UniProt database[128,129]. The sequence alignments were performed with the MSA program Clustal Omega (v0(1.2.4)) and further analyzed using the software ESPript3.0 (v3.0.10)[130,131]. As structure models either obtained crystal structures or AlphaFold3 models were used[59]. For structural alignments, if possible, crystal structures otherwise AlphaFold3 models were used. The alignment was performed using PyMOL[116,117].

## Data analysis and visualization

Fiji (ImageJ 2.0.0-rc-68/1.52h) was used for quantitative analysis of the immunoblots[132]. Raw data from most experiments was processed using Microsoft Excel 2011. Data was visualized and statistically analyzed in GraphPad Prism version 8 or version 9.5.1. Fitting of data was also performed in GraphPad Prism version 8 or version 9.5.1. SnapGene Viewer 5.1.4.1 was employed for DNA sequence handling and generation of plasmid maps (SnapGene software from Insightful Science; available at snapgene.com). PyMOL version 2.5.3 and PyMOL version 2.3.4 were used to generate visual representations of protein structures[133]. ChemDraw version 23.0.1 was used to draw chemical structures. Adobe Photoshop 22.3.1 and Adobe Illustrator 25.4.1 were used to create figures.

## Statistics and reproducibility

(De-)acetylation assays as well as SUMO2/ubiquitin cleavage assays were performed in at least two independent experiments as indicated with similar results obtained. Analytic SEC and subsequent SDS-PAGE and immunoblotting of non-site-specifically acetylated SnCE1 proteins was done in triplicate with showing one example of respective elution fractions. For x-ray crystallographic structure determination of SnCE1, data sets were collected from single crystals without merging of data sets. All assays were performed in independent technical or biological replicates as indicated resulting in similar results. For bar graphs, the standard deviations (SD) and mean values were depicted. No statistical method was used to predetermine sample size. Unpaired, two-tailed student's $t$-tests were performed to assess statistical significance with significance levels as indicated.

## Reporting summary

Further information on research design is available in the Nature Portfolio Reporting Summary linked to this article.

# Data availability

The coordinates and structure factors for the structures of wild type SnCE1 (PDB: 9QTE), SnCE1 C256A (PDB: 9QTF) and SnCE1-SUMO1-PA (PDB: 9QTG) were deposited in the PDB (http://www.rcsb.org). The structures of *Chlamydia trachomatis* ChlaDUB1 in complexes with ubiquitin (PDB: 6GZS) or Coenzyme A (PDB: 6GZT, *Hs*SIRT1•NAD$^+$•EX527 (PDB: 4I5I), *Hs*SIRT2•NAD$^+$•TNFn-5 (PDB: 8QT3), *Hs*SIRT3•NAD$^+$•EX527 (PDB: 4BV3), *Ec*CobB•HsH4 peptide (PDB: 1S5P) and SUMO1•Daxx (PDB-ID: 4WJQ) are deposited in the PDB. The mass spectrometry data is stored in the data repository MassIVE [https://massive.ucsd.edu/ProteoSAFe/dataset.jsp?accession=MSV000098143]. Source data are provided with this paper.

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

## Acknowledgements

We thank the group of Prof. Frauke Melchior and Dr. Annette Flotho, Heidelberg University, for sending recombinant RanGAP1-SUMO1 and a human RanGAP1 antibody. We thank Dr. Thomas Hermanns, University Cologne, for providing plasmids encoding for SnCE1-5, SENP1, SENP6 and of Ub/SUMO1/SUMO3 activity-based probes. We thank Felix Glinka for providing the plasmid encoding yeast Ulp1. We thank Kathrin Krassow for expert secretary assistance. This work was supported by the Deutsche Forschungsgemeinschaft (DFG, German Research Foundation)-grant No. 443535983 (Research Training Group 2719; RTG 2719; M.L.), Deutsche Forschungsgemeinschaft (DFG, German Research Foundation) grant No. 445120888 (INST 292/156-1 FUGG; M.L.) and grant No. 441529220 (INST 292/154-1 FUGG; M.L.). Support for the publication fee was provided by University of Greifswald's publication fund. We thank HZB/BESSY in Berlin and EMBL/DESY in Hamburg for continuous support in X-ray data collection.

## Author contributions

O.S. did expression, purification and crystallization of proteins and performed most biochemical experiments. B.G. helped in expression, purification and crystallization of proteins and performed molecular cloning and site-directed mutagenesis. L.-M.M., L.Sp. and V.K.-P. performed infection experiments, microscopy and subcellular fractionation experiments. O.S., J.H., M.L., S.S., G.P. collected X-ray data, O.S. and G.P. solved the structures. S.S., L.B., N.K., X.S. performed biochemical experiments. J.H. did technical support. M.D., K.H., U.T.B., S.K. performed structure predictions. R. A.-A., K.G., S.W., H.J., E.K. did cell culture experiments, L.St., K.S., C.H., U.V. performed and analyzed mass spectrometry experiments, J.-W.L., S.M., M.K. performed intact mass determination and analysed data, V.L., K.H. delivered constructs and discussed data, M.L. initiated, designed and supervised the study. O.S. and M.L. wrote the manuscript. All authors contributed to data analysis and gave comments on writing the manuscript.

## Funding

## Competing interests

The authors declare no competing interests.
