## [Transparent Peer Review file · Nature Communications]

Reprogramming of bacterial virulence by lysine acetylation

Corresponding Author: Professor Michael Lammers

Version 0:

Reviewer comments:

Reviewer #1

(Remarks to the Author)

In this manuscript Schmöker et al., functionally and structurally characterize a new regulatory mechanism of the bacterial pathogen *Simkania negevensis*. *Simkania negevensis* is a respiratory pathogen unrelated but with a similar lifestyle as *Legionella pneumophila*, developing a high number DUBs and ULPs to allow infection of the eukaryotic hosts. In particular the manuscript characterizes the deSUMOylating and deubiquitinating activity of the SnCE1, a member of the CE-clan protease family, and interestingly reveals a novel autoacetylation mechanism that regulates its activity inside the host cell. Interestingly SnCE1, oligomerization (tetramer) inhibits deSUMOylation, while the monomeric form is active. An irreversible autoacetylation shifts the equilibrium towards the monomeric state, with SUMO1/2 chain deSUMOylation activity.

The manuscript contains a massive amount of data: including three crystal structures (monomer, tetramer and in complex with SUMO1), as well as Mass Spectrometry KAct analysis; mutagenesis and biochemical characterization; recombinant acetylated expression variants; cell biology localization. I must say that in my opinion the manuscript is a little bit too long, particularly the introduction, results and discussion sections (I would recommend to shorten these sections), but the outcomes are interesting, particularly the autoacetylation regulatory part. The authors nicely describe the mechanism for the activity regulation of SnCE1. A dual SUMO/ubiquitin protease with acetyltransferase activity was initially reported within the CE-clan related ChlaDUB1. The authors discover the acetylated lysine that might be responsible for the activity regulation (tetramer-monomer shift), and other acetylated lysines, in particular at the unstructured N-terminal region, which they claim might be responsible for auto processing, and thus regulate the SnCE1 localization inside the host cell. Although the manuscript quality and novelty is sound, based on the results there are some important statements that should be either better argued or try to lower the tone not to overstate (in particular the autoproteolysis part).

General Issues

I think that it would be good if the authors reduce the length of the manuscript. There are some parts, at the beginning of the results, that could be removed. For example, regarding the catalysis of SnCE1, it is a canonical cysteine protease mechanism. It is very known, and similar to all cysteine proteases. Also, the description of the SnCE1 structure should be focused on the differences with other CE-clan members. Most of the described interactions are similar to all the CE-clan protease members (for example SUMO C-terminal tail binding). Only the VR regions should be considered in the main text to explain why it is more active for SUMO over ubiquitin, for example. However, Figure 2 is good to me, but the text is confusing with so much description.

The monomer-tetramer regulatory idea is great, in summary: monomer active over tetramer non active. But I can't find an experiment that checks deSUMOylase activity of the tetramer. First it would be good to prove that the WT shifts between these two states. Do WT forms a tetramer?, not by SEC but with other biophysical strategies ?? (probably it is autoacetylated and monomeric, as shown), and then conduct some activity tests with WT oligomer forms to show that monomer is more active than tetramer. For example, is the acetylated form more active than the monomer? Or prepare a K231 mutant, which would preclude tetramer formation, to compare to WT and check if it has more deSUMOylase activity. This is an important issue to understand the regulatory mechanism.

Regarding the proteolytic cleavage, it is also not so clear to me. In fig 5a the authors show the proteolysis of full-length SnCE1 by WB when it is not acetylated, with a shift of barely 1 or 2 KDa, 25KDa to 23 KDa (but FL shouldn't be 35 KDa ?, if I understood well). But in Fig 7a, in HEK cells, the proteolysis shifts from 35 to 27 KDa. The intensity levels (rel. band

intensity) in the gel do not correspond to the plot by eye (20% between non- and processed bands, it looks much lower), but probably I'm wrong. This figure is important to explain the autoproteolysis and the acetylation state, but I think that it should be clearer (intensity differences between WT and SnCE1 Y212A are very low).

In Fig 7c, again, is it the SnCE1 FL or a N-terminal truncation? WT is not processed, but C256A is processed, but only 1 or 2 KDa difference ?? Shouldn't you conduct this experiment with FL SnCE1 ?. Maybe I misunderstood something, but do you completely discard external proteolysis ? In particular in HEK cells with all cellular proteases ?. Proteolysis could depend on the acetylation state, but why autoproteolysis ?

Other issues.

Out of the five catalytic domain homologs of SnCE1 in *Simkania negevensis*, only the SnCE1 shows deconjugating and acetyltransferase activity. Have the authors compared the SnCE1 crystal structure with the AlphaFold models of SnCE2-5 ?. Is there any structural explanation for their lack of activity ?

In the crystal structure descriptions (line 207), it would be good to know the extension of the SnCE1 chain observed in the electron density maps. Particularly if there are missing regions (flexible to be observed in the crystal structure, but present in the protein).

The deSUMOylation activities in Fig1 do not seem to be very strong. Have the authors checked, a little bit more quantitatively, those activities, for example compared to other CE-clan members or with yeast or human SENPs.

Why do the authors check Ser208 in Fig2 ?

Since tetramerization impacts the deSUMOylation activity (Fig 3b), the WT the SEC profile is clearly shifted towards the monomer (in contrast to the C256A mutant), but I would expect an equilibrium. Any evidence that the WT can also form a tetramer ?

Also regarding the tetramer, in the presence of the complete N-terminal segment, is the tetramer formed ? Even with a transmembrane region ? Could the authors speculate whether the tetramer can be formed when attached to the ER membrane ?.

In Figure 4E, the different N-terminal constructs have different acetylation levels (in particular SnCE1 95-310). Any comments on that ?.

Perhaps one of the more intriguing questions is the autoprocessing activity. I know that it is already discussed, but DUBs or ULPs are highly specific for Ub/Ubl substrates. It is hard to understand autoprocessing using the catalytic triad in the active site. But the presence of a different active site, from an unstructured regions (N-terminal extension), are also very hard to imagine. A non-autoproteolytic activity is completely discarded ? Other hypothesis.

Reviewer #2

(Remarks to the Author)

The authors have performed significant amount of works in characterizing the functional significance of site-specific lysine acetylation on SnCE1. Since my background is chemical biology instead of bacteriology, I am not going to comment on the significance or novelty of the report. Nevertheless, I strongly supports works like this that performed in-depth investigation of site-specific PTM. Having said that, I do have some doubts regarding the data interpretation or recombinant protein characterization in the manuscript.

1. Line 190-193: We confirmed, although less efficiently reacting with the Ub-PA probe compared to the SUMO1/3-PA probes, SnCE1 shows weak DUB activity cleaving K11-, K48- and K63-linked di-ubiquitin (diUb) chains. Based on Fig 1a, I can't see the band corresponding to SnCE1-Ub, questioning the capability of Ub-PA to react with SnCE1. Also, for WT(-) lane of SUMO3-PA probe, there seems to be a band corresponding to SnCE1-SUMO3, which should not be possible. For Fig 1b, there seems to be a band corresponding to Ub for M1/M27/M29/K33 as well, indicating the capability of SnCE1 in cleaving these linkages. However, it would be helpful to provide a comparison of each dimer with and without SnCE1 for more objective analysis. At the moment, negative controls are missing (i.e., Ub dimers only without SnCE1).

2. Fig. 3a, the amount of C256A mutant is much less than the wt protein, consequently, the lack of AcK signals in IB may be due to lower concentration of the protein instead of the lack of catalytic activity.

3. C256A mutant seems to be able to cleave SUMO2 in Fig 5b but not in Fig 2b.

4. There is no intact mass data for all recombinant proteins (including those with AcK), consequently difficult to judge whether the expressed proteins have the expected amino acid sequence.

5. Writing and readability can be improved. At the moment, it seems to target very specialized audience.

Reviewer #3

(Remarks to the Author)

General Comments

In the manuscript NCOMMS-25-46095 "Reprogramming of bacterial virulence by lysine acetylation", the authors present a very comprehensive study to provide mechanistic insights on how lysine acetylation of virulence factors is used to reprogram virulence adjusting it to the host cells' metabolic state .

Overall, the manuscript is excellent in its presentation and experimental design. The introduction is very thorough in presenting previous work and current understanding of the field that then clearly address gaps in knowledge. Here the authors demonstrate that lysine acetylation enables the coordination of the activity of bacterial virulence within the host cells in precise space and time. They solve several crystal structures or the acetylated WT of SnCE 1, a non-acetylated mutant in the apo state and SnCE1 in complex with SUMO1.

Specific Comments

1. In the section Acetylation of SnCE1 at K231 is irreversible and mediates SnCE1 monomerization beginning on line 412 the authors refer the reader to Supplemental Figure 10. Can the authors confirm that the correct figure is being used – this figure only goes to aa 205 but the manuscript is discussing sites K209, K248 and beyond.

2. The authors state that there are regions in the SnCE1 protein that are difficult to detect by mass spectrometry because of "long peptides" after tryptic digest. A tryptic peptide of 20 aa and 21 aa is not considered to be a challenge to sequence by MS/MS and is still in the ideal range of amino acids in a peptide. I would remove that statement and just leave that you chose the sites for incorporated acetyl-lysine as just simply either previously identified by mass spec or present in functionally or structurally important regions in SnCE1.

Version 1:

Reviewer comments:

Reviewer #1

(Remarks to the Author)

I carefully reviewed the authors' response letter. The authors have correctly addressed and discussed all concerns raised by the reviewer. The reviewer fully agrees that this manuscript is suitable for publication in Nature Communications.

Reviewer #2

(Remarks to the Author)

The authors have responded to all of the comments, although the responses were not presented in a way that made it easy for me to verify exactly where and how the manuscript or Supporting Information had been amended. This is particularly troublesome for a manuscript describing such a comprehensive study with a large amount of data. The points I raised were not intended to make the revision process unnecessarily difficult for the authors. On the contrary, these were questions that arose naturally for me as a reader. Although the journal will publish the peer-review reports, it is unlikely that most readers will consult them while reading the manuscript. Therefore, the key purpose of the revision should be to clarify these issues directly in the manuscript rather than only in the response letter. Nevertheless, it would also be helpful if the authors could reproduce the relevant changes more explicitly in the response letter, so that I do not need to move back and forth repeatedly between the manuscript and the supplementary data to identify the exact revisions. While I have supported publication of the manuscript from the very beginning, these obstacles in the review process have greatly diminished my enthusiasm. Overall, a few minor issues remain, but I think the editor should be able to make the next decision without external review.

1. Okay.

2. The concern was straightforward: the C256A lane contained substantially less protein, so the absence of an AcK signal could simply reflect underloading. The authors do not provide a corrected blot with matched loading, nor do they present quantitative normalization demonstrating that the conclusion is robust. Instead, they argue that the antibody is sensitive and that other experiments elsewhere support loss of acetyltransferase activity. That is an indirect defence, not a direct response to the criticism. The clean solution would have been to repeat the experiment with equal loading and matched exposure.

3. Okay.

4. Okay, but the description "Presence of a reducing agent such as beta-mercaptoethanol or TCEP not containing thiol-groups was essential for SnCE1 catalytic deSUMOlyase activity" is ambiguous, because it appears to imply that beta-mercaptoethanol does not contain a thiol group.

5. The manuscript still contains many language and presentation problems. Some examples, although not a comprehensive list, are given below. Overall, the manuscript still requires thorough editing for grammar, spelling, and formatting.

For grammar, "Gram-negative bacteria use plethora of virulence factors" should be corrected to "Gram-negative bacteria use a plethora of virulence factors." "Treatment of cells with deacetylase inhibitors, i.e. SAHA and TSA, to inhibit classical HDACs and nicotinamide (NA) to inhibit sirtuins, show the acetylation of SnCE1 increases" should be corrected to

“Treatment of cells with deacetylase inhibitors, i.e. SAHA and TSA to inhibit classical HDACs and nicotinamide (NA) to inhibit sirtuins, shows that SnCE1 acetylation increases.” “Although, the mutant SnCE1 Y212A shows that the SnCE1 acetylation state directly affects autoproteolytic processing” should be corrected to “Although the mutant SnCE1 Y212A shows that the acetylation state of SnCE1 directly affects autoproteolytic processing.” “The fact, SnCE1 Y212A shows a stronger processing compared to the likewise monomeric wild type SnCE1” should be corrected to “The fact that SnCE1 Y212A shows stronger processing than the likewise monomeric wild-type SnCE1.” “different molecular strategies underly this dual AcT/ULP-activity” should be corrected to “different molecular strategies underlie this dual AcT/ULP activity.”

For typos, “whichstage” should be corrected to “which stage.”

For formatting, “SnCE1wild type” should be corrected to “SnCE1 wild type.”

Reviewer #3

(Remarks to the Author)

The authors have addressed all my comments and points of clarifications and have strengthened the manuscript. I have no new comments to add.

Point-by-point response to the reviewers' comments

Reply to all reviewers:

We thank the reviewers for carefully reading our manuscript and giving us these valuable and constructive suggestions. We integrated these suggestions into the revised manuscript from our point of view strongly improving the manuscript. We worked comprehensively on all open points. We worked on the language, shortened several sections as suggested by the reviewers and sub-divided some long paragraphs into suitable sections. Moreover, we performed additional experiments to resolve the open questions raised by the reviewers:

1. Ub-chain cleavage assays using diUb-chains of various linkage types as substrates (Fig. 1b)
2. Immunoblotting of different N-terminal truncation constructs of SnCE1 and of all mutants of SnCE1 applied in this study to be able to show and compare their acetylation states (Fig. 4e). These data show that the catalytic His-base, i.e. H190, is also important for the acetyltransferase (AcT) activity as described earlier for related virulence factors.
3. Addition of primary sequence alignments, AlphaFold3 models also in their complexes with CoA and structural superpositions thereof to be able to compare SnCE1-5 regarding their similarity, catalytic activity and specificity (Supplementary Fig. 16; Supplementary Data 1). This data gives explanations underlying the observation that SnCE1 is the only CE-clan virulence factor of the five (SnCE1-5) in *S. negevensis* that has an AcT activity and active as deSUMOylase.
4. Comparison of the SnCE1 deSUMOylase efficiency with known deSUMOylases (SENP1, SENP6, Ulp1) (Supplementary Fig. 3a). SnCE1 shows a similar, albeit slightly reduced, deSUMOylase activity compared to well-known deSUMOylases.
5. Determination of intact masses for all recombinantly expressed proteins used in this study (Supplementary Data 3,4). We could confirm the masses for the recombinant proteins. We observed adducts with the reducing agent b-mercaptoethanol due to presence of three reactive Cys side chains (the active site Cys and two Cys in the N-terminal and C-terminal region, respectively). These analyses were therefore complemented by assessing the deSUMOylase activity in presence of different reducing agents (b-mercaptoethanol and TCEP without thiols) and in absence of a reducing agent. These experiments show that presence of a reducing agent is needed for catalytic activity (Supplementary Fig. 3b).
6. Analyses concerning presence of proteases in the samples of recombinantly expressed SnCE1 to exclude that autocatalytic processing observed in the recombinant protein might be due in fact by presence of an exogenous protease (Supplementary Data 2). We do not identify any protease supporting our model of SnCE1-autoproteolytic processing.
7. Analysis of SnCE1 dimer existing in an equilibrium with the monomer by analytical size-exclusion chromatography (Supplementary Fig. 14). We pooled the tetramer fractions from a SEC run and re-applied it to a SEC column. We observed again a fraction eluting as monomer and a fraction eluting as tetramer showing the dynamic equilibrium.
8. Construction, expression and purification of two SnCE1 mutants, i.e. SnCE1 K231R and SnCE1 E217D K231R on the catalytically active wild type background, to experimentally validate the model that tetramer formation of SnCE1 interferes with its deSUMOylase activity (Supplementary Fig. 23 and Supplementary Fig. 24). The mutants did not elute as tetramer from analytical SEC, also not after deacetylation by SIRT1, suggesting that the mutations did slightly interfere with the conformation prohibiting tetramer formation. We still hypothesize that tetramer formation of SnCE1 impairs deSUMOylase activity as suggested by the presented structure of SnCE1 C256A.

Reviewer #1 (Remarks to the Author):

Point 1:

I think that it would be good if the authors reduce the length of the manuscript. There are some parts, at the beginning of the results, that could be removed. For example, regarding the catalysis of SnCE1, it is a canonical cysteine protease mechanism. It is very known, and similar to all cysteine proteases. Also, the description of the SnCE1 structure should be focused on the differences with other CE-clan members. Most of the described interactions are similar to all the CE-clan protease members (for example SUMO C-terminal tail binding). Only the VR regions should be considered in the main text to explain why it is more active for SUMO over ubiquitin, for example. However, Figure 2 is good to me, but the text is confusing with so much description.

Response:

We agree with the reviewer. We extensively worked on the manuscript and removed/shortened some parts and restructured some sections to enable a better readability, i.e. amongst others the explanation of the Cys-protease mechanism in the results section and the characterization of the YopJ-family and of ChlaDUB1 in the introduction. As suggested, we focus on the interactions including the variable regions in the main text. We moved Fig. 1g showing the catalytic mechanism exerted by SnCE1 to the Supplementary Information section (Supplementary Fig. 6). The sections that were removed were highlighted in yellow and are shown crossed out in the version of the manuscript showing track changes. We also substantially shortened the paragraph explaining the interactions of SnCE1 and SUMO1 and focused only on selected and important interactions for determination of specificity towards SUMO1. The more detailed description of the interactions was shifted to the Supplementary Information (Supplementary Information 11).

Point 2:

The monomer-tetramer regulatory idea is great, in summary: monomer active over tetramer non active. But I can't find an experiment that checks deSUMOylase activity of the tetramer. First it would be good to prove that the WT shifts between these two states. Do WT forms a tetramer?, not by SEC but with other biophysical strategies ?? (probably it is autoacetylated and monomeric, as shown), and then conduct some activity tests with WT oligomer forms to show that monomer is more active than tetramer. For example, is the acetylated form more active than the monomer? Or prepare a K231 mutant, which would preclude tetramer formation, to compare to WT and check if it has more deSUMOylase activity. This is an important issue to understand the regulatory mechanism.

Response:

The crystal structures of acetylated SnCE1 wildtype and unacetylated SnCE1 C256A support our findings showing that acetylated SnCE1 wild type forms a monomer while non-acetylated SnCE1 C256A forms a tetramer. The way the tetramer is formed structurally suggests that tetramer formation blocks access of SUMO to the S1' site. Moreover, also all variants of SnCE1 C256A, including the site-specifically acetylated proteins, elute as tetramers. There is only one exception, i.e. SnCE1 C256A AckK231. This clearly shows that acetylation at K231-shifts the equilibrium to the monomeric form. To validate whether the SnCE1 monomer exists in an equilibrium with a tetramer we therefore analysed the non-acetylated SnCE1 S208A C256A mutant. Running SnCE1 S208A C256A over the analytical SEC, pooling the fractions containing tetrameric SnCE1 and re-running those on analytical SEC shows that the tetramer exists indeed in an equilibrium with the monomer. We added this experiment as Supplementary Fig. 14. Notably, the pure deacetylation by treatment of SnCE1 wild type with the deacetylase SIRT1, although strongly reducing the overall acetylation level, still does not result in deacetylation of AckK231 as supported by immunoblotting and mass spectrometry. This means that even the deacetylated SnCE1 elutes as monomer, likely due to the fact that acetylation sites including AckK231 resist deacetylation by SIRT1 (Supp. Fig. 21c; Supp. Fig. 22a).

As suggested by the reviewer, to also experimentally validate the model that tetramer formation interferes with SnCE1 deSUMOylase activity, we aimed at creating mutants, which forms a tetramer but which is catalytically active. To this end, we mutated Lys231 to Arg thereby preventing acetylation at this position while preserving the positive charge. To exclude the possibility that exchange of Lys231 by the sterically more demanding Arg would impair formation of the salt bridge with Glu217, we also prepared the double mutant SnCE1 E217D K231R. These mutants were expressed and purified and analysed by analytical SEC and immunoblotting. Both mutants were lysine acetylated as judged by immunoblotting using an anti-acetyl-lysine antibody, however, the signal was strongly reduced compared to SnCE1 wild type (Fig. 4e; Supplementary Fig. 23, 24). Contradictory to our expectations, the SEC experiments show that both, SnCE1 K231R and SnCE1 E217D K231R, elute as pure monomers. The earlier results preparing site-specifically acetylated proteins show that K231-acetylation on the background of C256A is sufficient to stabilize the SnCE1 monomer. To exclude the possibility that other lysine acetylation sites would interfere with tetramer formation of SnCE1 K231R and SnCE1 E217D K231R we deacetylated the mutants by SIRT1 and analysed the deacetylated proteins by SEC. Both proteins were indeed deacetylated, however, they still elute as pure monomers (Supplementary Fig. 23,24). This firstly supports our data that indeed acetylation at K231 is essential to shift the monomer-tetramer equilibrium to the monomeric form. Secondly, these data furthermore show that both mutations are not capable to functionally replace K231 and E217 in SnCE1 wild type forming the salt bridge stabilizing the SnCE1 tetramer. Most likely, subtle differences in the conformation might be sufficient to abrogate tetramer formation. Along that line, both mutants were also less acetylated compared to SnCE1 wildtype suggesting that the mutations interfere with other properties such as binding of acetyl-CoA (Fig. 4e). As a summary, we were not able to construct a SnCE1 mutant that is catalytically active and

not acetylated at K231 still allowing proper formation of a tetramer. However, our experimental structures show that SUMO-binding to S1 is prohibited by tetramerization and that K231-acetylation stabilizes the SnCE1 monomer. This allows to draw the conclusion that tetramer formation interferes with SnCE1 deSUMOylase activity.

We wrote in the manuscript (lines 403-406):

“Notably, analytical SEC experiments reveal that catalytically dead, non-acetylated SnCE1 forms a tetramer in solution existing in an equilibrium with a monomer while acetylated SnCE1 wild type exclusively forms a monomer (Fig. 3b; Supplementary Fig. 1, 2b; Supplementary Fig. 14).”

and (lines 664-680):

“To test this hypothesis we aimed at constructing a catalytically active SnCE1 protein capable to form a tetramer. To this end, we designed two SnCE1 mutants, i.e. SnCE1 K231R and SnCE1 E217D K231R, which cannot be acetylated on K231 and should be capable to form a salt bridge shown to be essential for tetramer formation. We observed that both mutants eluted as pure monomer from analytical SEC (Supplementary Fig. 23). To exclude the possibility that other lysine acetylation sites in SnCE1 K231 or SnCE1 E217D K231R might prohibit tetramer formation we deacetylated SnCE1 K231R and SnCE1 E217D K231R by SIRT1 and performed analytical SEC. However, also these proteins only eluted as pure monomers (Supplementary Fig. 24). These data suggests that both mutants impair SnCE1 tetramer formation. As a consequence, we are not able to experimentally validate the impaired deSUMOylase activity due to formation of the SnCE1 tetramer. However, our structural data in combination with analytical SEC experiments both suggest that oligomerization of SnCE1 modulated by its acetylation state affects its deSUMOylase activity.”

Point 3:

Regarding the proteolytic cleavage, it is also not so clear to me. In fig 5a the authors show the proteolysis of full-length SnCE1 by WB when it is not acetylated, with a shift of barely 1 or 2 KDa, 25KDa to 23 KDa (but FL shouldn't be 35 KDa ?, if I understood well). But in Fig 7a, in HEK cells, the proteolysis shifts from 35 to 27 KDa. The intensity levels (rel. band intensity) in the gel do not correspond to the plot by eye (20% between non- and processed bands, it looks much lower), but probably I'm wrong. This figure is important to explain the autoproteolysis and the acetylation state, but I think that it should be clearer (intensity differences between WT and SnCE1 Y212A are very low). In Fig 7c, again, is it the SnCE1 FL or a N-terminal truncation? WT is not processed, but C256A is processed, but only 1 or 2 KDa difference ?? Shouldn't you conduct this experiment with FL SnCE1 ?. Maybe I misunderstood something, but do you completely discard external proteolysis ? In particular in HEK cells with all cellular proteases ?. Proteolysis could depend on the acetylation state, but why autoproteolysis ?

Response:

Fig. 5a showing the *in vitro* SUMO2-cleavage an N-terminally truncated SnCE1, i.e. SnCE1₇₄₋₃₁₀, is used and not full length. For all *in vitro* experiments the truncated SnCE1 construct SnCE1₇₄₋₃₁₀ was used if not indicated otherwise. Test expression showed that it is not possible to solubly express SnCE1 full length, including the N-terminal transmembrane region, in *E. coli*. We agree that the figure legend is misleading and we adjusted it to be more precise. In the figures we labelled all SnCE1 proteins and fragments thereof with „proc“ in the cell culture experiments indicating that SnCE1 is physiologically processed in the eukaryotic cells. For the recombinant protein we used the term „truncated“ if an N-terminally truncated protein was used. Moreover, we stated in the manuscript that for all experimnts we used the fragment. SnCE1₇₄₋₃₁₀, lacking the N-terminal transmembrane region, if not stated otherwise. But a difference in mass and the separation in two discrete fractions visible in the SDS-PAGE indicates proteolytic cleavage. As we used truncated SnCE1 in bacterial expressions, the mass shift is small, however, in the expression of full length SnCE1 in eukaryotic cells this would correspond to a cleavage to app. 10 kDa from the N-terminus.

Concerning the band intensities in Fig. 7a, we set the intensity of the signals obtained for the sum of unprocessed SnCE1 plus processed SnCE1 to 100% and determined the relative ratio of both. This is shown in the plot. The calculation resulted in these values. We show only one representative immunoblot in the main text but the others including our calculations are supplied in the Source Data file. Importantly, for wild type SnCE1 the processed fraction is always less than observed for the processed fraction of the SnCE1 Y212A mutant, supporting our statement that the mutant shows less processing indirectly indicating that the acetylation of SnCE1 affects proteolytic processing.

In Fig. 7c we used recombinantly expressed proteins, i.e. it is again the truncated protein SnCE1₇₄₋₃₁₀. For clarity, for all constructs we designed to lack a part from the N-terminus we used the wording “truncated” while a processing in eukaryotic cells also resulting in removing N-terminal residues is named as “processed” as it is processed in the cells. This is how we labelled the lanes in the blots and in the figure legends, i.e. SnCE1_{trunc} Or SnCE1_{proc}. We adjusted the figure legend to be more precise. We used the truncated SnCE1₇₄₋₃₁₀ (and further truncated versions, i.e. SnCE1₈₃₋₃₁₀, SnCE1₉₅₋₃₁₀ and SnCE1₁₀₄₋₃₁₀ for some experiments) for most *in vitro* experiments, as the presence of the hydrophobic transmembrane helix in the SnCE1 N-terminus renders the protein insoluble upon expression in *E. coli*. The results obtained with the recombinantly expressed truncated protein suggested a proteolytic processing to occur from the N-terminus with the cleavage site most likely present in the N-terminal region bewtween aa74-104 as the recombinant protein SnCE1₁₀₄₋₃₁₀ was not processed. To validate the

processing to occur in full length SnCE1 in eukaryotic cells, we initiated studies in HEK cells transiently expressing full length SnCE1 including the transmembrane domain.

Concerning the point raised by the reviewer regarding autoproteolysis versus external proteolysis, we conclude an autoproteolytic mechanism based on several observations. We observe proteolytic cleavage following deacetylation of WT SnCE1 which supports an autoproteolytic cleavage mechanism. Moreover, the expression of SnCE1 was done in *E. coli* cells engineered for expression of recombinant proteins, i.e based on *E. coli* BL21 (DE3). In these cells several genes encoding proteases were genomically deleted. Acetylated SnCE1 WT is not processed, deacetylated WT is processed and non-acetylated Y212A and C256A mutants are also processed. This strongly supports our model according to which deacetylation of SnCE1 drives autoproteolysis of SnCE1. We performed mass spectrometry to identify potential protease contaminations in our protein preparations. These data did not show presence of any protease contamination in the samples. We added the table as Supplementary Data 2.

According to the HEK cells, we cannot totally exclude that an external protease does contribute to proteolytic processing but from our data we think that autoproteolysis takes place. However, the data including expression of the non-acetylated Y212A mutant, which shows stronger processing compared to WT, also strongly supports that autoproteolysis is modulated by the acetylation state of SnCE1 by an endogenous process.

We wrote in the manuscript (lines 200-201):

"If not otherwise stated we used the construct SnCE1₁₇₄₋₃₁₀ for performing experiments."

and (lines 699-700):

"As support for an autoproteolytic processing, we did not identify any exogenous protease in the samples as by LC-MS/MS (Supplementary Data 2)."

Point 4:

Out of the five catalytic domain homologs of SnCE1 in *Simkania negevensis*, only the SnCE1 shows deconjugating and acetyltransferase activity. Have the authors compared the SnCE1 crystal structure with the AlphaFold models of SnCE2-5? Is there any structural explanation for their lack of activity?

Response:

We prepared AlphaFold3 models of SnCE1-5 in the apo states and bound to CoA (Supplementary Fig. 16). Structural alignments and primary sequence alignments show that Tyr212 which we show is essential for the AcT activity of SnCE1 is not present in SnCE2-5. Only SnCE3 has an aromatic side chain at the analogous position, i.e. F301. Besides, for several CE-clan related virulence factors orphan functions were described apart from their catalytic activity. This does not exclude the possibility that for SnCE2-5 the physiological activity and substrate specificity has just not been identified so far. In SnCE1-5 all residues of the catalytic triad (Cys-His-Asp) are totally conserved. Only in SnCE3 the Asp on the third position of the triad is replaced by a Gly with two Asn side chains directly C-terminal of this residue. In enzymes using a catalytic triad the third residue is not always totally conserved and its role can sometimes also be fulfilled by the protein's main chain or by other side chains positioning the base His. To this end, the missing Asp in the triad in SnCE3 does not rule out a catalytic activity as protease. Notably, in SnCE2-5 the aromatic gatekeeper, i.e. Trp191 in SnCE1, which mediates selectivity towards ubiquitin or Ubl is missing suggesting that these CE-clan members might act as proteases cleaving normal peptide bonds having non-ubiquitin/Ubl substrates which await their identification. We added Supplementary Fig. 16 showing superpositions of the AF3 models of SnCE1-5 and the complexes with CoA. We analysed their structural similarity by performing super-secondary structure matching (SSM) by Coot and showed the r.m.s.d. values.

We wrote in the manuscript (lines 505-509):

"AlphaFold3 structure predictions apo-SnCE1-5 and CoA-bound SnCE1-5 and superposition of these structural models suggest that all proteins share a CE-clan protease fold with varying degree of similarity towards SnCE1 (Supplementary Fig. 16). Moreover, key residues needed for CoA binding in SnCE1, including Tyr212, are missing in SnCE2-5 providing a potential explanation for the lack of AcT activity for SnCE2-5 (Supplementary Fig. 16)."

and (lines 1028-1034):

"This discovery might have implications also for SnCE2-5 for which no enzymatic activity could be identified so far maybe exerting a dominant negative effect³¹. Notable, SnCE2-5 lack the aromatic gatekeeper, i.e. Trp191 in SnCE1, known to mediate specificity towards ubiquitinated/SUMOylated substrates indicated these might be proteases with other unknown substrates (Supplementary Fig. 12, 16). Along this line, for other eukaryotic DUBs/Ubls functions were reported which were independent of their catalytic activities^{97,98}."

Point 5:

In the crystal structure descriptions (line 207), it would be good to know the extension of the SnCE1 chain observed in the electron density maps. Particularly if there are missing regions (flexible to be observed in the crystal structure, but present in the protein).

Response:

The apo-wild type structure obtained for wild type SnCE1 was obtained with the construct SnCE1₁₀₄₋₃₁₀, while the structure of catalytically dead SnCE1 C256A and the complex of SnCE1 with SUMO1 was obtained with the construct SnCE1₇₄₋₃₁₀. Especially the flexible N-terminal region (aa 74-107) is not resolved completely in all three crystal structures, starting at either K106, D107 or R108 (confidently modelled). For the C-terminal part in both apo structures the last 5-7 aa are not resolved, but can be confidently modelled in the SUMO1 co-structure due to stabilization by interaction with SUMO1. Moreover, the exact final or beginning residues differ in the multiple chains representing the asymmetric unit. In all deposited structures all parts of the structures were shown which are reliably supported by the experimental data. These were deposited in the protein data bank and will be released upon publication of the manuscript.

Point 6:

The deSUMOylation activities in Fig1 do not seem to be very strong. Have the authors checked, a little bit more quantitatively, those activities, for example compared to other CE-clan members or with yeast or human SENPs.

Response:

Our data and the data reported earlier shows that SnCE1 cleaves SUMO2-chains much more efficiently compared to ubiquitin-chains of various linkage types. To enable a more quantitative comparison with other known CE-clan members, we conducted a kinetics experiment directly comparing the deSUMOylase activities of SnCE1 and human SENP1, SENP6 and yeast Ulp1 (Supplementary Fig. 3a). These studies reveal that all deSUMOylases compared including SnCE1 cleave SUMO2-chains with similar efficiency. However, for SENP1, SENP6 and Ulp1 the chains were cleaved slightly more efficiently compared to SnCE1 as particularly obvious at the early time points of the kinetics. The data were depicted in Supplementary Fig. 3a.

We wrote in the manuscript (lines 215-219):

“Compared to well-known deSUMOylases human SENP1, human SENP6 and yeast Ulp1, SnCE1 cleaves SUMO2_[2-6]-chains slightly less efficient (Supplementary Fig. 3a). Presence of a reducing agent such as β -mercaptoethanol or TCEP not containing thiol-groups was essential for SnCE1 catalytic deSUMOylase activity (Supplementary Fig. 3b).”

Point 7:

Why do the authors check Ser208 in Fig2 ?

Responses:

When we observed the autoproteolytic processing of SnCE1 we studied a possible underlying mechanism. Our data show that the deSUMOylase/Act inactive mutant SnCE1 C256A shows proteolytic processing. Moreover, we reproducibly observed that SnCE1 C256A was capable to inefficiently cleave SUMO2-chains (see Fig. 2b). This fueled our assumption that only in the background of the C256A mutant, the Ser208 can replace Cys256 acting as nucleophile catalyzing inefficient cleavage of SUMO2-chains as the mutant SnCE1 S208A shows deSUMOylase activity comparable to SnCE1 WT. By exchanging a bulkier Cys to Ala the active site architecture changes possibly resulting in a state in which Ser208 is capable to take over as nucleophile to little extent. The Ser208 is directly located in the active site in contact distance to the catalytic His-base why we assumed this might be a nucleophile of importance for autoproteolytic processing. However, our results show that Ser208 is not involved in proteolytic cleavage.

We wrote in the manuscript (lines 271-273):

“Notably, for SnCE1 C256A we observed slight remaining activity after 60 min suggesting that Ser208 can replace Cys256 as nucleophile to some extent, however, only in the context of the C256A mutant (Fig. 2b). As support, the double mutant SnCE1 S208A/C256A is catalytically inactive (Fig. 2b).”

Point 8:

Since tetramerization impacts the deSUMOylation activity (Fig 3b), the WT the SEC profile is clearly shifted towards the monomer (in contrast to the C256A mutant), but I would expect an equilibrium. Any evidence that the WT can also form a tetramer?

Response:

We would like to refer to the answer given in response to point 2.

Point 9:

Also regarding the tetramer, in the presence of the complete N-terminal segment, is the tetramer formed? Even with a transmembrane region? Could the authors speculate whether the tetramer can be formed when attached to the ER membrane?

Response:

The SnCE1₇₄₋₃₁₀ as well as the shortest construct SnCE1₁₀₄₋₃₁₀ are able to form a tetramer in solution if both are on the background of the C256A mutation abolishing acetylation of SnCE1. This suggests tetramer formation is mediated by the C-clan protease catalytic domain. To this end, we assume that the highly flexible N-terminal region most likely does not contribute to tetramer formation. Along that line, AlphaFold3 structure predictions do also not indicate that the flexible N-terminal region somehow directly binds to the catalytic domain. Moreover, in our experimental structure of the SnCE1 C256A tetramer, the N-terminus is pointing towards the outside and not towards the interfaces of the tetramer. Our analytical SEC experiments show the tetramer eluting in an equilibrium with a monomer suggesting to be of moderate affinity/stability. However, membrane binding of SnCE1 mediated by the N-terminal region might significantly improve tetramer formation by a concept known as dimensionality reduction first reported by Max Delbrück (cytosolic 3D space versus membrane-bound 2D) as reported for other membrane binding proteins. Our data show that acetylation at K231 in SnCE1 can shift the equilibrium towards the monomeric form enabling regulation of the oligomeric state according to the metabolic state of the cell. Whether the tetramer can be formed at the ER membrane might therefore depend on the acetylation state of SnCE1.

Point 10:

In Figure 4E, the different N-terminal constructs have different acetylation levels (in particular SnCE1 95-310). Any comments on that?

Response:

We agree with the reviewer and performed additional experiments to show the acetylation signals from the different SnCE1 constructs including the N-terminal deletion constructs and the SnCE1 mutants (Fig. 4e). The reproducibly strongly impaired signal obtained by immunoblotting using an anti-acetyl lysine antibody for SnCE1 H190A suggests that this mutation catalytically impairs the AcT activity. It was reported earlier that the His-base of the catalytic triad is important for the catalytic AcT mechanism in YopJ-effectors (Zhang et al., (2017) Nature Plants 10.1038/nplants.2017.115). It is needed to deprotonate the substrate's lysine side chain to enable a nucleophilic attack on the acetyl-cystein intermediate formed during catalysis. Moreover, for the gatekeeper mutant SnCE1 W191A and the mutants SnCE1 K231R and SnCE1 E217D K231R we also observed a lower signal intensity in immunoblotting using an anti-acetyl lysine antibody. This suggests that these mutations slightly modulate the SnCE1 conformation interfering with other properties such as acetyl-CoA binding. For the SnCE1 truncation variants we observe the strongest signal for the longest construct SnCE1₇₄₋₃₁₀ supporting our mass spectrometry data showing some acetylation sites to reside in the N-terminal region covering residues 74-104.

We adapted the text accordingly (lines 464-464):

“The mutant SnCE1 H190A shows an almost non-detectable autoacetylation, suggesting a catalytic role also for the AcT activity, as reported earlier (Fig. 3d; Fig. 4e)⁴⁶. According to the AlphaFold3 model the CoA occupies the S1'-site with the CoA's sulfhydryl group capable to form a disulfide with the active site Cys256 (Fig. 4d; Supplementary Fig. 16).

and (lines 485-502):

„To this end, the mutation Y212A in SnCE1 might affect the precise orientation and position of $\alpha 3$ thereby reducing the propensity of tetramer formation. Interestingly, acetylation signals for the mutants SnCE1 K231R, SnCE1 E217D K231R and of the gatekeeper SnCE1 W191A were strongly reduced suggesting a potential structural role interfering with acetyl-CoA/CoA binding (Fig. 4e).”

Point 11:

Perhaps one of the more intriguing questions is the autoproteolytic activity. I know that it is already discussed, but DUBs or ULPs are highly specific for Ub/Ubl substrates. It is hard to understand autoproteolytic activity using the catalytic triad in the active site. But the presence of a different active site, from an unstructured region (N-terminal extension), are also very hard to imagine. A non-autoproteolytic activity is completely discarded? Other hypothesis.

Response:

We initially speculated that the active site would also mediate autoproteolytic processing, however, our mutational analyses clearly show that it is likely not directly involved in autoproteolysis as the mutant SnCE1 C256A and the double mutant SnCE1 S208A C256A, in which both of the potential nucleophiles were mutated, were still autoproteolytically processed. Some publications describe autoproteolytic cleavage mechanisms *in cis* using Ser/Thr/Cys as nucleophiles cleaving the preceding peptide bond in a mechanism resembling self-splicing inteins (Buller et al. (2011) doi/10.1073/pnas.1113633109). We observed a strong enrichment of Ser/Thr/Cys residues in the region we expect autoproteolysis to occur. We also analysed a mutant in which several Ser/Thr/Cys residues in this region which could play the role as nucleophile were mutated to Ala (mutant: SnCE1

C256A S74A/T76A/S77A/C81A/S84A/S86A/S87A/S90A). However, this mutant was also still processed suggesting a different nucleophile/mechanism being responsible for autoproteolytic processing. It is quite speculative but it might also be possible that the nucleophile is supplied by a different part of the protein or that tetramerization of SnCE1 plays a role in processing and that the nucleophile is supplied *in trans*. From our perspective it is important that the catalytic residues involved in autoproteolytic cleavage and the substrate residues are stabilized, correctly and precisely oriented and positioned for efficient autoproteolysis to occur. We are trying to elucidate the underlying mechanism as this might also be important for other proteins including virulence factors.

Reviewer #2 (Remarks to the Author):

Point 1:

Line 190-193: We confirmed, although less efficiently reacting with the Ub-PA probe compared to the SUMO1/3-PA probes, SnCE1 shows weak DUB activity cleaving K11-, K48- and K63-linked di-ubiquitin (diUb) chains. Based on Fig 1a, I can't see the band corresponding to SnCE1-Ub, questioning the capability of Ub-PA to react with SnCE1. Also, for WT(-) lane of SUMO3-PA probe, there seems to be a band corresponding to SnCE1-SUMO3, which should not be possible. For Fig 1b, there seems to be a band corresponding to Ub for M1/M27/M29/K33 as well, indicating the capability of SnCE1 in cleaving these linkages. However, it would be helpful to provide a comparison of each dimer with and without SnCE1 for more objective analysis. At the moment, negative controls are missing (i.e., Ub dimers only without SnCE1).

Response:

We agree the reactivity of SnCE1 with the Ub-PA probe is very inefficient compared to SUMO-PA probes. We only observed very low amount of crosslinking product for Ub-PA with SnCE1₁₀₄₋₃₁₀ as visible by the very faint band in the SDS-PAGE gel (Fig. 1a). The band representing SnCE1-Ub-PA is marked in Fig. 1a (upper panel). Moreover, we also see some remaining GST-SnCE1₇₄₋₃₁₀ fusion protein in the first two lanes which is also reacting slightly with the Ub-PA probe. These lanes were also marked (Fig. 1a). Along that line, we also observed the low efficiency of SnCE1 to cleave ubiquitin chains confirming earlier studies (Boll et al (2023) doi.org/10.1038/s41467-023-43144-y). To this end, we assume the SnCE1 DUB activity might be of lower physiological relevance compared to its deSUMOylase activity. The band visible in the WT(-) lane of SUMO3-PA in Fig. 1a is an impurity left from the purification. It is visible in all crosslinking experiments in the samples without addition of Ub-PA, SUMO1-PA or SUMO3-PA, showing that it is present in the SnCE1 wild type preparation.

We repeated the experiment shown in Fig. 1b and loaded control samples without SnCE1 next to the samples including SnCE1 for all diUb-chains as suggested by the reviewer. This reveals that SnCE1 is indeed capable to cleave M1/K27/K29/K33-diUb chains albeit with even lower efficiency compared to K63-diUb chains strongly supporting earlier studies that SnCE1 is primarily a deSUMOylase with little DUB activity inefficiently cleaving Ub-chains. We replaced Fig. 1b with the updated Fig.

Point 2:

Fig. 3a, the amount of C256A mutant is much less than the wt protein, consequently, the lack of AcK signals in IB may be due to lower concentration of the protein instead of the lack of catalytic activity.

Response:

The reviewer is right in saying that less SnCE1 C256A protein was loaded compared to SnCE1 WT in Fig. 3a, however, we show clearly in this manuscript also in other experiments that the mutation of the catalytic Cys, i.e. C256A, does completely abolish its acetyltransferase activity. For Fig. 3a we would expect that a signal in the anti-AcK immunoblot would be visible for SnCE1 C256A even at these amounts loaded as the antibody does detect SnCE1 acetylation with high level of sensitivity and intensity.

Point 3:

C256A mutant seems to be able to cleave SUMO2 in Fig 5b but not in Fig 2b.

Response:

The reviewer is right with the observation. We reproducibly observed that the mutant SnCE1 C256A shows almost no activity cleaving SUMO2-chains but there is some residual activity. Our hypothesis is that only in the background of the C256A mutant, the Ser208 can act as nucleophile catalyzing inefficient cleavage of SUMO2-chains as the single mutant SnCE1 S208A shows deSUMOylase activity comparable to SnCE1 WT. By exchanging a bulkier Cys to Ala the active site architecture changes possibly resulting in a architecture in which Ser208 is capable to take over as nucleophile to little extent. Ser208 is also in interaction distance to the active site base His190. To confirm this, the double-mutant SnCE1 S208A C256A is completely inactive (Fig. 5b). The fact we only observed the mono-SUMO in the sample with SnCE1 C256A in Fig. 2b but not in Fig 5a (not 5b as stated by the reviewer) might just be due to less loading of the SUMO2-chains or less efficient transfer in immunoblotting.

We wrote in the manuscript (lines 271-273):

"Notably, for SnCE1 C256A we observed slight remaining activity after 60 min suggesting that Ser208 can replace Cys256 as nucleophile to some extent, however, only in the context of the C256A mutant (Fig. 2b). As support, the double mutant SnCE1 S208A/C256A is catalytically inactive (Fig. 2b)."

Point 4:

There is no intact mass data for all recombinant proteins (including those with AcK), consequently difficult to judge whether the expressed proteins have the expected amino acid sequence.

Response:

Analysis of all recombinant proteins prepared in this study by SDS-PAGE and analytical size-exclusion chromatography (SEC) reveal that all proteins are pure and they behave as expected according to their expected molecular weights. We understand

that it is important to provide data for the intact masses for site-specifically acetylated proteins while it is unusual to provide intact masses for all recombinant proteins. However, we performed mass spectrometry to determine the intact masses of all proteins presented in this study. For all recombinantly expressed proteins we were able to deduce the correct molecular masses (Supplementary Data 3 and 4). However, for the proteins SnCE1 Y212A Ack231 and SnCE1 C256A H190A we could not assign the correct masses although both proteins behaved as expected in SDS-PAGE analyses and analytical SEC experiments. Importantly, results obtained with these two proteins were not used to draw any conclusions important for this study. Notably, we observed also modification of cysteine side chains by formation of adducts with β -mercaptoethanol (+76 Da). This is a well known modification observed in intact mass determination of proteins (Xing et al (2008) 10.1021/pr800456q; Orsatti et al (2002) 10.1002/rcm.814). SnCE1 has a total of three Cys side chains, two are located in the unstructured N- and C-terminal regions, respectively, and the third is the active site Cys. We did not observe modification of the active site Cys in the structure of SnCE1 wild type. The fact that reactive Cys side chains were present in CE-clan protease related virulence factors was also described earlier for ChlaDUB1 (Fischer et al. (2017) 10.7554/eLife.21465). To exclude that a potential modification by β -mercaptoethanol would interfere with catalytic activity of SnCE1, we purified SnCE1 and SnCE1 C256A without reducing agent and with TCEP as alternative reducing agent lacking thiol groups and performed a deSUMOylase assay. These data show that presence of a reducing agent such as β -mercaptoethanol or TCEP is necessary as the protein without reducing agent is not capable to cleave SUMO2-chains (Supplementary Fig. 3b).

We wrote in the manuscript (lines 217-219):

“Presence of a reducing agent such as β -mercaptoethanol or TCEP not containing thiol-groups was essential for SnCE1 catalytic deSUMOylase activity (Supplementary Fig. 3b).”

Point 5:

Writing and readability can be improved. At the moment, it seems to target very specialized audience.

Response:

We agree with the reviewer and worked on the readability of the manuscript. We shortened the introductory and further sections and subdivided some paragraphs to allow a better readability as visible in the manuscript file with track changes.

Reviewer #3 (Remarks to the Author):

Point 1:

In the section Acetylation of SnCE1 at K231 is irreversible and mediates SnCE1 monomerization beginning on line 412 the authors refer the reader to Supplemental Figure 10. Can the authors confirm that the correct figure is being used – this figure only goes to aa 205 but the manuscript is discussing sites K209, K248 and beyond.

Response:

We thank the reviewer for stating this. We noticed that the numbering was not correct for the alignments shown in Supplementary Fig. 5 and Supplementary Fig. 10 (in revised manuscript Supplementary Fig. 7 and 12). In both alignments only the catalytic CE-clan protease domains were shown as the N-terminal regions are regarding their length and sequence quite diverse. We adjusted the numbering for both alignments.

Point 2:

The authors state that there are regions in the SnCE1 protein that are difficult to detect by mass spectrometry because of “long peptides” after tryptic digest. A tryptic peptide of 20 aa and 21 aa is not considered to be a challenge to sequence by MS/MS and is still in the ideal range of amino acids in a peptide. I would remove that statement and just leave that you chose the sites for incorporated acetyl-lysine as just simply either previously identified by mass spec or present in functionally or structurally important regions in SnCE1.

Response:

We thank the reviewer and removed the sentence as suggested.

In this revised article we worked concisely on the final open points addressed by reviewer 2. Find below the response to these open points.

Point-by-point response to the open points raised by reviewer 2:

Point 1: Nevertheless, it would also be helpful if the authors could reproduce the relevant changes more explicitly in the response letter, so that I do not need to move back and forth repeatedly between the manuscript and the supplementary data to identify the exact revisions. While I have supported publication of the manuscript from the very beginning, these obstacles in the review process have greatly diminished my enthusiasm.

Response: We thought that it would be possible to reproduce what we changed. The problem is that every reviewer had some points that needed correction so that it might have become rather complex. However, we stated all changes with exact line numbers in the point-by-point response all referring to the document we submitted that included track-changes to be able to follow our changes. In the document with track-changes we highlighted text that was deleted in yellow and the text that was modified in cyan to allow to get a better overview about the changes. Moreover, all figures in the main text and supplementary information were cited. We totally agree with the reviewer that it might have been difficult to follow the changes we have made and apologize for the circumstances caused.

Point 2: The concern was straightforward: the C256A lane contained substantially less protein, so the absence of an AcK signal could simply reflect underloading. The authors do not provide a corrected blot with matched loading, nor do they present quantitative normalization demonstrating that the conclusion is robust. Instead, they argue that the antibody is sensitive and that other experiments elsewhere support loss of acetyltransferase activity. That is an indirect defence, not a direct response to the criticism. The clean solution would have been to repeat the experiment with equal loading and matched exposure.

Response: We apologize for not repeating the immunoblotting experiment (Fig. 3a). We think that our conclusions drawn for this experiment are valid though. The experiment shown here and throughout the manuscript reveal that the mutant SnCE1 C256A is not acetylated. The main message of this experiment, the reviewer refers to, is rather that the acetylation signal increases upon addition of acetyl-CoA confirming that it is the donor-molecule for the acetylation. This is not the case for the catalytically inactive mutant SnCE1 C256A showing the autoacetyltransferase activity.

Point 3: Okay, but the description “Presence of a reducing agent such as beta-mercaptoethanol or TCEP not containing thiol-groups was essential for SnCE1 catalytic deSUMOlyase activity” is ambiguous, because it appears to imply that beta-mercaptoethanol does not contain a thiol group.

Response: We thank reviewer 2 for carefully reading our manuscript. We have re-written the sentence as requested: „Presence of a reducing agent such as β -mercaptoethanol or TCEP, the latter not containing a thiol-group,…”

Point 4: The manuscript still contains many language and presentation problems. Some examples, although not a comprehensive list, are given below. Overall, the manuscript still requires thorough editing for grammar, spelling, and formatting.

Response:

We worked on the manuscript and corrected errors of grammar and typos as far as we could with our knowledge on English grammar.

Individual points:

- A) "Gram-negative bacteria use plethora of virulence factors" should be corrected to "Gram-negative bacteria use a plethora of virulence factors."

Response: corrected as suggested

- B) "Treatment of cells with deacetylase inhibitors, i.e. SAHA and TSA, to inhibit classical HDACs and nicotinamide (NA) to inhibit sirtuins, show the acetylation of SnCE1 increases" should be corrected to "Treatment of cells with deacetylase inhibitors, i.e. SAHA and TSA to inhibit classical HDACs and nicotinamide (NA) to inhibit sirtuins, shows that SnCE1 acetylation increases."

Response: corrected as suggested

- C) "Although, the mutant SnCE1 Y212A shows that the SnCE1 acetylation state directly affects autoproteolytic processing" should be corrected to "Although the mutant SnCE1 Y212A shows that the acetylation state of SnCE1 directly affects autoproteolytic processing."

Response: corrected as suggested

- D) "The fact, SnCE1 Y212A shows a stronger processing compared to the likewise monomeric wild type SnCE1" should be corrected to "The fact that SnCE1 Y212A shows stronger processing than the likewise monomeric wild-type SnCE1."

Response: corrected as suggested

- E) "different molecular strategies underly this dual AcT/ULP-activity" should be corrected to "different molecular strategies underlie this dual AcT/ULP activity."

Response: corrected as suggested

- F) For typos, "whichstage" should be corrected to "which stage." For formatting, "SnCE1wild type" should be corrected to "SnCE1 wild type."

Response: corrected as suggested